# Fairness via Independence: A General Regularization Framework for Machine Learning

**Yezi Liu & Hanning Chen & Wenjun Huang**
Department of Electrical Engineering and Computer Science
University of California, Irvine
Irvine, CA, USA
{yezil3,hanningc,wenjunh3}@uci.edu

**Yang Ni**
Purdue University Northwest
Hammond, IN, USA
yangni@purdue.edu

**Mohsen Imani**[*]
Department of Electrical Engineering and Computer Science
University of California, Irvine
Irvine, CA, USA
m.imani@uci.edu

## Abstract

Fairness in machine learning has emerged as a central concern, as predictive models frequently inherit or even amplify biases present in training data. Such biases often manifest as unintended correlations between model outcomes and sensitive attributes, leading to systematic disparities across demographic groups. Existing approaches to fair learning largely fall into two directions: incorporating fairness constraints tailored to specific definitions, which limits their generalizability, or reducing the statistical dependence between predictions and sensitive attributes, which is more flexible but highly sensitive to the choice of distance measure. The latter strategy in particular raises the challenge of finding a principled and reliable measure of dependence that can perform consistently across tasks. In this work, we present a general and model-agnostic approach to address this challenge. The method is based on encouraging independence between predictions and sensitive features through an optimization framework that leverages the Cauchy–Schwarz (CS) Divergence as a principled measure of dependence. Prior studies suggest that CS Divergence provides a tighter theoretical bound compared to alternative distance measures used in earlier fairness methods, offering a stronger foundation for fairness-oriented optimization. Our framework, therefore, unifies prior efforts under a simple yet effective principle and highlights the value of carefully chosen statistical measures in fair learning. Through extensive empirical evaluation on four tabular datasets and one image dataset, we show that our approach consistently improves multiple fairness metrics while maintaining competitive accuracy.

## 1 Introduction

Fairness in machine learning has garnered growing concern, as machine learning (ML) models are playing key roles in many high-stakes decision-making scenarios, such as credit scoring (Leal, 2022), the job market (Hu & Chen, 2018), healthcare (Grote & Keeling, 2022; Xu et al., 2025), and education (Bøyum, 2014; Kizilcec & Lee, 2022). Among the various fairness notions, group fairness is one of the most extensively studied ones as it addresses the prediction disparities across demographic groups, including gender, age, skin color, and region (Mehrabi et al., 2021; Dwork et al., 2012; Barocas et al., 2017). While many group fairness ML algorithms are proposed, they have challenges in their applications, especially the *generalizability*, i.e., their adaptation to different fairness notions, and *robustness*, i.e., the stability of the fairness when they encounter a slight change of model parameter.

---

[*]Corresponding author.

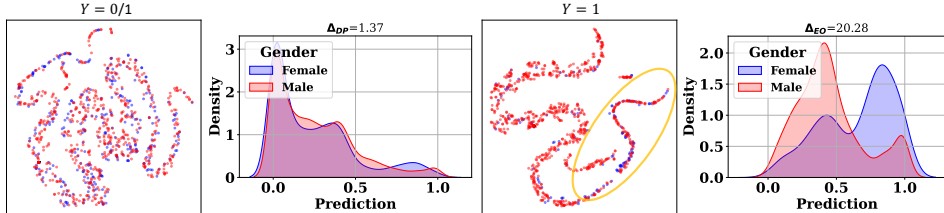

Figure 1: From left to right: (1) Prediction distribution of *all classes*; (2) T-SNE plot of embeddings for samples from *all classes*; (3) Prediction distribution of *class 1*; (4) T-SNE plot of embeddings for samples from Adult, and the sensitive attribute is gender. The blue points represent samples with sensitive attribute 0, while the red points represent samples with sensitive attribute 1.

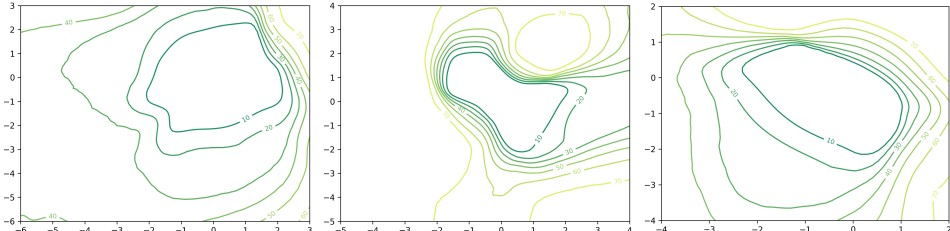

Figure 2: Fairness loss landscapes evaluated using three functions, presented from left to right: Kullback-Leibler (KL) divergence, Hilbert-Schmidt Independence Criterion (HSIC), and Cauchy-Schwarz (CS) divergence. A smaller inner circle indicates greater robustness. Among these methods, the CS divergence achieves the smallest inner circle, ranging from $-2$ to $1$, while the inner circles of KL and HSIC divergences both span from $-2$ to $2$.

Existing group fairness approaches can be intrinsically categorized into two main approaches based on their debiasing objectives: *i)* directly integrate the fairness notion into the training objective, *ii)* minimizing the correlation between predictions and sensitive attributes. Methods belongs to *i)* such as a demographic parity (DP) regularizer, and an equality of opportunity (EO) regularizer. The benefit of this approach is that the model trained by the target fairness objective can perform well on specific fairness notions. For example, the machine learning model trained at demographic parity has a high possibility of achieving good demographic parity in testing. However, such methods limited their generalizability to other fairness notions (shown in Figure 1). Method *ii)* solves from a more fundamental way that can deal with generalizability, including using information theory, or an adversarial approach to minimize the correlation between the prediction and the sensitive attribute. The most straightforward way is to use a distance measurement that assesses the relationship between the sensitive attribute and the prediction, thus minimizing this distance during the training. This enables the generalizability, but ascribes the pressure of the fairness performance to the quality of the distance measurement.

Existing fairness regularizers mainly assessed this correlation using gap parity, Maximum Mean Discrepancy (MMD) (Gretton et al., 2012), Kullback-Leibler (KL) divergence, and the Hilbert-Schmidt Independence Criterion (HSIC). However, the current fairness regularizers are sensitive to the model parameter change, making them less robust in maintaining fairness, responding to a small change in the model parameters (shown in Figure 2). Theoretical studies have shown that the Cauchy-Schwarz divergence provides a tighter bound compared to the Kullback-Leibler divergence and gap parity, suggesting its potential to improve fairness in machine learning models. Motivated by this, we would like to know if using CS divergence can result in more generalizable and more consistent utility–fairness trade-off across hyperparameter settings than standard regularizers due to the benefit of a tighter upper bound of CS divergence. In light of this, we propose a new fairness regularizer based on the Cauchy-Schwarz divergence for fair machine learning. To evaluate the generalizability, we tested the fairness under a wide range of fairness notions proposed by previous studies, and to evaluate the robustness, we visualize whether a small change in the learned model parameters can influence the fairness. We summarize our contributions as follows:

- We introduce the Cauchy-Schwarz divergence to fair machine learning and present a novel regularization method.

- We elucidate the relationships between the Cauchy-Schwarz regularizer and other fairness regularizers, emphasizing its superior effectiveness in debiasing.

- Our experimental results, obtained from four tabular datasets and one image dataset, validate the efficacy of the proposed Cauchy-Schwarz regularizer in achieving fairness across multiple fairness notions simultaneously.

This develops CS divergence in a fairness-specific setting that is complementary to prior work on CS-based density estimation, clustering, and representation learning.

## 2 PRELIMINARIES

In this section, we establish the foundational concepts for our study. We start by exploring the notion of fairness in machine learning, including the relevant notations. Next, we provide an overview of general fairness-aware machine learning methods. Finally, we introduce the Cauchy-Schwarz divergence and discuss its benefits in reducing bias.

**Problem scope.** This paper focuses on in-process group fairness in the context of binary classification with binary-sensitive attributes. In contrast, group fairness seeks to ensure that machine learning models treat different demographic groups equitably, where groups are defined based on sensitive attributes such as gender, race, and age (Feldman et al., 2015; Zemel et al., 2013).

**Notations.** Under this setting, we consider a dataset $\mathcal{D} = \{(\mathbf{x}_i, y_i, s_i)\}_{i=1}^{M}$, where $M$ is the number of samples, $\mathbf{x}_i \in \mathbb{R}^d$ represents the features excluding the sensitive attribute, $y_i \in \{0, 1\}$ is the label of the downstream task, and $s_i \in \{0, 1\}$ is the sensitive attribute of the $i$-th sample. The predicted probability for the $i$-th sample is denoted as $z_i \in [0, 1]$, computed by the machine learning model as $z_i = f(\mathbf{x}_i, s_i) : \mathbb{R}^d \to [0, 1]$. The binary prediction is represented as $\hat{y}_i \in \{0, 1\}$, defined by $\hat{y}_i = \mathbf{1}_{\{\}} \geq t[z_i]$, where $\mathbf{1}_{\{\}} \geq t(\cdot)$ is the indicator function that evaluates whether its input is greater than or equal to the threshold $t$. Finally, $X$, $Y$, $S$, and $\hat{Y}$ denote random variables corresponding to $\mathbf{x}_i$, $y_i$, $s_i$, and $\hat{y}_i$, respectively.

**Problem Formulation.** Generally, the fairness objective can be summarized as follows:

$$\min_{f} \quad \mathcal{L}_{utility} + \lambda \mathcal{L}_{fairness}, \tag{1}$$

where the term $\mathcal{L}_{utility}$ denotes the loss function that measures the utility of the model, often a binary entropy loss, for our binary classification problem, while $\mathcal{L}_{fairness}$ (also shown in Figure 2) indicates the fairness constraint applied in the model. The parameter $\lambda$ is used to control the trade-off between utility and fairness.

### 2.1 GROUP FAIRNESS

There are many ways to define and measure the group fairness. Each definition focuses on distinct statistical measures aimed at achieving balance among subgroups within the data. Among these, Demographic Parity and Equal Opportunity are the most popular ones, other popular fairness definitions are summarized in Appendix C.2.

**Demographic Parity (DP).** Demographic Parity (Zafar et al., 2017; Feldman et al., 2015; Dwork et al., 2012; Liu et al., 2026) mandates that the predicted outcome $\hat{Y}$ be independent of the sensitive attribute $S$, expressed mathematically as $\hat{Y} \perp S$. Most of the existing literature primarily addresses binary classification and binary attributes, where $Y \in \{0, 1\}$ and $S \in \{0, 1\}$. Similar to the concept of equal opportunity, the metric evaluating the DP fairness is defined by:

$$\triangle_{DP} = |P(\hat{Y}|S = 0) - P(\hat{Y}|S = 1)|. \tag{2}$$

A lower value of $\triangle_{DP}$ signifies a fairer classifier.

**Equal Opportunity (EO).** Equal Opportunity (Hardt et al., 2016) mandates that a classifier achieves equal true positive rates across various subgroups, striving towards the ideal of a perfect classifier. The corresponding fairness measurement for EO can be articulated as follows:

$$\triangle_{EO} = |P(\hat{Y}|Y = 1, S = 0) - P(\hat{Y}|Y = 1, S = 1)|. \tag{3}$$

A low $\triangle_{EO}$ indicates that the difference in the probability of an instance in the positive class being assigned a positive outcome is relatively small for both subgroup members. Both DP and EO can be effectively extended to problems involving multi-class classifications and multiple sensitive attribute

categories. Note that in the binary classification task, $\triangle_{DP}$ and $\triangle_{EO}$ are sometimes calculated after binarizing $P(\hat{Y})$. Specifically, $\triangle_{DP}$ is defined as $\triangle_{DP} = |P(\hat{Y} = 1|S = 0) - P(\hat{Y} = 1|S = 1)|$, and $\triangle_{EO}$ is defined as $\triangle_{EO} = |P(\hat{Y} = 1|Y = 1, S = 0) - P(\hat{Y} = 1|Y = 1, S = 1)|$ (Beutel et al., 2017; Dai & Wang, 2021; Dong et al., 2022).

## 2.2 CAUCHY-SCHWARZ DIVERGENCE

Motivated by the well-known Cauchy-Schwarz (CS) inequality for square-integrable functions[1], which holds with equality if and only if $p(\mathbf{x})$ and $q(\mathbf{x})$ are linearly dependent, we can define a measure of the distance between $p(\mathbf{x})$ and $q(\mathbf{x})$. This measure is referred to as the CS divergence (Principe et al., 2000; Yu et al., 2023), given by:

$$D_{\text{CS}}(p; q) = -\log\left(\frac{\left(\int p(\mathbf{x})q(\mathbf{x})dx\right)^2}{\int p(\mathbf{x})^2 dx \int q(\mathbf{x})^2 dx}\right). \tag{4}$$

The CS divergence, denoted as $D_{\text{CS}}$, is symmetric for any two probability density functions (PDFs) $p$ and $q$, satisfying $0 \le D_{\text{CS}} < \infty$. The minimum divergence is achieved if and only if $p(\mathbf{x}) = q(\mathbf{x})$.

## 3 WHAT MAKES A GOOD FAIRNESS REGULARIZER?

Current fair machine learning algorithms adopt a variety of approaches, prompting us to explore the essential properties that make for effective fairness regularizers. By categorizing these algorithms into three types and conducting preliminary experiments, we aim to identify the key characteristics that contribute to mitigating bias in machine learning models.

| Reg. | Fairness Objective ($\mathcal{L}_{fairness}$) |
|------|-----------------------------------------------|
| DP | $|\mathbb{E}(\hat{Y}|S = 0) - \mathbb{E}(\hat{Y}|S = 1)|$ |
| EO | $|\mathbb{E}(\hat{Y}|Y = 1, S = 0) - \mathbb{E}(\hat{Y}|Y = 1, S = 1)|$ |
| MMD | $D_{\text{MMD}}(Z_{S=0}, Z_{S=1})$ |
| HSIC | $D_{\text{HSIC}}(\hat{Y}, S)$ |
| PR | $D_{\text{PR}}(\hat{Y}, S)$ |

Table 1: Fairness regularizers (Reg.) and objectives.

### 3.1 BALANCING THE PREDICTION ACROSS DIFFERENT SENSITIVE GROUPS

The first is to directly integrate the fairness notions, such as DP and EO, into the fairness objective.

$$\mathcal{L}_{\text{fairness}} = D(\mathbb{P}, \mathbb{Q}) \quad \text{where}$$

$$\begin{cases} \mathbb{P} = P(\hat{Y} \mid S = 0), \quad \mathbb{Q} = P(\hat{Y} \mid S = 1) \text{ for DP}, \\ \mathbb{P} = P(\hat{Y} \mid Y = 1, S = 0), \ \mathbb{Q} = P(\hat{Y} \mid Y = 1, S = 1) \text{ for EO}. \end{cases} \tag{5}$$

Calculating the distance between $\mathbb{P}$ and $\mathbb{Q}$ has many ways by using difference distance measurement $D$, and the most used one is to calculate the absolute distance between the mean empirical estimations:

$$\mathcal{L}_{fairness} = |\mathbb{E}(\mathbb{P}) - \mathbb{E}(\mathbb{Q})|, \tag{6}$$

where the expected values are calculated as the mean of summation since $\mathbb{P}$ and $\mathbb{Q}$ are discrete distributions.

Previous fair machine learning studies have shown that the fairness loss of the testing can be upper bounded by the loss of training. Therefore, *a distance function having a tighter generalization error bound used in training will lead to a better fairness guarantee for testing.*

$$\mathcal{L}_{fairness} = |\mathbb{E}(\hat{Y}|S = 0) - \mathbb{E}(\hat{Y}|S = 1)|. \tag{7}$$

We can see that the basic idea is to balance the prediction distribution between two sensitive groups. Therefore, for this type of approach, we can also use other distance measurements: Therefore, except for the absolute distance between the two prediction distributions.

### 3.2 BALANCING THE LATENT REPRESENTATION ACROSS DIFFERENT SENSITIVE GROUPS

Distance measures and minimization:

$$\mathcal{L}_{fairness} = D(Z_{S=0}, Z_{S=1}), \tag{8}$$

where $Z$ is the latent representation from the neural networks, and $Z_{S=0}$ and $Z_{S=1}$ are the representation when the sensitive attribute is 1 or 0. The distance metric $D(\cdot)$ here can be a Mean Maximum Discrepancy (Louizos et al., 2016).

---

[1] $\left(\int p(\mathbf{x})q(\mathbf{x})\,dx\right)^2 \le \int p(\mathbf{x})^2\,dx \int q(\mathbf{x})^2\,dx$

**Extension to multiple sensitive attributes.** While our theoretical development is presented for a single sensitive attribute $S$ for notational simplicity, the proposed CS-based regularizer naturally extends to multiple sensitive attributes. Let $S = (S_1, \ldots, S_K)$ denote a vector of sensitive attributes. One can either (i) treat $S$ as a joint variable and apply the same CS divergence to $(\hat{Y}, S)$, i.e., penalize $\tilde{D}_{\mathrm{CS}}(P_{\hat{Y}|S}, P_{\hat{Y}}P_S)$, or (ii) sum the CS divergences over individual attributes, e.g., $\sum_{k=1}^{K} \tilde{D}_{\mathrm{CS}}(P_{\hat{Y}|S_k}, P_{\hat{Y}}P_{S_k})$, depending on whether joint or per-attribute control is desired. Both variants use exactly the same empirical estimator as in (Equation (13)) and do not require any algorithmic changes; only the definition of $S$ in the mini-batch needs to be updated.

## 3.3 Minimizing the Relationship Between Predictions and Sensitive Attributes

The goal of this category is to ensure that a fair machine learning algorithm's predictions retain minimal sensitive information.

$$\mathcal{L}_{fairness} = D(\hat{Y}, S), \tag{9}$$

The HSIC and PR in Table 1 belong to this category. Specifically, the $D_{\mathrm{PR}}(\hat{Y}, S)$ is defined as:

$$D_{\mathrm{PR}}(\hat{Y}, S) = \sum_{\hat{y} \in \hat{Y}} \sum_{s \in S} p(\hat{y}, s) \log\left(\frac{p(\hat{y}, s)}{p(\hat{y})p(s)}\right). \tag{10}$$

Note that, adversarial debiasing methods (Zhang et al., 2018) fall into this category because they employ discriminators to predict sensitive group membership from the learned encoded representations. These methods aim to make sensitive attributes difficult to deduce from the encoded representations.

## 4 Cauchy-Schwarz Fairness Regularizer

In this section, we first introduce three prominent fairness regularizers that assess distribution distance using different metrics: Mean Maximum Discrepancy, Kullback-Leibler divergence, and Hilbert-Schmidt Independence Criterion (HSIC). For each metric, we explore its relationship with CS divergence. Subsequently, we explain how CS divergence can be utilized to achieve fairness.

## 4.1 How Can the Cauchy-Schwarz Divergence Be Applied to Mitigate Bias?

Given samples $\{\mathbf{x}_i^p\}_{i=1}^m$ and $\{\mathbf{x}_i^q\}_{i=1}^n$ drawn independently and identically distributed (i.i.d.) from $p(\mathbf{x})$ and $q(\mathbf{x})$ respectively, we can estimate the empirical CS divergence. This estimation can be performed using the kernel density estimator (KDE) as described in (Parzen, 1962) and follows the empirical estimator formula in (Jenssen et al., 2006).

**Proposition 4.1.** *Given two sets of observations $\{\mathbf{x}_i^p\}_{i=1}^{N_1}$ and $\{\mathbf{x}_j^q\}_{j=1}^{N_2}$, let $p$ and $q$ denote the distributions of two groups. The empirical estimator of the CS divergence $D_{CS}(p; q)$ is then given by:*

$$\tilde{D}_{CS}(p; q) = \log\left(\frac{1}{N_1^2} \sum_{i,j=1}^{N_1} \kappa(\mathbf{x}_i^p, \mathbf{x}_j^p)\right) + \log\left(\frac{1}{N_2^2} \sum_{i,j=1}^{N_2} \kappa(\mathbf{x}_i^q, \mathbf{x}_j^q)\right)$$
$$- 2\log\left(\frac{1}{N_1 N_2} \sum_{i=1}^{N_2} \sum_{j=1}^{N_2} \kappa(\mathbf{x}_i^p, \mathbf{x}_j^q)\right). \tag{11}$$

The proof of this proposition is detailed in Appendix B.1. where $\kappa$ represents a kernel function, such as the Gaussian kernel defined as $\kappa_\sigma(x, x') = \exp(-\|x - x'\|_2^2/2\sigma^2)$. In the following sections, we will explore the relationship between this kernel function and the existing fairness regularizer.

As mentioned earlier, the goal of fairness is to ensure an equal distribution of predictions across sensitive attributes. To achieve this, fairness-aware algorithms focus on minimizing the dependency of predictions on these sensitive attributes. Therefore, effectively modeling the relationship between the outcome variable $Y$ and the sensitive attribute $S$ becomes crucial. The prediction distribution over the sensitive attribute $S$ is defined as follows:

$$\mathbb{P} = P(\hat{Y} \mid S = 0); \quad \mathbb{Q} = P(\hat{Y} \mid S = 1). \tag{12}$$

By substituting the distribution of predictions over the sensitive attribute into Equation (20), where $p = \mathbb{P}$ and $q = \mathbb{Q}$, we can define the objective we aim to solve as follows:

$$\min_{\theta} \mathcal{L}_{\text{BCE}} + \alpha \tilde{D}_{\text{CS}}\left(\mathbb{P}, \mathbb{Q}\right) + \frac{\beta}{2} \|\theta\|_2^2, \tag{13}$$

where $\mathcal{L}_{\text{BCE}}$ is the binary cross-entropy loss, which measures the classifier's accuracy. It is defined as:

$$\mathcal{L}_{\text{BCE}} = \frac{1}{M} \sum_{i=1}^{M} -Y_i \log \hat{Y}_i, \tag{14}$$

where $\hat{Y}_i$ is the predicted output obtained from the training model parameterized by $\theta$. This model can be a Multi-Layer Perceptron for tabular data or a ResNet for image data. Additionally, $\|\theta\|_2^2$ serves as an $L_2$ regularizer.

**Computational complexity.** Naively computing the CS divergence on all $n$ training examples has $O(n^2)$ cost, since it involves pairwise interactions between samples. In practice, as is standard for kernel-based regularizers such as `MMD` and `HSIC`, we evaluate the CS-based fairness loss on *mini-batches*. Given a batch of size $B$ (typically $B \ll n$), the additional cost per optimization step is $O(B^2)$ and is implemented using vectorized matrix operations that reuse the same mini-batches as the prediction loss. Under typical batch sizes used in our experiments, this overhead is modest and comparable to that of existing kernel-based fairness methods, making the `CS` regularizer computationally practical for both tabular and image models.

## 4.2 WHY IS THE CS DIVERGENCE MORE EFFECTIVE FOR ENSURING FAIRNESS?

The CS Divergence is particularly well-suited for promoting fairness due to several key reasons:

**(1) Closed-form solution for the mixture of Gaussians.** The CS divergence has several advantageous properties, one of which is that it provides a *closed-form solution for the mixture of Gaussians* (Kampa et al., 2011). This particular property has facilitated its successful application in various tasks, including deep clustering (Trosten et al., 2021), disentangled representation learning (Tran et al., 2022), and point-set registration (Sanchez Giraldo et al., 2017).

**(2) CS Divergence has a tighter error bound than the KL divergence.**

**Proposition 4.2.** *For any $d$-variate Gaussian distributions $p \sim \mathcal{N}(\boldsymbol{\mu}_p, \Sigma_p)$ and $q \sim \mathcal{N}(\boldsymbol{\mu}_q, \Sigma_q)$, where $\Sigma_p$ and $\Sigma_q$ are positive definite, the following inequality holds:*

$$D_{\text{CS}}(p; q) \leq D_{\text{KL}}(p; q) \ \text{ and } \ D_{\text{CS}}(p; q) \leq D_{\text{KL}}(q; p). \tag{15}$$

The proof can be found in Appendix B.3. *It is important to note that the divergences are being compared under the same model parameter $\theta$.*

**Remark on the Gaussian setting.** Proposition 4.2 should be interpreted as a stylized comparison that is carried out under a Gaussian model solely for analytical convenience. The Gaussian assumption allows us to derive closed-form expressions for both the CS and KL divergences and to make their relationship explicit. It is *not* required by our training procedure, and it is not assumed anywhere in the empirical evaluation; in particular, the CS-based regularizer we optimize is defined for arbitrary distributions of predictions and sensitive attributes.

**(3) CS divergence can provide tighter bounds than MMD and DP when the distributions are far apart or when the scale of the embeddings varies significantly.** Based on Remark A.1, we know that CS divergence employs cosine distance, while MMD relies on Euclidean distance. In addition, DP Equation (5) utilizes a mean disparity, which is a Manhattan distance for the mean estimations of two distributions. CS divergence measures the angle between two distributions in the feature space, focusing on the difference in direction rather than magnitude. In cases where the distributions have significantly different variances or scales, MMD and DP may yield a large distance even if the distributions are aligned in the feature space. In contrast, CS divergence normalizes this comparison, resulting in a more accurate measure of similarity and thereby providing a tighter generalization bound. This normalization enhances the robustness of CS divergence, preventing MMD and DP from overestimating the discrepancy due to their reliance on an unnormalized distance measure.

| | Methods | ACC (%) | ↑ | AUC (%) | ↑ | $\Delta_{DP}$ (%) | ↓ | $\Delta_{EO}$ (%) | ↓ |
|---|---|---|---|---|---|---|---|---|---|
| | | | **Utility** | | | | **Fairness** | | |
| **Adult** — Gender | MLP | $85.63_{\pm0.34}$ | — | $90.82_{\pm0.23}$ | — | $16.52_{\pm0.91}$ | — | $8.43_{\pm3.20}$ | — |
| | DP | $82.42_{\pm0.39}$ | -3.75% | $86.91_{\pm0.80}$ | -4.31% | $\underline{1.29}_{\pm0.95}$ | 92.19% | $20.15_{\pm1.13}$ | -139.03% |
| | MMD | $81.90_{\pm0.68}$ | -4.36% | $85.27_{\pm0.52}$ | -6.11% | $2.47_{\pm0.52}$ | 85.05% | $17.53_{\pm1.36}$ | -107.95% |
| | HSIC | $\underline{82.89}_{\pm0.23}$ | -3.20% | $\underline{87.25}_{\pm0.41}$ | -3.93% | $2.66_{\pm0.54}$ | 83.90% | $18.47_{\pm1.22}$ | -119.10% |
| | PR | $81.81_{\pm0.52}$ | -4.46% | $85.38_{\pm0.82}$ | -5.99% | $\mathbf{0.71}_{\pm0.40}$ | 95.70% | $\underline{12.45}_{\pm2.38}$ | -47.69% |
| | **CS** | $\mathbf{83.31}_{\pm0.47}$ | -2.71% | $\mathbf{90.15}_{\pm0.49}$ | -0.74% | $2.42_{\pm0.85}$ | 85.35% | $\mathbf{2.27}_{\pm1.04}$ | 73.07% |
| **Adult** — Race | MLP | $84.42_{\pm0.31}$ | — | $90.15_{\pm0.36}$ | — | $13.47_{\pm0.83}$ | — | $9.25_{\pm3.86}$ | — |
| | DP | $\underline{83.64}_{\pm0.78}$ | -0.92% | $88.45_{\pm0.32}$ | -1.89% | $2.45_{\pm0.67}$ | 81.81% | $2.16_{\pm1.06}$ | 76.65% |
| | MMD | $83.12_{\pm0.82}$ | -1.54% | $88.36_{\pm0.67}$ | -1.99% | $2.58_{\pm0.75}$ | 80.85% | $3.33_{\pm0.93}$ | 64.00% |
| | HSIC | $\mathbf{84.98}_{\pm0.17}$ | 0.66% | $\mathbf{90.90}_{\pm0.19}$ | 0.83% | $7.90_{\pm0.72}$ | 41.35% | $2.11_{\pm0.18}$ | 77.19% |
| | PR | $82.13_{\pm1.16}$ | -2.71% | $87.44_{\pm0.33}$ | -3.01% | $\mathbf{1.53}_{\pm0.83}$ | 88.64% | $\underline{0.86}_{\pm0.60}$ | 90.70% |
| | **CS** | $83.53_{\pm0.53}$ | -1.05% | $\underline{90.26}_{\pm0.47}$ | 0.12% | $\underline{2.16}_{\pm0.61}$ | 83.96% | $\mathbf{0.44}_{\pm0.12}$ | 95.24% |
| **COMPAS** — Gender | MLP | $66.85_{\pm0.72}$ | — | $72.10_{\pm0.94}$ | — | $13.22_{\pm3.32}$ | — | $11.41_{\pm5.83}$ | — |
| | DP | $64.20_{\pm1.58}$ | -3.96% | $70.64_{\pm1.05}$ | -2.02% | $5.78_{\pm0.33}$ | 56.28% | $6.78_{\pm1.61}$ | 40.58% |
| | MMD | $\underline{64.82}_{\pm1.62}$ | -3.04% | $70.72_{\pm0.92}$ | -1.91% | $3.09_{\pm0.92}$ | 76.63% | $3.15_{\pm4.37}$ | 72.39% |
| | HSIC | $63.17_{\pm3.46}$ | -5.50% | $71.17_{\pm0.84}$ | -1.29% | $\underline{1.84}_{\pm0.43}$ | 86.08% | $\underline{2.60}_{\pm0.63}$ | 77.21% |
| | PR | $\mathbf{64.95}_{\pm0.15}$ | -2.84% | $\mathbf{72.12}_{\pm0.75}$ | 0.03% | $3.85_{\pm0.60}$ | 70.88% | $3.91_{\pm1.02}$ | 65.73% |
| | **CS** | $64.25_{\pm0.97}$ | -3.89% | $\underline{71.53}_{\pm0.61}$ | -0.79% | $\mathbf{1.30}_{\pm0.47}$ | 90.17% | $\mathbf{0.44}_{\pm0.13}$ | 96.14% |
| **COMPAS** — Race | MLP | $66.99_{\pm1.05}$ | — | $72.46_{\pm0.88}$ | — | $17.24_{\pm4.15}$ | — | $19.44_{\pm4.63}$ | — |
| | DP | $64.98_{\pm3.72}$ | -3.00% | $72.09_{\pm1.03}$ | 0.51% | $8.70_{\pm1.12}$ | 49.54% | $7.04_{\pm2.13}$ | 63.79% |
| | MMD | $64.41_{\pm2.04}$ | -3.85% | $72.10_{\pm1.83}$ | 0.50% | $4.42_{\pm2.11}$ | 74.36% | $5.60_{\pm1.25}$ | 71.19% |
| | HSIC | $64.52_{\pm2.20}$ | -3.69% | $72.16_{\pm0.94}$ | 0.41% | $\underline{2.21}_{\pm0.68}$ | 87.18% | $\underline{2.72}_{\pm0.87}$ | 86.01% |
| | PR | $\mathbf{67.22}_{\pm0.90}$ | 0.34% | $\mathbf{72.86}_{\pm0.87}$ | -0.55% | $5.60_{\pm1.12}$ | 67.52% | $6.52_{\pm1.30}$ | 66.46% |
| | **CS** | $\underline{65.62}_{\pm1.24}$ | -2.05% | $\underline{72.70}_{\pm1.06}$ | 0.33% | $\mathbf{1.79}_{\pm0.96}$ | 89.62% | $\mathbf{1.48}_{\pm1.64}$ | 92.39% |
| **ACS-I** — Gender | MLP | $82.04_{\pm0.27}$ | — | $90.16_{\pm0.18}$ | — | $10.26_{\pm4.68}$ | — | $2.13_{\pm3.64}$ | — |
| | DP | $81.32_{\pm0.17}$ | -0.88% | $\underline{89.33}_{\pm0.15}$ | -0.92% | $0.96_{\pm0.22}$ | 90.64% | $5.37_{\pm0.32}$ | -152.11% |
| | MMD | $80.93_{\pm0.55}$ | -1.35% | $88.44_{\pm1.71}$ | -1.91% | $2.45_{\pm0.65}$ | 76.12% | $4.91_{\pm1.48}$ | -130.52% |
| | HSIC | $\underline{81.40}_{\pm0.12}$ | -0.78% | $\mathbf{89.53}_{\pm0.10}$ | -0.70% | $1.54_{\pm0.18}$ | 84.99% | $4.95_{\pm0.39}$ | -132.39% |
| | PR | $80.03_{\pm0.30}$ | -2.45% | $88.10_{\pm0.26}$ | -2.28% | $\mathbf{0.35}_{\pm0.20}$ | 96.59% | $\underline{4.54}_{\pm0.41}$ | -113.15% |
| | **CS** | $\mathbf{81.86}_{\pm0.94}$ | -0.22% | $89.15_{\pm0.60}$ | -1.12% | $\underline{0.77}_{\pm0.38}$ | 92.5% | $\mathbf{0.90}_{\pm0.46}$ | 57.75% |
| **ACS-I** — Race | MLP | $81.23_{\pm0.14}$ | — | $90.16_{\pm0.18}$ | — | $10.06_{\pm1.84}$ | — | $7.42_{\pm0.66}$ | — |
| | DP | $\underline{81.25}_{\pm0.13}$ | 0.02% | $\underline{89.45}_{\pm0.11}$ | -0.79% | $\underline{0.56}_{\pm0.30}$ | 94.43% | $4.53_{\pm0.48}$ | 38.95% |
| | MMD | $80.22_{\pm1.22}$ | -1.24% | $88.42_{\pm1.63}$ | -1.93% | $1.45_{\pm0.89}$ | 85.59% | $4.01_{\pm0.54}$ | 45.96% |
| | HSIC | $\mathbf{81.41}_{\pm0.15}$ | 0.22% | $\mathbf{89.67}_{\pm0.12}$ | -0.54% | $1.04_{\pm0.53}$ | 89.66% | $\underline{2.77}_{\pm0.35}$ | 62.67% |
| | PR | $80.27_{\pm0.26}$ | -1.18% | $88.45_{\pm0.21}$ | -1.90% | $\mathbf{0.37}_{\pm0.30}$ | 96.32% | $4.25_{\pm0.49}$ | 42.72% |
| | **CS** | $80.78_{\pm0.85}$ | -0.55% | $89.14_{\pm0.94}$ | -1.13% | $0.81_{\pm0.28}$ | 91.95% | $\mathbf{1.35}_{\pm0.64}$ | 81.81% |

Table 2: Fairness performance of existing fair models on the tabular datasets, considering race and gender as sensitive attributes. ↑ indicates accuracy improvement **compared to MLP**, with higher accuracy reflecting better performance, and ↓ denotes fairness improvement **compared to MLP**, where lower values indicate better fairness. Green values denote better than MLP on the corresponding metric (ACC/AUC ↑; $\Delta_{DP}/\Delta_{EO}$↓), while red denote worse. All results are based on 10 runs for each method. The best results for each metric and dataset are highlighted in **bold** text.

**Distribution-free applicability.** We also emphasize that the optimization framework of Section 4 is distribution-free. The empirical CS divergence used as a fairness loss is implemented via a kernel-based estimator that only depends on samples from the joint distribution of predictions and sensitive attributes and does not impose any parametric form on this distribution. Consequently, the same regularizer can be applied to both tabular and image models. Our experiments in Section 5, which span four tabular datasets and one image dataset with clearly non-Gaussian distributions, show that the CS-based regularizer consistently improves group-fairness metrics while maintaining competitive utility, supporting the practical relevance of the theoretical insight obtained from the Gaussian comparison in Proposition 4.2.

**Choice of kernel.** Throughout this work, we instantiate the empirical CS divergence with a Gaussian (RBF) kernel and choose its bandwidth using the median heuristic, following common practice in kernel-based dependence measures such as MMD and HSIC (Gretton et al., 2012; 2005). This choice is kept fixed across datasets and models to ensure a controlled comparison between regularizers. In principle, our framework is not restricted to a particular kernel family: alternative kernels (e.g.,

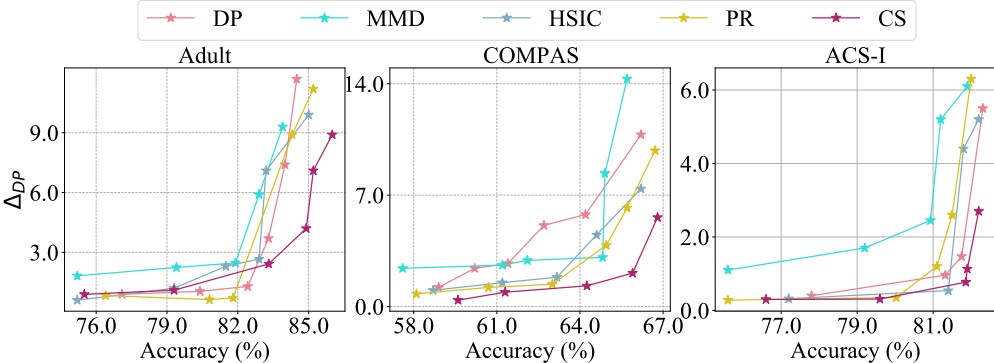

Figure 3: Fairness-accuracy trade-off curves on the test sets for (left) `Adult`, (middle) `COMPAS`, and (bottom) `ACS-I`. Ideally, results should be positioned in the bottom-right corner.

Laplacian or polynomial) can be plugged into the same estimator without changing the overall training objective or algorithm (Schölkopf & Smola, 2002; Shawe-Taylor & Cristianini, 2004). A more systematic investigation of how different kernel families and bandwidth-selection strategies affect the utility-fairness trade-off is an interesting direction for future work.

# 5 EXPERIMENTS

In this section, we evaluate the effectiveness of the `CS` fairness regularizer from several perspectives: **(1)** utility and fairness performance, **(2)** the tradeoff between utility and fairness, **(3)** prediction distributions across different sensitive groups, **(4)** T-SNE plots for these sensitive groups, and **(5)** the sensitivity of parameters in Equation (13). Our evaluation encompasses five datasets with diverse sensitive attributes, including four tabular datasets: `Adult`, `COMPAS`, `ACS-I`, and `ACS-T`, as well as one image dataset, `CelebA-A`. Utility performance is assessed based on accuracy and the area under the curve (AUC), while fairness performance is measured using $\triangle_{DP}$ Equation (2) and $\triangle_{EO}$ Equation (3). Detailed information about the datasets and baselines can be found in the Appendices E and F. We denote an observation drawn from the results as **Obs.**.

## 5.1 FAIRNESS AND UTILITY PERFORMANCE

We conducted experiments on five datasets along with their corresponding baselines, as previously mentioned. For each dataset, we performed 10 different splits to ensure robustness in our results. We calculated the mean and standard deviation for each metric across these splits. The accuracy and fairness performance of the downstream tasks is in Table 2. Our observations are as follows:

**Obs. 1: `CS` achieves the best or near-best $\triangle_{EO}$ on most datasets and is competitive on $\triangle_{DP}$, while maintaining high utility.** Notably, `CS` demonstrates exceptional fairness performance on the image dataset, `CelebA-A`, where the disparity in the 'Young' and 'Non-Young' groups sees a $\triangle_{DP}$ reduction of 97.36% and a $\triangle_{EO}$ reduction of 98.58%. Furthermore, in the `Adult` and `ACS-I` datasets, which include gender groups, traditional methods such as `DP`, `MMD`, `HSIC`, and `PR` do not effectively optimize for EO fairness. In contrast, the proposed `CS` achieves significant reductions in $\triangle_{EO}$ by 72.12% and 63.85%, respectively, compared to MLP.

**Obs. 2: `CS` achieves good fairness performance with a small sacrifice in utility.** Specifically, `CS` exhibits a decrease of less than 3.1% in accuracy and less than 2.2% in AUC. The only exception is observed with `COMPAS` when gender is treated as a sensitive attribute, resulting in a slightly higher accuracy loss of 3.6%. Notably, `CS` demonstrates either equivalent or improved AUC performance, with increases of 0.02% and 0.58% on `Adult` for the gender and race groups, respectively, as well as a 0.35% increase on `COMPAS` for the race group. Among the baselines, `HSIC` ranks highest in utility, achieving the best performance on `ACS-I` for the race group and on `ACS-T` for both the gender and race groups. This is followed by `PR`, which shows the best utility on `COMPAS` for both the gender and race groups, as well as on `CelebA-A` for the gender group.

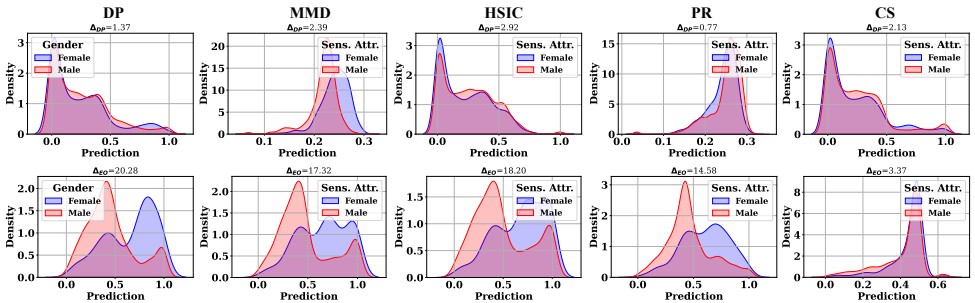

Figure 4: Prediction distributions for female and male groups in the `Adult` dataset. The top row shows kernel density estimates of the raw predictions $\hat{Y}$ for all target labels, grouped by gender, while the bottom row shows the prediction densities for the positive class, $\hat{Y} = 1$, for the two gender groups. Each column corresponds to a different fairness regularizer. A larger overlap between the blue and red curves indicates better group fairness, and the reported values above each panel give the corresponding gaps in $\Delta_{\text{DP}}$ (top row) and $\Delta_{\text{EO}}$ (bottom row).

## 5.2 HOW DO ACCURACY AND FAIRNESS TRADE-OFF IN BASELINE MODELS AND CS?

We evaluate the trade-off between accuracy and $\triangle_{DP}$ for the baselines by varying the fairness hyperparameters (Yao et al., 2023; Deka & Sutherland, 2023). The results are presented in Figure 3, where the x-axis represents the target accuracy, while the y-axis shows the average Demographic Parity (DP) across both positive and negative target classes. It is important to note that the figure in the bottom right corner represents the optimal result.

**Obs. 3: At the same utility level, `CS` is the most effective method in promoting fairness.** Analyzing the results, we find that CS consistently achieves the lowest $\triangle_{DP}$ across most accuracy levels, with this effect becoming more pronounced at higher accuracy levels. This is evidenced by the significant gap in $\triangle_{DP}$ between CS and other baselines. It is important to note that while all baselines can demonstrate good fairness when the optimization prioritizes fairness over task objectives, the task objective remains critical for the practical application of these models. **Obs. 4: High accuracy can sometimes lead to worse fairness compared to MLP, as the fairness objective becomes more challenging to optimize when there is a stronger focus on task-specific objectives.** As shown in Table 2, the $\triangle_{DP}$ for MMD is over 14.0, which is greater than the average $\triangle_{DP}$ of 13.22 for MLP. However, these fairness regularizers generally prove effective in controlling bias in representations, especially when more emphasis is placed on the task-specific objective. Notably, some datasets with particularly sensitive attributes pose greater challenges for achieving fairness. For instance, the COMPAS dataset, which includes gender as a sensitive attribute, demonstrates this difficulty. One possible explanation is the relatively small sample size of COMPAS, which contains only $6,172$ samples, significantly fewer than other datasets where fairness is easier to achieve. **Obs. 5: `CS` displays a significant increase in $\triangle_{DP}$ at a slower rate than other baselines as accuracy increases.** We analyze the slope of the lines representing the increase in $\triangle_{DP}$ with rising accuracy. Many methods, such as PR and DP, demonstrate strong fairness performance at low accuracy levels; however, they quickly lose control over fairness as accuracy begins to increase. This is evident from the abrupt rise in $\triangle_{DP}$ observed at around $82.0\%$ on `Adult`, $63.0\%$ on COMPAS, and $81.0\%$ on ACS-I. In contrast, CS only exhibits a sudden increase at $85.0\%$, $65.5\%$, and $81.5\%$ for the same datasets, respectively.

## 5.3 HOW CAN THE CS FAIRNESS REGULARIZER PERFORM WELL ON BOTH DP AND EO?

We visualize the kernel density estimate plot [2] of the predictions $\hat{Y}$ across different sensitive groups to analyze how CS achieves a better balance of various fairness definitions compared to other baselines. The *first row* displays the predictions for all target classes, specifically $Y = 0$ and $Y = 1$, grouped by sensitive attributes. In this row, the blue areas represent the prediction density for $S = 0$, while the red areas indicate the prediction density for $S = 1$. The *second row* illustrates the prediction

---

[2]https://seaborn.pydata.org/generated/seaborn.kdeplot.html

density for the positive target class, $Y = 1$, across two different sensitive groups. Figure 4 presents the results for Adult based on gender and race groups, with additional results for other datasets available in Appendix D.2.

**Obs. 6: CS effectively optimizes the prediction distributions for the two sensitive groups, specifically $\hat{Y}|S = 0$ and $\hat{Y}|S = 1$. Additionally, it optimizes the prediction distributions for these groups within the positive target group, i.e., $\hat{Y}|S = 0, Y = 1$ and $\hat{Y}|S = 1, Y = 1$.** Achieving DP and EO fairness requires different objectives. For instance, DP directly optimizes the $\triangle_{DP}$, which results in reduced effectiveness for achieving EO fairness. This is evident across all datasets, as DP ranks among the worst, achieving $7/10$ of the lowest EO fairness scores on $\triangle_{EO}$ when tested on five datasets with two types of sensitive attributes. The distribution plots for DP further illustrate this, showing a generally larger gap between the two sensitive groups in the EO plots compared to other methods. In contrast, CS consistently minimizes the prediction density gap between the two sensitive groups.

## 5.4 PARAMETER SENSITIVITY ANALYSIS

For all models, we tune the hyperparameters using cross-validation on the training set. The hyperparameters for these variants are determined through grid search during cross-validation. Specifically, we vary the parameters $\alpha$ and $\beta$ in Equation (13) across the ranges $(1e - 6, 150)$ and $(1e - 3, 10)$, respectively. In this experiment, we specifically visualize the values of $\alpha$ in the range $(1e - 4, 1e - 1)$ for CS.

The heatmap in Figure 5 illustrates the accuracy and $\triangle_{DP}$ across various combinations of $\alpha$ and $\beta$ values for the Adult. In the accuracy plots, darker colors indicate higher values, which are preferable, while lighter colors in the $\triangle_{DP}$ plots represent better fairness performance.

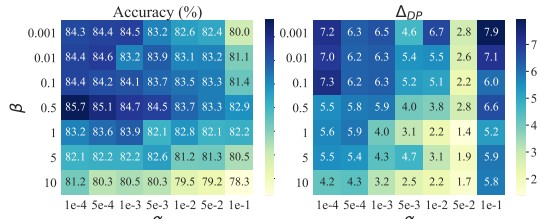

Figure 5: **Parameter sensitivity of the CS regularizer on ADULT**: heatmaps show test accuracy (left) and $\triangle_{DP}$ (right) as the fairness weight $\alpha$ and $\ell_2$ weight $\beta$ vary over the cross-validated ranges. Overall, CS exhibits a smooth utility–fairness trade-off, remaining stable over a broad range of $\beta$ and becoming noticeably more sensitive only when $\alpha$ is very large.

**Obs. 7: The highest accuracy is achieved when $\alpha$ is set to its smallest value, $1e - 4$, while the best fairness is obtained with $\alpha = 5e - 2$.** Notably, fairness drops significantly when $\alpha$ increases from $5e - 2$ to $1e - 1$. Generally, smaller values of $\alpha$ can still yield satisfactory fairness performance when paired with an appropriate range of $\beta$, specifically around $5 - 10$. **Obs. 8: The fairness performance is more sensitive to changes in $\alpha$ than in $\beta$.** For instance, adjusting $\beta$ from $1e - 3$ to $10$, which represents a $10,000\times$ increase, results in only a slight decrease in $\triangle_{DP}$ from $7.2$ to $4.2$. In contrast, increasing $\alpha$ from $1e - 2$ to $5e - 2$, a $5\times$ change, leads to a significant drop in $\triangle_{DP}$ from $6.7$ to $2.8$, when keeping $\beta$ fixed at $1e - 3$.

## 6 CONCLUSION

In this paper, we introduce a novel fair machine learning method called the Cauchy-Schwarz (CS) fairness regularizer. Empirically, our approach achieves a more consistent utility–fairness trade-off across hyperparameter settings than standard regularizers, and yields more generalizable fairness by minimizing the Cauchy–Schwarz divergence between the prediction distribution and the sensitive attributes. We demonstrate that the CS divergence provides a tighter bound compared to both the Kullback-Leibler divergence and the Maximum Mean Discrepancy, as well as the mean disparity used in Demographic Parity regularization. This superiority is particularly evident when the distributions are significantly different or when there is substantial variation in the scale of the embeddings. As a result, our CS fairness regularizer delivers improved fairness performance in practical scenarios. While our work currently only evaluates on general machine learning tasks, and thus leaves future work to other tasks such as graph learning.

## ACKNOWLEDGEMENTS

This work was supported in part by Nthe DARPA Young Faculty Award, the National Science Foundation (NSF) under Grants #2431561, #2127780, #2319198, #2321840, #2312517, and #2235472, the Semiconductor Research Corporation (SRC), the Office of Naval Research through the Young Investigator Program Award, Grants #N00014-21-1-2225 and #N00014-22-1-2067, Army Research Office Grant #W911NF2410360, and the National Defense Science & Engineering Graduate (ND-SEG) Fellowship Program. Additionally, support was provided by the Air Force Office of Scientific Research under Award #FA9550- 22-1-0253, along with generous gifts from Xilinx and Cisco.

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

## A  WHAT IS THE RELATIONSHIP BETWEEN CS DIVERGENCE AND EXISTING DISTRIBUTION DISTANCE MEASURES?

To illustrate the advantages of the CS fairness regularizer, we begin by summarizing the commonly used distribution distance metrics: Maximum Mean Discrepancy (MMD), Kullback-Leibler divergence (KL), and Hilbert-Schmidt Independence Criterion (HSIC).

**Demographic Parity Regularizer.** The demographic parity regularizer is widely utilized in fairness-focused machine learning studies (Chuang & Mroueh, 2020). It aims to optimize the mean disparity between two *prediction distributions*. This regularizer can be formally expressed as:

$$\text{DP}(p;q) = |\frac{1}{N_1} \sum_{i}^{N_1} p(\mathbf{x}_i) - \frac{1}{N_2} \sum_{j}^{N_2} q(\mathbf{x}_j)|, \tag{16}$$

where $\mathbf{x}_i$ and $\mathbf{x}_j$ are data points from $S = 0$ and $S = 1$, in the context of fairness. In the following, we represent $\mathbf{x}_i$ with distribution $p$ and $\mathbf{x}_j$ with distribution $q$ as $\mathbf{x}_i^p$ and $\mathbf{x}_i^q$ for simplicity. However, only optimizing on the mean disparity of two distributions cannot always generate an optimized DP or EO, as the Equation (16) equals 0 is a necessary but not sufficient condition for achieving DP and EO.

**Mean Maximum Discrepancy.** One of the most widely used distance metrics is the Mean Maximum Discrepancy (MMD) (Gretton et al., 2012). In the context of fairness, previous studies have employed MMD as a regularizer to enforce statistical parity among the *embeddings* of different sensitive groups within a machine learning model (Deka & Sutherland, 2023; Louizos et al., 2016). This approach aims to facilitate fair representation learning.

$$\widetilde{\text{MMD}}^2(p;q) = \frac{1}{N_1^2} \sum_{i,j=1}^{N_1} \kappa(\mathbf{x}_i^p, \mathbf{x}_j^p) + \frac{1}{N_2^2} \sum_{i,j=1}^{N_2} \kappa(\mathbf{x}_i^q, \mathbf{x}_j^q)$$
$$- \frac{2}{N_1 N_2} \sum_{i=1}^{N_1} \sum_{j=1}^{N_2} \kappa(\mathbf{x}_i^p, \mathbf{x}_j^q). \tag{17}$$

By comparing with Equation (20), we observe that the CS divergence introduces a logarithmic term for each component of the MMD. Through simple transformations, we can deduce the following:

*Remark* A.1. CS divergence measures the cosine distance between empirical mean embedding $\boldsymbol{\mu}_p = \frac{1}{N_1} \sum_{i=1}^{N_1} f(\mathbf{x}_i^p)$ and $\boldsymbol{\mu}_q = \frac{1}{N_2} \sum_{j=1}^{N_2} f(\mathbf{x}_j^q)$ in a Reproducing Kernel Hilbert Space, while MMD utilizes Euclidean distance.

**Kullback-Leibler Divergence.** Kullback-Leibler (KL) Divergence is a key concept in information bottleneck theory, where it is used to quantify the mutual information between two probability distributions. This metric has gained popularity across various domains, including fair machine learning (Kamishima et al., 2012).

$$D_{\text{KL}} = \int p(\mathbf{x}) \log \left( \frac{p(\mathbf{x})}{q(\mathbf{x})} \right) \tag{18}$$

**Hilbert-Schmidt Independence Criterion (HSIC).** Let $K$ and $L$ denote the Gram matrices for the variables $x$ and $y$, respectively. Specifically, $K$ is defined such that $K_{ij} = \kappa(\mathbf{x}_i, \mathbf{x}_j)$, and $L$ is defined as $L_{ij} = \kappa(\mathbf{y}_i, \mathbf{y}_j)$, where $\kappa$ is the Gaussian kernel function given by $\kappa = \exp\left(-\frac{\|\cdot\|^2}{2\sigma^2}\right)$. The Hilbert-Schmidt Independence Criterion (HSIC) can be estimated using the following expression (Gretton et al., 2007):

$$\widetilde{\text{HSIC}}(p;q) = \frac{1}{N^2} \sum_{i,j}^{N} K_{ij} L_{ij} + \frac{1}{N^4} \sum_{i,j,q,r}^{N} K_{ij} L_{qr}$$
$$- \frac{2}{N^3} \sum_{i,j,q}^{N} K_{ij} L_{iq} = \frac{1}{N^2} \operatorname{tr}(KHLH), \tag{19}$$

where $H = I - \frac{1}{N}\mathbb{1}\mathbb{1}^T$ represents a centering matrix of size $N \times N$. In this expression, $I$ is the identity matrix, $\mathbb{1}$ is a vector of ones, and $\frac{1}{N}\mathbb{1}\mathbb{1}^T$ computes the average across the columns, effectively centering the data by subtracting the mean from each entry.

Compared to Equation (17), The HSIC can be interpreted as the MMD between the joint distribution $p(\mathbf{x}, \mathbf{y})$ and the product of their marginal distributions $p(\mathbf{x})p(\mathbf{y})$.

# B    DETAILS ON THE RELATION OF CS AND EXISTING FAIRNESS REGULARIZERS

## B.1    PROOF OF PROPOSITION 4.1

**Proposition 1.** *Given two sets of observations $\{\mathbf{x}_i^p\}_{i=1}^{N_1}$ and $\{\mathbf{x}q_j\}_{j=1}^{N_2}$, let $p$ and $q$ denote the distributions of two groups. The empirical estimator of the CS divergence $D_{CS}(p;q)$ is given by:*

$$\tilde{D}_{CS}(p;q) = \log\left(\frac{1}{N_1^2}\sum_{i,j=1}^{N_1}\kappa(\mathbf{x}_i^p, \mathbf{x}_j^p)\right) + \log\left(\frac{1}{N_2^2}\sum_{i,j=1}^{N_2}\kappa(\mathbf{x}_i^q, \mathbf{x}_j^q)\right)$$
$$- 2\log\left(\frac{1}{N_1 N_2}\sum_{i=1}^{N_1}\sum_{j=1}^{N_2}\kappa(\mathbf{x}_i^p, \mathbf{x}_j^q)\right). \tag{20}$$

*Proof.* The CS divergence is defined as:

$$D_{\text{CS}}(p;q) = -\log\left(\frac{\left(\int p(\mathbf{x})q(\mathbf{x})\,\mathrm{d}\mathbf{x}\right)^2}{\int p(\mathbf{x})^2\,\mathrm{d}\mathbf{x}\int q(\mathbf{x})^2\,\mathrm{d}\mathbf{x}}\right), \tag{21}$$

where $\hat{p}(\mathbf{x}) = \frac{1}{M}\sum_{i=1}^{M}\kappa_\sigma(\mathbf{x} - \mathbf{x}_j^p)$ and $\hat{q}(\mathbf{x}) = \frac{1}{N}\sum_{i=1}^{N}\kappa_\sigma(\mathbf{x} - \mathbf{x}_j^q)$ are kernel density estimation. Then we can obtain:

$$\int \hat{p}^2(\mathbf{x})\,\mathrm{d}\mathbf{x} = \frac{1}{M^2}\sum_{i=1}^{M}\sum_{j=1}^{M}\kappa_{\sqrt{2}\sigma}(\mathbf{x}_i^p - \mathbf{x}_j^p). \tag{22}$$

By a similar approach,

$$\int \hat{q}(\mathbf{z})^2\,\mathrm{d}\mathbf{x} = \frac{1}{N^2}\sum_{i=1}^{N}\sum_{j=1}^{N}\kappa_{\sqrt{2}\sigma}(\mathbf{x}_i^q - \mathbf{x}_j^q), \tag{23}$$

and

$$\int \hat{p}(\mathbf{x})\hat{q}(\mathbf{x})\,\mathrm{d}\mathbf{x} = \frac{1}{MN}\sum_{i=1}^{M}\sum_{j=1}^{N}\kappa_{\sqrt{2}\sigma}(\mathbf{x}_i^q - \mathbf{x}_j^p). \tag{24}$$

Substituting (Equation (22))-(Equation (24)) into Eq. (21), we obtain:

$$\widetilde{D}_{\text{CS}}(p;q) = \log\left(\frac{1}{M^2}\sum_{i,j=1}^{M}\kappa_{\sqrt{2}\sigma}(\mathbf{x}_i^p - \mathbf{x}_j^p)\right) + \log\left(\frac{1}{N^2}\sum_{i,j=1}^{N}\kappa_{\sqrt{2}\sigma}(\mathbf{x}_i^q - \mathbf{x}_j^q)\right)$$
$$- 2\log\left(\frac{1}{MN}\sum_{i=1}^{M}\sum_{j=1}^{N}\kappa_{\sqrt{2}\sigma}(\mathbf{x}_i^q - \mathbf{x}_j^p)\right). \tag{25}$$

$\square$

## B.2    PROOF OF REMARK A.1

**Remark 1.** *CS divergence measures the cosine distance between $\boldsymbol{\mu}_p$ and $\boldsymbol{\mu}_q$ in a Reproducing Kernel Hilbert Space, while MMD utilizes Euclidean distance.*

*Proof.* Let $\mathcal{H}$ be a Reproducing Kernel Hilbert Space (RKHS) associated with a kernel $\kappa(\mathbf{x}_i^p, \mathbf{x}_j^q) = \langle f(\mathbf{x}_i^p), f(\mathbf{x}_j^q) \rangle_{\mathcal{H}}$ (Yu et al., 2024). The mean embeddings of two distributions $p$ and $q$ in $\mathcal{H}$ are denoted by $\boldsymbol{\mu}_p = \frac{1}{N_1} \sum_{i=1}^{N_1} f(\mathbf{x}_i^p)$ and $\boldsymbol{\mu}_q = \frac{1}{N_2} \sum_{j=1}^{N_2} f(\mathbf{x}_j^q)$ in $\mathcal{H}$, respectively. The CS divergence defined by Equation (20) can thus be written as:

$$\widetilde{D}_{\text{CS}}(p; q) = -2 \log \frac{\langle \boldsymbol{\mu}_p, \boldsymbol{\mu}_q \rangle_{\mathcal{H}}}{\|\boldsymbol{\mu}_p\|_{\mathcal{H}} \|\boldsymbol{\mu}_q\|_{\mathcal{H}}} = -2 \log D_{\text{COS}}(\boldsymbol{\mu}_p, \boldsymbol{\mu}_q)$$

Here, $\langle \cdot, \cdot \rangle_{\mathcal{H}}$ denotes the inner product in the RKHS, and $\| \cdot \|_{\mathcal{H}}$ represents the norm induced by the inner product. The mean embeddings $\boldsymbol{\mu}_p$ and $\boldsymbol{\mu}_q$ are elements of $\mathcal{H}$. Thus, the CS divergence is computed based on the cosine distance $D_{\text{COS}}$ between $\boldsymbol{\mu}_p$ and $\boldsymbol{\mu}_q$.

Similarly, the Maximum Mean Discrepancy (MMD) between distributions $p$ and $q$ defined in Equation (17) can be written as:

$$\text{MMD}^2(p, q) = \|\boldsymbol{\mu}_p - \boldsymbol{\mu}_q\|_{\mathcal{H}}^2 = D_{\text{EUC}}(\boldsymbol{\mu}_p, \boldsymbol{\mu}_q).$$

Thus, the MMD measures the Euclidean distance between the mean embeddings of $p$ and $q$ in the RKHS $\mathcal{H}$, i.e., the $\boldsymbol{\mu}_p$ and $\boldsymbol{\mu}_q$. $\qquad\square$

### B.3 Proof of Proposition 4.2

**Proposition 2.** *For any d-variate Gaussian distributions $p \sim \mathcal{N}(\boldsymbol{\mu}_p, \Sigma_p)$ and $q \sim \mathcal{N}(\boldsymbol{\mu}_q, \Sigma_q)$ with positive definite $\Sigma_p$ and $\Sigma_q$, the following inequality holds:*

$$D_{\text{CS}}(p; q) \leq D_{\text{KL}}(p; q) \quad \text{and} \quad D_{\text{CS}}(p; q) \leq D_{\text{KL}}(q; p). \tag{26}$$

*Proof.* The KL divergence for $p$ and $q$ is given by:

$$D_{\text{KL}}(p; q) = \frac{1}{2} \left( \text{tr}(\Sigma_q^{-1} \Sigma_p) - d + (\boldsymbol{\mu}_q - \boldsymbol{\mu}_p)^{\top} \Sigma_q^{-1} (\boldsymbol{\mu}_q - \boldsymbol{\mu}_p) + \log \left( \frac{|\Sigma_q|}{|\Sigma_p|} \right) \right). \tag{27}$$

The CS divergence is expressed as (Kampa et al., 2011):

$$D_{\text{CS}}(p; q) = -\log(d_{xy}) + \frac{1}{2} \log(d_{xx}) + \frac{1}{2} \log(d_{yy}), \tag{28}$$

where: $\tag{29}$

$$d_{pq} = \frac{\exp \left( -\frac{1}{2} (\boldsymbol{\mu}_p - \boldsymbol{\mu}_q)^{\top} (\Sigma_p + \Sigma_q)^{-1} (\boldsymbol{\mu}_p - \boldsymbol{\mu}_q) \right)}{\sqrt{(2\pi)^d |\Sigma_p + \Sigma_q|}}, \tag{30}$$

$$d_{pp} = \frac{1}{\sqrt{(2\pi)^d |2\Sigma_p|}}, \quad d_{qq} = \frac{1}{\sqrt{(2\pi)^d |2\Sigma_q|}}. \tag{31}$$

We simplify:

$$D_{\text{CS}}(p; q) = \frac{1}{2} (\boldsymbol{\mu}_q - \boldsymbol{\mu}_p)^{\top} (\Sigma_p + \Sigma_q)^{-1} (\boldsymbol{\mu}_q - \boldsymbol{\mu}_p) + \frac{1}{2} \log \left( \frac{|\Sigma_p + \Sigma_q|}{2^d \sqrt{|\Sigma_p||\Sigma_q|}} \right). \tag{32}$$

When the mean vectors differ, based on the property (Horn & Johnson, 2012), $\Sigma_q^{-1} - (\Sigma_p + \Sigma_q)^{-1}$ is positive semi-definite given $\Sigma_p = \Sigma_q$, we have:

$$\begin{aligned}
&2(D_{\text{CS}}(p; q) - D_{\text{KL}}(p; q)) \\
&= (\boldsymbol{\mu}_q - \boldsymbol{\mu}_p)^{\top} (\Sigma_p + \Sigma_q)^{-1} (\boldsymbol{\mu}_q - \boldsymbol{\mu}_p) \\
&\quad - (\boldsymbol{\mu}_q - \boldsymbol{\mu}_p)^{\top} \Sigma_q^{-1} (\boldsymbol{\mu}_q - \boldsymbol{\mu}_p) \leq 0.
\end{aligned} \tag{33}$$

When the covariance matrices differ, let $I$ be the $d$-dimensional identity matrix (Yin et al., 2024):

$$
\begin{aligned}
2(D_{\text{CS}}(p;q) - D_{\text{KL}}(p;q)) &= \log\left(\frac{|\Sigma_p + \Sigma_q|}{2^d\sqrt{|\Sigma_p||\Sigma_q|}}\right) \\
&\quad - \log\left(\frac{|\Sigma_q|}{|\Sigma_p|}\right) - \text{tr}(\Sigma_q^{-1}\Sigma_p) + d \\
&= -d\log 2 + \log\left(|\Sigma_q^{-1}\Sigma_p + I|\right) \\
&\quad + \frac{1}{2}\log\left(|\Sigma_q^{-1}\Sigma_p|\right) - \text{tr}(\Sigma_q^{-1}\Sigma_p) + d.
\end{aligned}
\tag{34}
$$

We have $|\Sigma_q^{-1}\Sigma_p| \leq \left(\frac{1}{d}\text{tr}(\Sigma_q^{-1}\Sigma_p)\right)^d$, and $|\Sigma_q^{-1}\Sigma_p + I| \leq \left(1 + \frac{1}{d}\text{tr}(\Sigma_q^{-1}\Sigma_p)\right)^d$. Thus, based on Equation (34), we can obtain:

$$
\begin{aligned}
&2(D_{\text{CS}}(p;q) - D_{\text{KL}}(p;q)) \\
&\leq -d\log 2 + d\log\left(1 + \frac{1}{d}\text{tr}(\Sigma_q^{-1}\Sigma_p)\right) \\
&\quad + \frac{d}{2}\log\left(\frac{1}{d}\text{tr}(\Sigma_q^{-1}\Sigma_p)\right) - \text{tr}(\Sigma_q^{-1}\Sigma_p) + d.
\end{aligned}
\tag{35}
$$

The combined Equation (33) and Equation (35), we can obtain:

$$
2(D_{\text{CS}}(p;q) - D_{\text{KL}}(p;q)) \leq 0,
\tag{36}
$$

Similarly, we can obtain $2(D_{\text{CS}}(q;p) - D_{\text{KL}}(q;p)) \leq 0$. In conclusion, we conclude:

$$
D_{\text{CS}}(p;q) \leq D_{\text{KL}}(p;q) \quad \text{and} \quad D_{\text{CS}}(p;q) \leq D_{\text{KL}}(q;p).
\tag{37}
$$

$\square$

## C    RELATED WORK

In this section, we first review relevant prior studies, beginning with an overview of algorithmic fairness in machine learning. We then narrow our focus to regularization-based in-processing methods, which are central to our approach.

### C.1    ALGORITHMIC FAIRNESS IN MACHINE LEARNING

The importance of fairness in machine learning has grown significantly as the demand for unbiased decision-making models for individuals and groups increases (Wan et al., 2023; Le Quy et al., 2022; Liu et al., 2025d). This is especially critical in high-stakes applications where the consequences of biased decisions can be severe. Fairness is commonly categorized into three main types: *Individual fairness* (Yurochkin et al., 2019; Mukherjee et al., 2020; Yurochkin & Sun, 2020; Kang et al., 2020; Mukherjee et al., 2022), which aims to ensure that similar individuals are treated similarly; *Group fairness* (Hardt et al., 2016; Verma & Rubin, 2018; Li et al., 2020; Ling et al., 2023), which focuses on achieving fairness across predefined subgroups, often defined by sensitive attributes such as gender or race; *Counterfactual fairness* (Kusner et al., 2017; Agarwal et al., 2021; Zuo et al., 2022), which seeks to ensure fairness by considering how decisions would hold under alternative scenarios. Given the widespread adoption of group fairness metrics in real-world applications and the increasing development of in-processing techniques for deep neural network models (Liu, 2023; Liu & Shen, 2025; Liu et al., 2025e; 2024), we focus on benchmarking these methods to ensure group fairness in neural networks, particularly for tabular and image data.

Various techniques for mitigating bias in machine learning models can be categorized into three main approaches: *pre-processing*, *in-processing*, and *post-processing*. *Pre-processing* methods focus on addressing biases present in the dataset itself to ensure that the trained model exhibits fairness (Kamiran & Calders, 2012; Calmon et al., 2017a). For instance, these techniques may involve rebalancing the dataset or modifying the data collection process (Calmon et al., 2017b). *In-processing* methods, on the other hand, adjust the training objectives by incorporating fairness constraints directly into the learning process (Kamishima et al., 2012; Zhang et al., 2018; Madras

et al., 2018; Zhang et al., 2022; Buyl & De Bie, 2022; Alghamdi et al., 2022; Shui et al., 2022; Mehrotra & Vishnoi, 2022). This approach aims to ensure that the model learns fair representations during training. Finally, *post-processing* methods modify the predictions made by classifiers after the model has been trained, with the goal of promoting fairness across different groups (Hardt et al., 2016; Jiang et al., 2020; Tsaousis & Alghamdi, 2022). By categorizing these techniques, we can better understand the different strategies available for mitigating bias in machine learning systems.

## C.2 REGULARIZATION-BASED IN-PROCESSING METHODS

In this paper, we explore three types of regularization-based in-processing methods. First, *Gap Regularization* (Chuang & Mroueh, 2020) streamlines the optimization process by offering a smooth approximation of real-world loss functions, which are typically non-convex and difficult to optimize directly. This category includes methods such as DP, EO, and EOD. Second, the *Independence* approach integrates fairness constraints into the optimization, aiming to mitigate the influence of protected attributes on model predictions while maintaining overall performance. Notable examples of this approach include PR (Kamishima et al., 2012) and HSIC (Li et al., 2019). Lastly, *adversarial debiasing* seeks to minimize utility loss while hindering an adversary's ability to accurately predict the protected attributes. This approach encompasses methods like ADV (Zhang et al., 2018; Louppe et al., 2017; Beutel et al., 2017; Edwards & Storkey, 2015; Adel et al., 2019) and LAFTR (Madras et al., 2018).

Beyond the algorithmic-fairness literature, recent studies have highlighted several important developments in LLM research, including retrieval-augmented recurrent models for long-context processing, memory-controlled long-context agents, lifelong model editing (Wang et al., 2025; 2026a;b), efficient LLM unlearning (Liu et al., 2025a;b), and knowledge-intensive LLM reasoning (Liu et al., 2025c). While these works are not designed for fairness, they reflect the growing importance of controllability and stability in modern LLM systems. This broader trend suggests a promising direction for adapting fairness-aware regularization to more complex LLM settings.

## C.3 CAUCHY-SCHWARZ DIVERGENCE IN OTHER ML SETTINGS

Cauchy-Schwarz (CS) divergence has also been studied and applied in several machine-learning problems outside of algorithmic fairness. Early work used CS divergence together with Parzen window density estimates to build information-theoretic criteria for clustering and graph-based learning, and to relate CS divergence to Mercer kernels and graph cuts (Jenssen et al., 2006). More recent studies have employed CS divergence as a training objective for representation learning and deep models, for example, in information-bottleneck formulations for regression (Yu et al., 2024), domain adaptation (Yin et al., 2024), and CS-regularized autoencoders that improve density estimation and clustering performance (Tran et al., 2022). Conditional variants of CS divergence have further been developed for time-series analysis and sequential decision making (Yu et al., 2025). These works demonstrate that CS divergence is a versatile discrepancy measure for density estimation and representation learning; our contribution is complementary, as we systematically develop and evaluate CS divergence as an *in-processing fairness regularizer* with dedicated theoretical analysis and extensive experiments in the algorithmic-fairness setting.

In contrast to these applications, which primarily target density estimation, clustering, or representation learning objectives, our focus is on *algorithmic fairness*. To the best of our knowledge, our work is the first to systematically develop Cauchy-Schwarz divergence as an *in-processing fairness regularizer*, with theoretical analysis tailored to group-fairness notions and an extensive empirical study of the resulting utility-fairness trade-offs.

## C.4 IS THE REPRESENTATION LEARNED BY APPLYING CS VIEWED AS FAIR?

To further validate that CS can learn fair representations, we visualize the T-SNE embeddings of the latent space from the last layer before the prediction layer (Van der Maaten & Hinton, 2008)[3]. Figure 6 displays the representations learned from the last embedding layer on the Adult, COMPAS,

---

[3]https://scikit-learn.org/stable/modules/generated/sklearn\protect\penalty\z@.manifold.TSNE.html

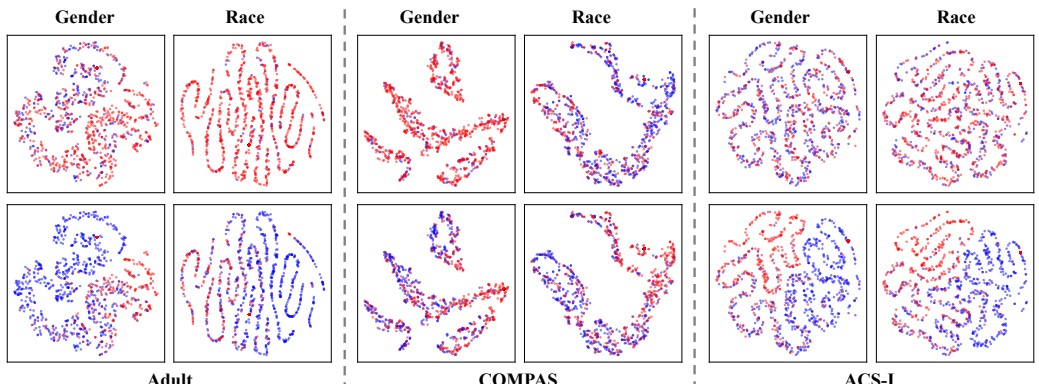

Figure 6: T-SNE visualizations of the latent representations on `Adult`, `COMPAS`, and `ACS-I`, colored by the target attribute (top) and the sensitive attribute (bottom).

and `ACS-I` datasets, while Figure 12 presents the results for `ACS-T` and `CelebA-A`. Based on these visualizations, we make the following observations:

**The `CS` can learn representations that are indistinguishable between sensitive groups.** This observation validates the effectiveness of `CS` in learning fair representations. Specifically, the plots in the first row of Figure 6 illustrate the embedding visualization of two sensitive groups: blue for $S=0$ and red for $S=1$. Overall, the points are uniformly dispersed, with no clear clusters of nodes sharing the same color. This indicates that the embeddings are learned independently of the sensitive attribute. Although some groups have a greater number of data points—such as in the `Adult` dataset with the sensitive attribute *race*, where the ratio of $S=0:S=1$ is $1:9.20$, and in the `COMPAS` dataset with *gender*, where the ratio is $1:4.17$ (as shown in Table 4)—the distribution of points in both colors remains even.

**The `CS` can learn distinguishable representations for different target attributes.** Observing the second row of Figure 6, we can identify a distinct pattern in the distribution of the blue and red points across different locations in the plot. Among these, the embedding for `ACS-I` exhibits the clearest pattern, followed by `Adult`. This observation is consistent with the utility results presented in Table 2, which show a decrease in accuracy and AUC as the degree of negativity increases, particularly evident in the ↑ columns compared to the MLP. In contrast, `COMPAS` presents a greater challenge in ensuring utility while considering fairness, as indicated by the less distinct pattern in the learned embeddings, corroborated by the most significant utility drops in Table 2.

## D  MORE EXPERIMENTAL RESULTS

### D.1  EXPERIMENTS ON IMAGE DATASET

In this section, we present the experimental results on the `CelebA-A` image dataset. The `CelebA-A` face attributes dataset (Liu et al., 2015) contains over $200,000$ face images, where each image has $40$ human-labeled attributes. Among the attributes, we select 'Attractive' as a binary classification task and consider 'Gender' and 'Young' as sensitive attributes. The results are presented in Table 3.

The results show a similar finding with the tabular dataset, demonstrating that 1) `DP` method always achieves a lower $\Delta_{DP}$ but a relatively high $\Delta_{EO}$. 2) `HSIC` is a more promising fair model to achieve equal opportunity.

### D.2  MORE PREDICTION DISTRIBUTIONS OVER THE SENSITIVE GROUPS

As described in Section 5.3, the kernel density plots in this subsection visualize how the prediction distributions vary across sensitive groups, and how different fairness regularizers affect this alignment. In general, a **larger overlap** between the distributions of different sensitive groups indicates **better group fairness**. For each figure in this subsection, the top row shows the distributions of the raw

| | Methods | | ACC (%) | ↑ | AUC (%) | ↑ | $\Delta_{DP}$ (%) | ↓ | $\Delta_{EO}$ (%) | ↓ |
|---|---|---|---|---|---|---|---|---|---|---|
| | | | | Utility | | | | Fairness | | |
| ACS-T | | MLP | $66.21_{\pm0.95}$ | — | $73.78_{\pm0.25}$ | — | $8.32_{\pm2.67}$ | — | $5.11_{\pm3.55}$ | — |
| | Gender | DP | $65.38_{\pm0.29}$ | -1.25% | $72.40_{\pm0.38}$ | -1.87% | $\underline{0.29}_{\pm0.15}$ | 96.51% | $1.83_{\pm0.26}$ | 64.19% |
| | | MMD | $64.48_{\pm0.27}$ | -2.61% | $\underline{72.92}_{\pm0.31}$ | -1.17% | $1.22_{\pm0.36}$ | 85.34% | $2.11_{\pm0.49}$ | 58.71% |
| | | HSIC | $\mathbf{66.01}_{\pm0.29}$ | -0.30% | $\mathbf{73.16}_{\pm0.32}$ | -0.84% | $0.98_{\pm0.26}$ | 88.22% | $\underline{1.00}_{\pm0.28}$ | 80.43% |
| | | PR | $62.72_{\pm1.01}$ | -5.27% | $69.36_{\pm0.85}$ | -5.99% | $0.78_{\pm0.50}$ | 90.63% | $1.07_{\pm0.36}$ | 79.06% |
| | | **CS** | $\underline{65.70}_{\pm0.42}$ | -0.77% | $72.83_{\pm0.58}$ | -1.29% | $\mathbf{0.17}_{\pm0.08}$ | 97.96% | $\mathbf{0.75}_{\pm0.22}$ | 85.32% |
| | Race | MLP | $66.38_{\pm0.42}$ | — | $73.69_{\pm0.63}$ | — | $9.28_{\pm1.63}$ | — | $6.21_{\pm1.63}$ | — |
| | | DP | $64.96_{\pm0.23}$ | -2.14% | $71.86_{\pm0.23}$ | -2.48% | $\underline{0.82}_{\pm0.33}$ | 91.16% | $1.30_{\pm0.26}$ | 79.07% |
| | | MMD | $\underline{65.71}_{\pm0.65}$ | -1.01% | $70.57_{\pm0.52}$ | -4.23% | $3.97_{\pm0.97}$ | 57.22% | $1.55_{\pm0.79}$ | 75.04% |
| | | HSIC | $\mathbf{65.81}_{\pm0.24}$ | -0.86% | $\mathbf{72.92}_{\pm0.23}$ | -1.04% | $1.75_{\pm0.31}$ | 81.14% | $\mathbf{0.43}_{\pm0.23}$ | 93.08% |
| | | PR | $64.25_{\pm0.87}$ | -3.21% | $70.25_{\pm0.30}$ | -4.67% | $1.56_{\pm0.87}$ | 83.19% | $\underline{1.21}_{\pm0.74}$ | 80.52% |
| | | **CS** | $65.16_{\pm0.45}$ | -1.84% | $\underline{72.56}_{\pm0.72}$ | -1.41% | $\mathbf{0.55}_{\pm0.19}$ | 94.07% | $1.38_{\pm0.46}$ | 77.78% |
| CelebA-A | | RN | $78.14_{\pm0.47}$ | — | $86.58_{\pm0.55}$ | — | $51.66_{\pm0.97}$ | — | $35.67_{\pm1.11}$ | — |
| | Gender | DP | $62.42_{\pm4.79}$ | -20.12% | $66.86_{\pm3.19}$ | -22.78% | $\mathbf{0.46}_{\pm0.25}$ | 99.11% | $4.84_{\pm2.37}$ | 86.43% |
| | | MMD | $62.54_{\pm4.26}$ | -19.96% | $66.47_{\pm3.85}$ | -23.23% | $1.39_{\pm0.64}$ | 97.31% | $5.89_{\pm3.12}$ | 83.49% |
| | | HSIC | $63.39_{\pm3.63}$ | -18.88% | $69.33_{\pm3.25}$ | -19.92% | $2.24_{\pm0.36}$ | 95.66% | $\underline{3.83}_{\pm2.22}$ | 89.26% |
| | | PR | $\mathbf{65.51}_{\pm3.52}$ | -16.16% | $\mathbf{71.70}_{\pm2.88}$ | -17.19% | $4.00_{\pm0.52}$ | 92.26% | $5.05_{\pm2.57}$ | 85.84% |
| | | CS | $\underline{65.05}_{\pm3.80}$ | -16.75% | $\underline{71.42}_{\pm2.46}$ | -17.51% | $\underline{0.98}_{\pm0.62}$ | 98.10% | $\mathbf{1.53}_{\pm1.05}$ | 95.71% |
| | Young | RN | $78.14_{\pm0.47}$ | — | $86.67_{\pm0.53}$ | — | $41.74_{\pm1.17}$ | — | $18.35_{\pm1.56}$ | — |
| | | DP | $\mathbf{66.78}_{\pm3.61}$ | -14.54% | $\mathbf{73.95}_{\pm3.44}$ | -14.68% | $2.43_{\pm0.83}$ | 94.18% | $\underline{0.91}_{\pm1.77}$ | 95.04% |
| | | MMD | $65.82_{\pm4.87}$ | -15.77% | $72.84_{\pm3.61}$ | -15.96% | $3.49_{\pm0.83}$ | 91.64% | $1.60_{\pm0.71}$ | 91.28% |
| | | HSIC | $\underline{66.04}_{\pm3.01}$ | -15.49% | $73.08_{\pm2.69}$ | -15.68% | $1.99_{\pm0.55}$ | 95.23% | $1.04_{\pm0.60}$ | 94.33% |
| | | PR | $62.98_{\pm4.69}$ | -19.40% | $69.63_{\pm4.02}$ | -19.66% | $\underline{1.32}_{\pm0.49}$ | 96.84% | $1.82_{\pm0.53}$ | 90.08% |
| | | CS | $65.33_{\pm4.26}$ | -16.39% | $\underline{73.15}_{\pm3.84}$ | -15.60% | $\mathbf{1.28}_{\pm0.40}$ | 96.93% | $\mathbf{0.30}_{\pm0.12}$ | 98.37% |

Table 3: The fairness performance on the tabular dataset for existing fair models, and we consider race and gender as sensitive attributes. A higher accuracy metric indicates better performance. ↑ represents the accuracy improvement compared to MLP. A lower fairness metric indicates better fairness. ↓ represents the improvement of fairness compared to MLP. Green values denote better than MLP on the corresponding metric (ACC/AUC ↑; $\Delta_{DP}/\Delta_{EO}$↓), while red values denote worse. The results are based on 10 runs for all methods.

predictions $\hat{Y}$ for all target labels, grouped by a given sensitive attribute, while the bottom row focuses on the positive class.

For example, in Figure 7: the first row plots the prediction densities for *Race*: the blue shaded area corresponds to $\hat{Y} \mid \text{Race} = \text{Black}$ and the red shaded area corresponds to $\hat{Y} \mid \text{Race} = \text{White}$; the second row then plots the prediction densities for the positive class, i.e., $\hat{Y} = 1$, again conditioned on the two race groups (blue for $\hat{Y} = 1 \mid \text{Race} = \text{Black}$ and red for $\hat{Y} = 1 \mid \text{Race} = \text{White}$).

The other figures are interpreted analogously for their respective sensitive attributes. Since the degree of overlap can sometimes be difficult to judge by eye, we also print the corresponding group-fairness metric in each subfigure: the first row reports $\Delta_{\text{DP}}$ (demographic parity gap) and the second row reports $\Delta_{\text{EO}}$ (equalized odds gap).

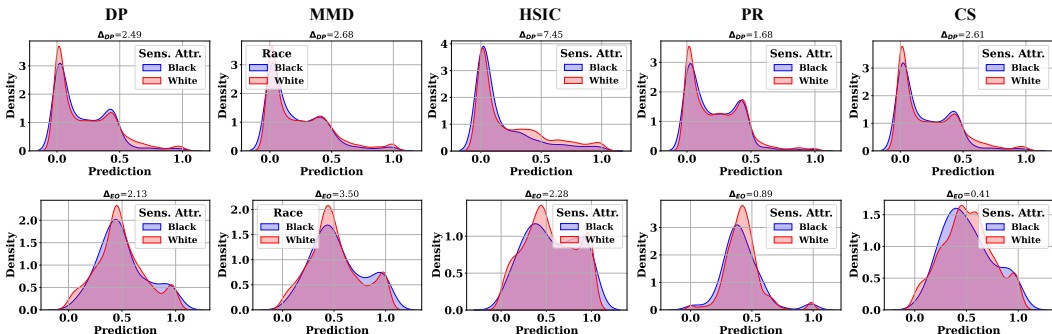

Figure 7: Prediction distributions for black and white groups in the `Adult` dataset.

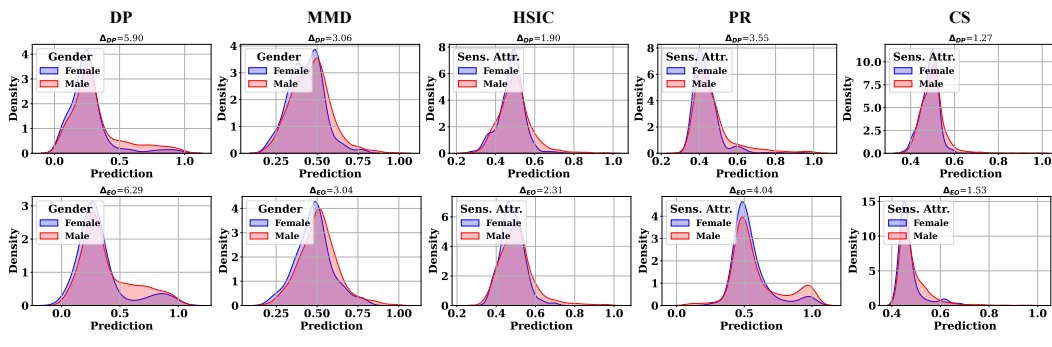

(a) Prediction distributions for female and male groups in the `COMPAS` dataset.

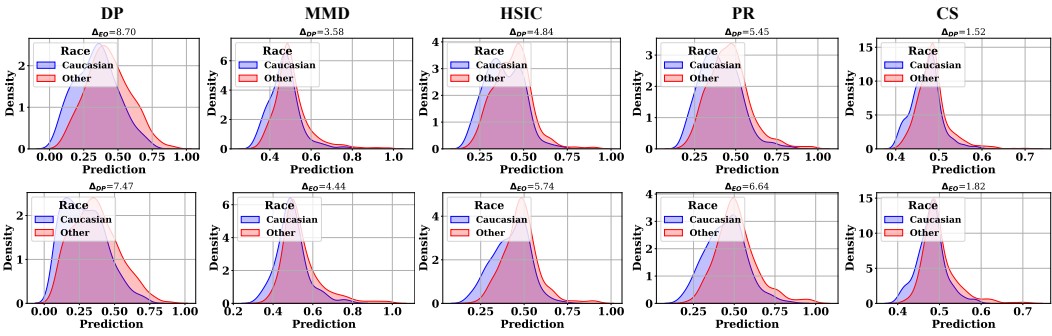

(b) Prediction distributions for Caucasian and (all) other groups in the `COMPAS` dataset.

Figure 8: Prediction distribution over gender and race in the `COMPAS` dataset.

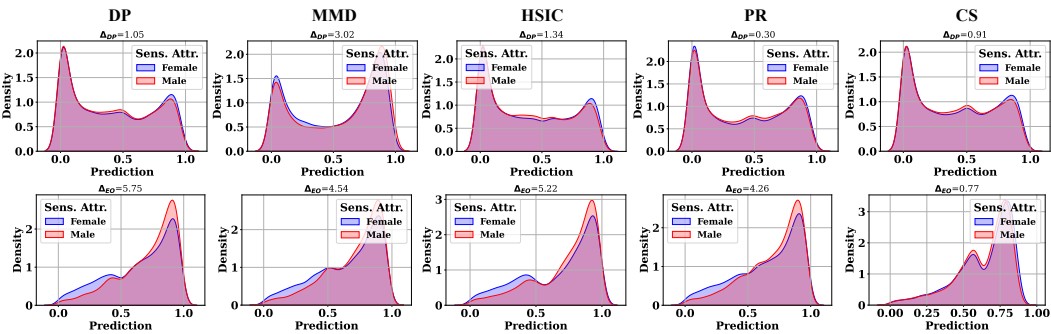

(a) Prediction distributions for female and male groups in the `ACS-I` dataset.

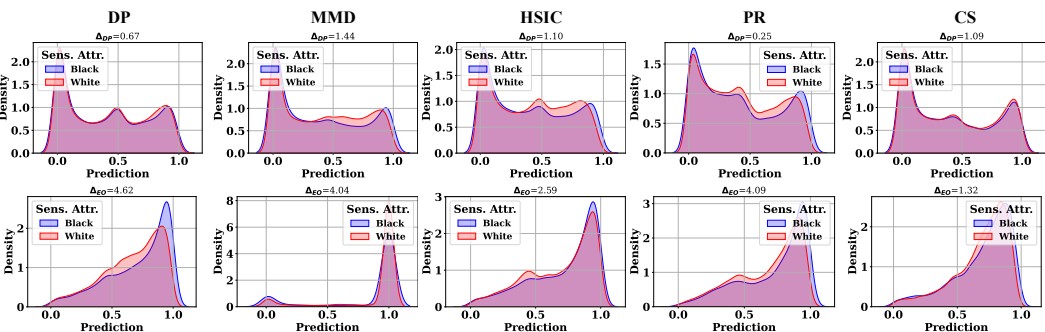

(b) Prediction distributions for black and white groups in the `ACS-I` dataset.

Figure 9: Accuracy and $\triangle_{DP}$ trade-off on `ACS-I` with sensitive attribute gender and race. Results located in the bottom-right corner are preferable.

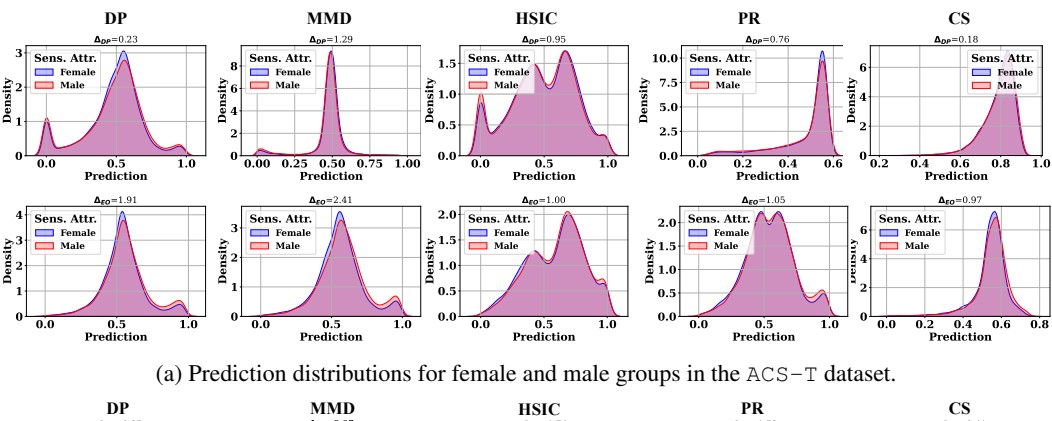

(a) Prediction distributions for female and male groups in the `ACS-T` dataset.

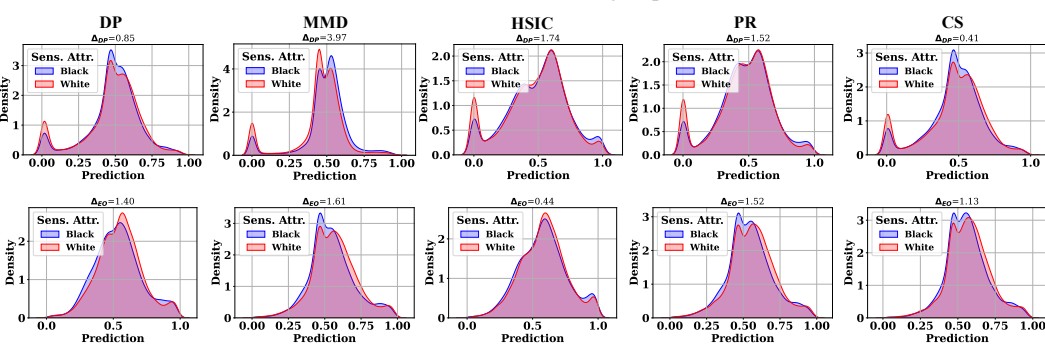

(b) Prediction distributions for black and white groups in the `ACS-T` dataset.

Figure 10: Accuracy and $\triangle_{DP}$ trade-off on `ACS-T` with sensitive attribute gender and race. Results located in the bottom-right corner are preferable.

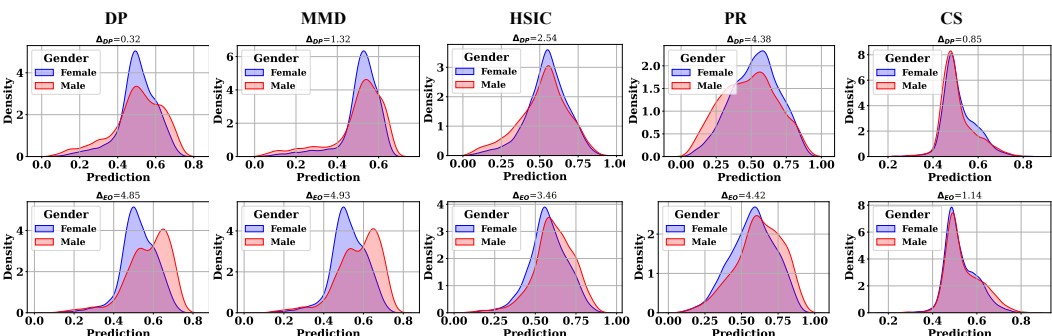

(a) Prediction distributions for female and male groups in the `CelebA-A` dataset.

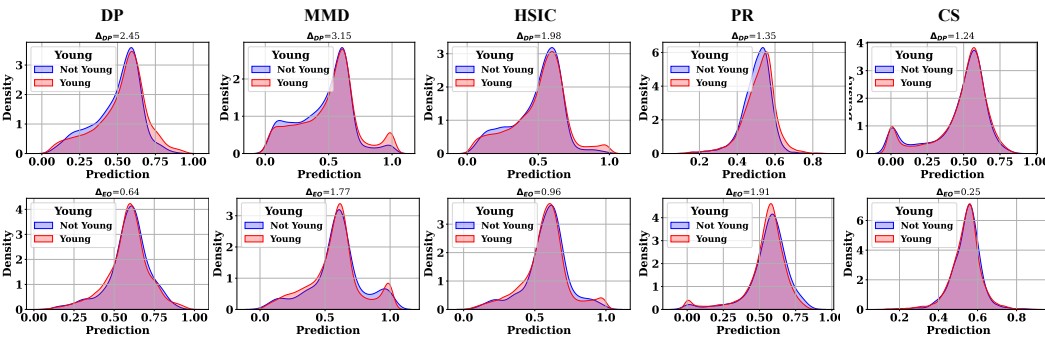

(b) Prediction distributions for young and non-yong groups in the `CelebA-A` dataset.

Figure 11: Accuracy and $\triangle_{DP}$ trade-off on `CelebA-A` with sensitive attribute gender and race. Results located in the bottom-right corner are preferable.

### D.3 MORE T-SNE PLOTS

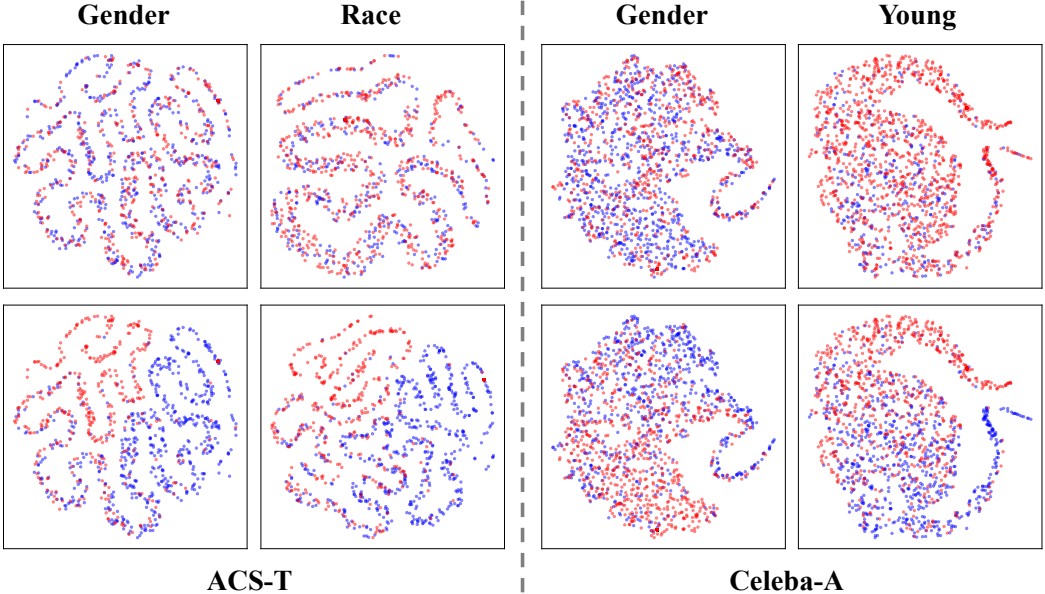

Figure 12: Accuracy and $\triangle_{DP}$ trade-off on `ACS-T` and `CelebA-A`. Results located in the bottom-right corner are preferable.

In addition to the T-SNE plots shown in Figure 6, which show the results on three datasets, we also include the T-SNE plots on the two remaining datasets `ACS-T` and `CelebA-A` in Figure 12.

## E DATASET DESCRIPTIONS AND DETAILS

We conducted experiments on five datasets, including four tabular datasets and one image dataset. The introduction of these datasets is as follows:

- **Adult**[4] (Dua & Graff, 2017) The `Adult` dataset includes data from $45,222$ individuals based on the 1994 US Census. The primary task is to predict whether an individual's income exceeds \$50k USD, using various personal attributes. In this analysis, we focus on gender and race as sensitive attributes.
- **COMPAS**[5] (Larson et al., 2016) The `COMPAS` dataset contains records of criminal defendants and is designed to predict the likelihood of recidivism within two years. It encompasses various attributes related to the defendants, including their criminal history, gender, and race.
- **ACS-I** and **ACS-T**[6] (Ding et al., 2021) The ACS dataset is derived from the American Community Survey (ACS) Public Use Microdata Sample and encompasses several prediction tasks. These tasks include predicting whether an individual's income exceeds \$50k and whether an individual is employed, with features such as race, gender, and other relevant characteristics tailored to each task.
- **CelebA-A**[7] (Liu et al., 2015) The CelebFaces Attributes dataset comprises $20,000$ face images of $10,000$ distinct celebrities. Each image is annotated with 40 binary labels representing various facial attributes, including gender, hair color, and age. In this study, we focus on the 'attractive' label for a binary classification task, while considering 'young' and 'gender' as sensitive attributes.

The detailed statistics for the aforementioned datasets are summarized as follows:

---

[4]https://archive.ics.uci.edu/ml/datasets/adult
[5]https://github.com/propublica/compas-analysis
[6]https://github.com/zykls/folktables
[7]https://mmlab.ie.cuhk.edu.hk/projects/CelebA.html

| Dataset | Task | Sen. Attr. $(S)$ | #Samples | #Feat. | Class $Y$ $0:1$ | 1st $S$ $0:1$ | 2nd $S$ $0:1$ |
|---------|------|------------------|----------|--------|-----------------|---------------|---------------|
| Adult | Income | Gender, Race | $45,222$ | $101$ | $1:0.33$ | $1:2.08$ | $1:9.20$ |
| COMPAS | Credit | Gender, Race | $6,172$ | $405$ | $1:0.83$ | $1:4.17$ | $1:0.52$ |
| ACS-I | Income | Gender, Race | $195,665$ | $908$ | $1:0.70$ | $1:0.89$ | $1:1.62$ |
| ACS-T | Travel Time | Gender, Race | $172,508$ | $1,567$ | $1:0.94$ | $1:0.89$ | $1:1.61$ |
| CelebA-A | Attractive | Gender, Young | $202,599$ | $48 \times 48$ | $1:0.95$ | $1:0.71$ | $1:3.45$ |

Table 4: The table presents the statistics of the datasets. #Feat. refers to the total number of features after preprocessing[8]. The ratio $0:1$ represents the proportion between the two categories of the target label or sensitive attributes.

## F  BASELINES DETAILS

We consider four widely used fairness methods: DP, MMD, HSIC, and PR. Specifically, DP and HSIC minimize the demographic parity and Hilbert-Schmidt Independence Criterion, respectively. MMD learns a classifier that optimizes the Mean Maximum Discrepancy. We also include base models MLP and RN for tabular data and image data, respectively.

- DP: It is a gap regularization method for demographic parity (Chuang & Mroueh, 2020). As these fairness definitions cannot be optimized directly, gap regularization is differentiable and can be optimized using gradient descent.
- MMD: The Maximum Mean Discrepancy (MMD) (Gretton et al., 2012) is a metric used to measure the distance between probability distributions. Previous research has leveraged MMD to enhance fairness in machine learning models, specifically in variational autoencoders (Louizos et al., 2016) and MLPs (Deka & Sutherland, 2023). In this paper, we build on the methodologies from earlier works (Zhao & Meng, 2015) to compute the MMD baseline.
- HSIC: It minimizes the Hilbert-Schmidt Independence Criterion between the prediction accuracy and the sensitive attributes (Gretton et al., 2005; Baharlouei et al., 2020; Li et al., 2019).
- Prejudice Remover (PR) (Kamishima et al., 2012) (Prejudice Remover) minimizes the prejudice index, which is the mutual information between the prediction accuracy and the sensitive attributes.

## G  MORE FAIRNESS DEFINITIONS

## H  DETAILS OF THE GROUP FAIRNESS

In this section, we provide the details of the group fairness. We first introduce the definition of group fairness. Then, we introduce the existing group fairness metrics and algorithms.

- DP (Demographic Parity or Statistical Parity) (Zemel et al., 2013). A classifier satisfies demographic parity if the predicted outcome $\hat{Y}$ is independent of the sensitive attribute $S$, i.e., $P(\hat{Y} \mid S = 0) = P(\hat{Y} \mid S = 1)$.
- prule (Zafar et al., 2017). A classifier satisfies $p\%$-rule if the ratio between the probability of subjects having a certain sensitive attribute value assigned the positive decision outcome and the probability of subjects not having that value also assigned the positive outcome should be no less than $p/100$, i.e., $|P(\hat{Y} = 1 \mid S = 1)/P(\hat{Y} = 1 \mid S = 0)| \leq p/100$.
- EOpp (Equality of Opportunity) (Hardt et al., 2016). A classifier satisfies equalized opportunity if the predicted outcome $Y$ is independent of the sensitive attribute $S$ when the label $Y = 1$, i.e., $P(\hat{Y} \mid S = 0, Y = 1) = P(\hat{Y} \mid S = 1, Y = 1)$.
- EOdd (Equalized Odds) (Hardt et al., 2016). A classifier satisfies equalized odds if the predicted outcome $Y$ is independent of the sensitive attribute $S$ conditioned on the label $Y$, i.e., $P(\hat{Y} \mid S = 0, Y = y) = P(\hat{Y} \mid S = 1, Y = y), y \in \{0, 1\}$.
- ACC (Accuracy Parity). A classifier satisfies accuracy parity if the error rates of different sensitive attribute values are the same, i.e., $P(\hat{Y} \neq Y \mid S = 0) = P(\hat{Y} \neq Y \mid S = 1), y \in \{0, 1\}$.
- aucp (ROC AUC Parity). A classifier satisfies ROC AUC parity if its area under the receiver operating characteristic curve with w.r.t. different sensitive attribute values is the same.

- ppv (Predictive Parity Value Parity) A classifier satisfies predictive parity value parity if the probability of a subject with a positive predictive value belonging to the positive class w.r.t. different sensitive attribute values are the same, i.e., $P(Y = 1 \mid \hat{Y}, S = 0) = P(Y = 1 \mid \hat{Y}, S = 1)$.

- bnegc (Balance for Negative Class). A classifier satisfies balance for the negative class if the average predicted probability of a subject belonging to the negative class is the same w.r.t. different sensitive attribute values, i.e., $\mathbb{E}[f(X) \mid Y = 0, S = 0] = \mathbb{E}[f(X) \mid Y = 0, S = 1]$.

- bposc (Balance for Positive Class). A classifier satisfies balance for the negative class if the average predicted probability of a subject belonging to the positive class is the same w.r.t. different sensitive attribute values, i.e., $\mathbb{E}[f(X) \mid Y = 1, S = 0] = \mathbb{E}[f(X) \mid Y = 1, S = 1]$.

- abcc (Area Between Cumulative density function Curves) (Han et al., 2023) is proposed to precisely measure the violation of demographic parity at the distribution level. The new fairness metrics directly measure the difference between the distributions of the prediction probability for different demographic groups

# I  ADDITION EXPERIMENTS

## I.1  ADDITION EXPERIMENTS ON MORE FAIRNESS METRICS

We provide additional results comparing our framework with baselines under the following fairness notions: Predictive Parity (PPV) (Chouldechova, 2017), p%-Rule (PRULE) (Zafar et al., 2017), Balance for Positive Class (BFP) (Kleinberg et al., 2016), and Balance for Negative Class (BFN) (Kleinberg et al., 2016). The dataset is Adult, using gender as the sensitive attribute. All other experimental settings are consistent with Table 1 in the paper.

| Method | $\Delta_{PPV}$ ($\downarrow$) | PRULE ($\uparrow$) | $\Delta_{BFP}$ ($\downarrow$) | $\Delta_{BFN}$ ($\downarrow$) |
|---|---|---|---|---|
| DP | $27.35 \pm 5.64$ | $81.21 \pm 9.04$ | $\mathbf{11.25 \pm 2.75}$ | $5.15 \pm 0.44$ |
| MMD | $35.19 \pm 6.33$ | $85.83 \pm 7.15$ | $18.32 \pm 3.74$ | $3.49 \pm 0.25$ |
| HSIC | $37.25 \pm 3.19$ | $\underline{96.18 \pm 2.12}$ | $16.47 \pm 1.21$ | $4.04 \pm 0.32$ |
| PR | $\mathbf{25.46 \pm 3.17}$ | $89.57 \pm 7.39$ | $21.45 \pm 2.37$ | $\underline{3.46 \pm 0.28}$ |
| CS | $31.59 \pm 4.35$ | $\mathbf{97.75 \pm 3.24}$ | $\underline{15.25 \pm 2.58}$ | $\mathbf{3.18 \pm 0.36}$ |

Table 5: Fairness performance comparison on the Adult dataset, with gender as the sensitive attribute under $\Delta_{PPV}$, PRULE, $\Delta_{BFP}$, and $\Delta_{BFN}$.

We observe the following:

- CS generally achieves the best fairness trade-off performance across the four tested fairness notions.

- On the Adult, BFN is generally minimized more effectively than BFP.

- Since BFN is related to EO, the ranking of $\Delta_{\text{BFN}}$ aligns with $\Delta_{\text{EO}}$ in Table 1 of the paper. Note that, as stated in previous studies (Kleinberg et al., 2016), there is an inherent trade-off between BFP and BFN in practice.

## I.2  ADDITIONAL EXPERIMENTS ON COMBINING MULTIPLE REGULARIZER TERMS SIMULTANEOUSLY

We conducted additional experiments where we combined both KL divergence and CS divergence as regularizers. The experiments were performed on the Adult, with gender as the sensitive attribute.

Actually, combining multiple fairness objectives has several drawbacks, which is why most existing studies avoid using multiple regularizers. Instead, they often choose to add simple constraint terms. The key drawbacks of combining fairness regularizers are summarized as follows:

---

[8] We adopt the preprocessing in previous studies (Le Quy et al., 2022; Mehrabi et al., 2021) involving identifying the target labels and sensitive attributes, and then selecting the relevant features for the analysis.

| Method | $\Delta_{DP}$ ($\downarrow$) | $\Delta_{EO}$ ($\downarrow$) |
|---|---|---|
| DP | $1.29 \pm 0.95$ | $20.15 \pm 1.13$ |
| CS | $2.42 \pm 0.85$ | $2.27 \pm 1.04$ |
| KL | $2.77 \pm 0.86$ | $10.42 \pm 4.34$ |
| CS+KL | $2.46 \pm 1.25$ | $13.42 \pm 6.12$ |
| CS+0.5KL | $2.25 \pm 1.14$ | $9.33 \pm 6.36$ |

Table 6: The fairness performance on `Adult` (gender).

- The CS divergence is upper-bounded by the KL divergence. Therefore, adding KL as an additional fairness objective is theoretically redundant and will not provide further benefits.

- Adding KL or other fairness metrics increases computational complexity, making the optimization process more challenging.

These experimental results further show the significance of our contribution: identifying a suitable, tighter-bounded fairness regularizer that balances effectiveness and computational efficiency.

## I.3 ADDITIONAL EXPERIMENTAL ON EODD AND EOPP

In this subsection, we provide additional experimental results on EOdd and EOpp, defined in Appendix H. Each line represents the results for one of the hyperparameter values $\alpha = 0.2/0.8/1.4$, denoted as 'lam' in the legend. We observe that both EOdd and EOpp regularizers perform well on the EO metrics but do not perform as well on the DP metrics. We summarize the results as follows:

Table 7: Fairness performance comparison on the ACS-I (gender) on additional metrics.

| **Method** | $\Delta_{\text{PPV}}$ ($\downarrow$) | PRULE ($\uparrow$) | $\Delta_{\text{BFP}}$ ($\downarrow$) | $\Delta_{\text{BFN}}$ ($\downarrow$) | $\Delta_{\text{DP}}$ ($\downarrow$) | $\Delta_{\text{EO}}$ ($\downarrow$) |
|---|---|---|---|---|---|---|
| DP | $11.86 \pm 6.54$ | $\mathbf{97.14 \pm 12.41}$ | $4.83 \pm 4.35$ | $3.97 \pm 4.02$ | $0.96 \pm 0.22$ | $5.37 \pm 0.32$ |
| EOdd | $8.38 \pm 1.28$ | $84.75 \pm 3.88$ | $\underline{0.43 \pm 0.63}$ | $\mathbf{0.31 \pm 0.73}$ | $5.76 \pm 1.42$ | $\underline{0.64 \pm 1.25}$ |
| Eopp | $\underline{8.00 \pm 1.65}$ | $83.36 \pm 4.41$ | $\mathbf{0.34 \pm 1.00}$ | $\underline{1.24 \pm 1.43}$ | $6.32 \pm 0.81$ | $\mathbf{0.52 \pm 1.29}$ |
| **CS** | $\mathbf{7.00 \pm 8.35}$ | $\underline{96.90 \pm 8.35}$ | $4.39 \pm 1.32$ | $2.77 \pm 1.83$ | $\mathbf{0.77 \pm 0.38}$ | $0.90 \pm 0.46$ |

The observations from the results align with our claims, and the CS regularizer demonstrates significant effectiveness:

- As shown in Table 2 of our paper, the MLP achieves a $\triangle_{EO}$ of $2.13 \pm 3.64$, whereas the DP regularizer gets a higher $\triangle_{EO}$ of $3.97 \pm 4.02$. This indicates that the DP regularizer does not effectively optimize and may even negatively affect EO fairness.

- EO-based methods (EOdd and EOpp) show the worst performance in terms of DP fairness, even compared to other baselines such as MMD, HSIC, and PR (as reported in Table 2 in the paper). In particular, EOpp reaches approximately 15 in $\triangle_{DP}$ on the Adult dataset, as shown in the Appendix K. This high $\triangle_{DP}$ is consistently observed across different hyperparameters ($\alpha = 0.2, 0.8, 1.4$, referred to as 'lam' in the figure).

## I.4 ADDITIONAL EXPERIMENTS ON PRE-PROCESSING AND POST-PROCESSING BASELINES

We have added a post-processing method, PostEO (Hardt et al., 2016) on the `Adult` dataset (with gender as the sensitive attribute).

The PostPro method is specifically designed to optimize for EO (Hardt et al., 2016), which explains its lower $\Delta_{EO}$.

However, both pre-processing and post-processing methods share a common limitation: they result in lower utility (ACC or AUC). Considering the need for a balanced trade-off between fairness and utility, CS emerges as the most favorable option in our comparison.

| Method | ACC ($\uparrow$) | AUC ($\uparrow$) | $\triangle_{DP}$ ($\downarrow$) | $\triangle_{EO}$ ($\downarrow$) |
|---|---|---|---|---|
| DP | $82.42 \pm 0.39$ | $86.91 \pm 0.80$ | $1.29 \pm 0.95$ | $20.15 \pm 1.13$ |
| CS | $83.31_{\pm 0.47}$ | $90.15_{\pm 0.49}$ | $2.42 \pm 0.85$ | $2.27 \pm 1.04$ |
| PR | $81.81 \pm 0.52$ | $85.38 \pm 0.82$ | $0.71 \pm 0.40$ | $12.45 \pm 2.38$ |
| PostPro | $80.25 \pm 0.83$ | $84.35 \pm 0.98$ | $5.75 \pm 1.67$ | $2.12 \pm 1.44$ |

Table 8: Comparison of methods on various metrics.

## I.5 ADDITIONAL BASELINES WITH DEPENDENCE MEASURES

To further contextualize our CS-based fairness regularizer, we additionally compare against several classical dependence measures between the model prediction $\hat{Y}$ and the sensitive attribute $S$: Hirschfeld–Gebelein–Rényi maximal correlation (HGR), mutual information (MI), and (distance) covariance (dCov) or empirical distance covariance. In all cases, we regularize the model by penalizing the corresponding dependence between $\hat{Y}$ and $S$.

**HGR maximal correlation (HGR).** The Hirschfeld–Gebelein–Rényi maximal correlation (Hirschfeld, 1935; Gebelein, 1941; Rényi, 1959) between two random variables $X$ and $S$ is defined as:

$$\rho_{\mathrm{HGR}}(X, S) = \sup_{f,g} \mathrm{Corr}\big(f(X), g(S)\big) \ \text{ s.t. } \ \mathbb{E}[f(X)] = \mathbb{E}[g(S)] = 0, \ \mathbb{E}[f(X)^2] = \mathbb{E}[g(S)^2] = 1, \tag{38}$$

where the supremum is taken over square-integrable functions $f$ and $g$. We instantiate an HGR-based fairness regularizer by penalizing $\rho_{\mathrm{HGR}}(\hat{Y}, S)$ using a neural estimator.

**Mutual information (MI).** Mutual information (Cover, 1999) between $X$ and $S$ is

$$I(X; S) = \iint p_{X,S}(x, s) \log \frac{p_{X,S}(x, s)}{p_X(x) \, p_S(s)} \, \mathrm{d}x \, \mathrm{d}s, \tag{39}$$

or, for discrete variables,

$$I(X; S) = \sum_x \sum_s p_{X,S}(x, s) \log \frac{p_{X,S}(x, s)}{p_X(x) \, p_S(s)}. \tag{40}$$

Equivalently, MI can be written as a Kullback–Leibler divergence

$$I(X; S) = D_{\mathrm{KL}}\big(p_{X,S} \,\big\|\, p_X p_S\big). \tag{41}$$

Our MI-based baseline regularizes the mutual information between the prediction and the sensitive attribute, $I(\hat{Y}; S)$, using a differentiable estimator.

**Distance covariance (dCov) and empirical distance covariance.** Distance covariance (Székely et al., 2007) between $X \in \mathbb{R}^p$ and $S \in \mathbb{R}^q$ is defined via their joint and marginal characteristic functions $\varphi_{X,S}$, $\varphi_X$, and $\varphi_S$ as

$$\mathrm{dCov}^2(X, S) = \int_{\mathbb{R}^{p+q}} \left| \varphi_{X,S}(t, u) - \varphi_X(t) \, \varphi_S(u) \right|^2 w(t, u) \, \mathrm{d}t \, \mathrm{d}u, \tag{42}$$

for a suitable weight function $w(t, u)$. In practice, we use the standard empirical distance covariance estimator. Given samples $\{(x_i, s_i)\}_{i=1}^n$, define pairwise distances $a_{ij} = \|x_i - x_j\|$ and $b_{ij} = \|s_i - s_j\|$, and their double-centered versions:

$$A_{ij} = a_{ij} - \bar{a}_{i\cdot} - \bar{a}_{\cdot j} + \bar{a}_{\cdot\cdot}, \tag{43}$$

$$B_{ij} = b_{ij} - \bar{b}_{i\cdot} - \bar{b}_{\cdot j} + \bar{b}_{\cdot\cdot}, \tag{44}$$

where $\bar{a}_{i\cdot}$ and $\bar{a}_{\cdot j}$ denote row and column means, and $\bar{a}_{\cdot\cdot}$ is the grand mean (and analogously for $b$). The empirical distance covariance is then:

$$\widehat{\mathrm{dCov}}^2(X, S) = \frac{1}{n^2} \sum_{i=1}^n \sum_{j=1}^n A_{ij} B_{ij}. \tag{45}$$

Our dCov-based regularizer penalizes $\widehat{\mathrm{dCov}}^2(\hat{Y}, S)$.

**Experimental setup.** To assess how our CS-based regularizer compares with other classical dependence measures, we conduct an additional experiment on the ADULT dataset using *gender* as the sensitive attribute. We keep **all settings identical** to the main experiment on ADULT in Section 5: the same data split and preprocessing, the same MLP classifier architecture, optimizer, batch size, learning rate, number of epochs, and the same protocol for tuning the fairness-regularization weight. The only change is the choice of the fairness loss term $\mathcal{L}_{\text{fair}}$.

Concretely, for each additional baseline, we replace the CS-based fairness loss in Equation (13) with the corresponding dependence measure between the model prediction $\hat{Y}$ and the sensitive attribute $S$: (i) the **HGR** baseline penalizes the Hirschfeld–Gebelein–Rényi maximal correlation $\rho_{\text{HGR}}(\hat{Y}, S)$ using a neural estimator; (ii) the **MI** baseline penalizes the mutual information $I(\hat{Y}; S)$ between prediction and sensitive attribute, estimated with a differentiable MI estimator; and (iii) the **dCov** baseline penalizes the empirical distance covariance $\widehat{\text{dCov}}^2(\hat{Y}, S)$ defined before. In all cases, the overall training objective retains the same form as Equation (1), and we evaluate the resulting models on accuracy, AUC, $\Delta_{\text{DP}}$, and $\Delta_{\text{EO}}$ using the same test split as in the main experiments. The results are summarized in Table 9.

| Method | ACC ($\uparrow$) | AUC ($\uparrow$) | $\triangle_{DP}$ ($\downarrow$) | $\triangle_{EO}$ ($\downarrow$) |
|---|---|---|---|---|
| HGR | $80.13 \pm 1.35$ | $84.20 \pm 1.27$ | $3.82 \pm 0.84$ | $\underline{6.82} \pm 3.77$ |
| dCov | $\underline{82.31} \pm 0.62$ | $\underline{85.39} \pm 0.89$ | $4.75 \pm 1.67$ | $12.41 \pm 1.44$ |
| PR | $81.81 \pm 0.52$ | $85.38 \pm 0.82$ | $\mathbf{0.71} \pm 0.40$ | $12.45 \pm 2.38$ |
| CS | $\mathbf{83.31} \pm 0.47$ | $\mathbf{90.15} \pm 0.49$ | $\underline{2.42} \pm 0.85$ | $\mathbf{2.27} \pm 1.04$ |

Table 9: Additional comparison of with HGR and dCov.

From the results in Table 9, we observe that the proposed CS-based regularizer achieves the best overall performance among all dependence-based baselines: it attains the highest ACC and AUC while keeping both $\Delta_{\text{DP}}$ and $\Delta_{\text{EO}}$ low, confirming that CS offers a robust utility–fairness trade-off. More specifically, (i) **HGR** is theoretically a very strong dependence measure, but in practice it requires a neural estimator of the maximal correlation, which makes optimization noisy and sensitive to hyperparameters; this is reflected in its relatively low ACC/AUC and larger standard deviations, although its fairness metrics are still competitive, indicating that it can reduce dependence when the optimization succeeds. (ii) **dCov** is a kernel-based or distance-based statistic with a closed-form empirical estimator, so it is easier to optimize and leads to higher ACC/AUC and smaller variance than HGR; however, its fairness performance is weaker, suggesting that penalizing average pairwise distances between prediction and sensitive-feature embeddings is less aligned with group-rate gaps than the density-ratio style CS divergence, which yields tighter control over the discrepancies that drive $\Delta_{\text{DP}}$ and $\Delta_{\text{EO}}$. (iii) **PR** (an MI-based regularizer) achieves very small $\Delta_{\text{DP}}$, consistent with its design of directly reducing mutual information between $\hat{Y}$ and $S$ and thereby aligning the marginal prediction rates across groups, but its $\Delta_{\text{EO}}$ remains large and its utility is moderate, as MI does not explicitly constrain the conditional error rates $P(\hat{Y} \mid Y, S)$ that underlie equalized odds. Overall, these observations support CS as the most balanced choice among the considered dependence measures.

**Discussion: Relation between MI and our KL- and PR-based regularizers.** For completeness, we briefly discuss how the generic MI baseline above relates to the KL-based fairness losses and the Prejudice Remover (PR) baseline used in our main experiments. By definition, mutual information is a Kullback–Leibler divergence between the joint distribution and the product of the marginals:

$$I(X; S) = D_{\text{KL}}(p_{X,S} \,\|\, p_X p_S). \tag{46}$$

Thus, MI and KL belong to the same family of information-theoretic discrepancy measures: MI uses KL to quantify *any* deviation of $p_{X,S}$ from independence, while many fairness regularizers based on KL (including the KL term used in our fairness-loss landscape in Figure 2) penalize Kullback–Leibler divergences between *conditional* distributions, such as $D_{\text{KL}}(p_{\hat{Y}|S=0} \,\|\, p_{\hat{Y}|S=1})$. These conditional KL penalties are closely related to MI but not identical: if all $p_{\hat{Y}|S=s}$ coincide, then both the conditional KL and $I(\hat{Y}; S)$ vanish, yet vanishing conditional KL for one pair of groups does not necessarily minimize the full KL between $p_{\hat{Y},S}$ and $p_{\hat{Y}} p_S$.

The Prejudice Remover (PR) (Kamishima et al., 2012) can be viewed as an explicit MI-based regularizer. PR minimizes the *prejudice index*, which is defined as the mutual information between the prediction and the sensitive attribute, $I(\hat{Y}; S)$, under a log-linear model of the conditional odds. In this sense, PR instantiates the generic MI regularization principle with a particular parametric form and optimization scheme.

In summary, the generic MI baseline, the KL-based fairness penalties, and PR all enforce independence between $\hat{Y}$ and $S$ using KL divergence in different guises: MI regularizes the KL divergence between $p_{\hat{Y},S}$ and $p_{\hat{Y}}p_S$; Our KL-based fairness loss penalizes KL divergences between group-conditional predictions distributions; and PR minimizes a parametric approximation of $I(\hat{Y}; S)$. Our CS-based regularizer complements this family by replacing the KL-based dependence measure with the Cauchy–Schwarz divergence, which enjoys closed-form kernel estimators and the tighter bounds analyzed in the main text.

### I.6 ADVERSARIAL METHODS EXPERIMENTS

We conducted additional experiments using Adversarial Debiasing (Louppe et al., 2017), which we refer to as ADV below.

| Method | ACC ($\uparrow$) | AUC ($\uparrow$) | $\Delta_{\text{DP}}$ ($\downarrow$) | $\Delta_{\text{EO}}$ ($\downarrow$) |
|---|---|---|---|---|
| DP | $82.42 \pm 0.39$ | $86.91 \pm 0.80$ | $1.29 \pm 0.95$ | $20.15 \pm 1.13$ |
| CS | $83.04 \pm 0.51$ | $90.84 \pm 0.35$ | $2.13 \pm 0.89$ | $2.35 \pm 1.15$ |
| ADV | $81.58 \pm 1.26$ | $83.08 \pm 0.75$ | $16.3 \pm 7.5$ | $14.2 \pm 8.6$ |

Table 10: Additional experiment on the fairness performance of ADV on the Adult dataset (gender attribute).

From Table 10, we observe that:

- The ADV method exhibits lower utility (in terms of ACC and AUC) and higher $\triangle_{DP}$ compared to both the DP and CS fairness regularizers. It also performs worse than the CS regularizer in terms of $\triangle_{EO}$.

- ADV also shows a higher variance in accuracy, likely due to the greater difficulty of optimizing adversarial objectives compared to the DP and CS regularization approaches.

## J MORE EXPERIMENTAL DETAILS

In this section, we describe the details of the experimental setup. In this work, we adopted a straightforward stopping strategy. We employ a linear decay strategy for the learning rate, halving it every 50 training step. The model training is stopped when the learning rate decreases to a value below $1e^{-5}$. Across all datasets, we use a weight decay of 0.0, StepLR with a step size of 50 and a gamma value of 0.1, and train for 150 epochs using the Adam Optimizer (Kingma & Ba, 2014). The batch size and learning rate vary depending on the dataset, with specific values provided below. Additionally, Table 11 lists the range of the control hyperparameter $\beta$ for each fairness approach. The experiments were executed using NVIDIA RTX A4000 GPUs with 16GB GDDR6 Memory.

### J.1 HYPERPARAMETER SETTINGS

**1. Training Hyperparameters:**

- Tabular data (Adult, COMPAS, ACS-I, and ACS-T):
    - Learning rate: $1e^{-2}$
    - Weight decay: 0.0
    - StepLR_step: 50
    - StepLR_gamma: 0.1
    - Training epochs: 150

- Batch sizes: $1,024$ on `Adult`, $32$ on `COMPAS`, $4,096$ on `ACS-I`, $4,096$ on `ACS-T`
- Image data (`CelebA-A`):
  - Learning rate: $1e^{-3}$
  - Weight decay: 0.0
  - StepLR_step: 50
  - StepLR_gamma: 0.1
  - Training epochs: 150
  - Batch sizes: 256.

## 2. Architecture Hyperparameters:

- Multilayer perceptron:
  - Number of layers: 3
  - Number of hidden neurons: $\{512, 256, 64\}$
- ResNet-18 (He et al., 2016):
  - Model: https://github.com/pytorch/vision/blob/main/torchvision/models/resnet.py

## J.2 HYPERPARAMETER SELECTION

To implement `CS` and the baseline methods, we adjust the hyperparameter $\beta$ by tuning it within a specified range. The details of the hyperparameter selection process and the specific range for $\beta$ are provided below:

| Method | Fairness Control Hyperparameter $\beta$ |
|--------|------------------------------------------|
| DP | $0.5, 1.0, 1.2, 1.4, 1.6, 1.8, 2.0, 2.5, 3.0, 3.5, 4$ |
| HSIC | $0.1, 1, 5, 10, 50, 100, 200, 300, 400, 500, 600, 700, 800, 900, 1,000$ |
| PR | $0.05, 0.2, 0.3, 0.40, 0.50, 0.7, 0.9, 1.0$ |
| ADV | $0.5, 1.0, 1.2, 1.4, 1.6, 1.8, 2.0, 2.5, 3.0, 3.5$ |
| CS | $1e^{-6}, 1e^{-5}, 1e^{-4}, 1e^{-3}, 1e^{-2}, 2e^{-2}, 5e^{-2}, 0.1, 0.5, 1.0, 2.0, 3.0, 4.0, 50, 150$ |

Table 11: The selections of fairness control hyperparameter blue$\beta$.

# K  CURVES

In this section, we show some important curves we recorded.

*Note: The line represents the mean values, and the shaded area indicates the variation across all runs.*

## K.1  EODD ADDITIONAL RESULTS

**[Adult (Gender)]**

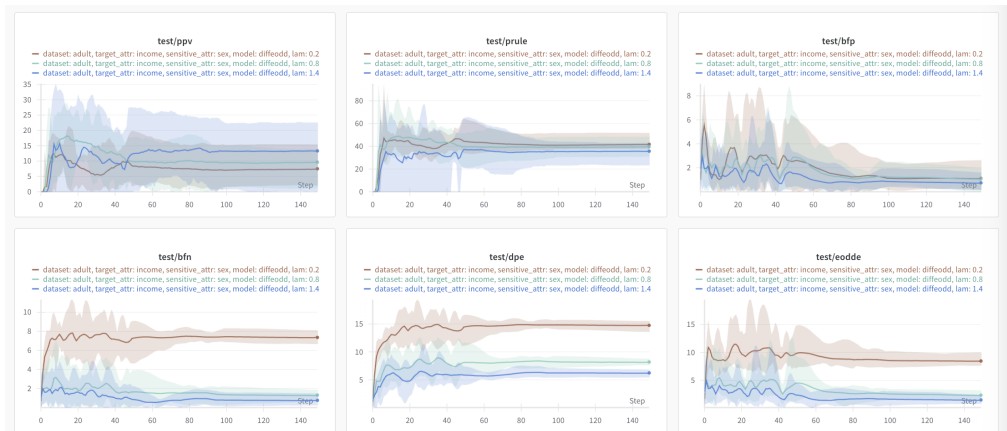

Figure 13: Other metrics ($\triangle_{PPV}$ ($\downarrow$), $PRULE(\uparrow)$, $\triangle_{BFP}$ ($\downarrow$), $\triangle_{BFN}$ ($\downarrow$) ), $\triangle_{DP}$ (Shown as 'dpe'), and eodde.

**Adult (Gender)**

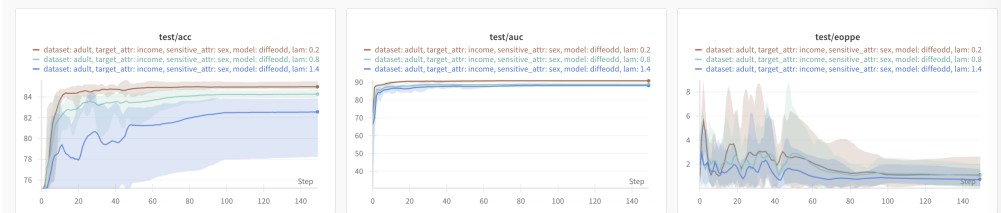

Figure 14: ACC, AUC and $\triangle_{EO}$ (Shown as "eoppe")

**ACS-I (Gender)**

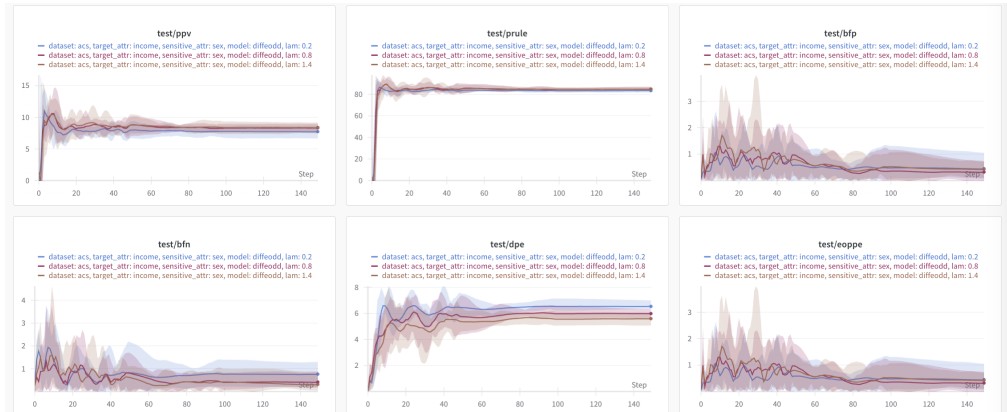

Figure 15: Other metrics ($\triangle_{PPV}$ ($\downarrow$), $PRULE(\uparrow)$, $\triangle_{BFP}$ ($\downarrow$), $\triangle_{BFN}$ ($\downarrow$) ), $\triangle_{DP}$ (Shown as 'dpe'), and eodde.

**ACS-I (Gender)**

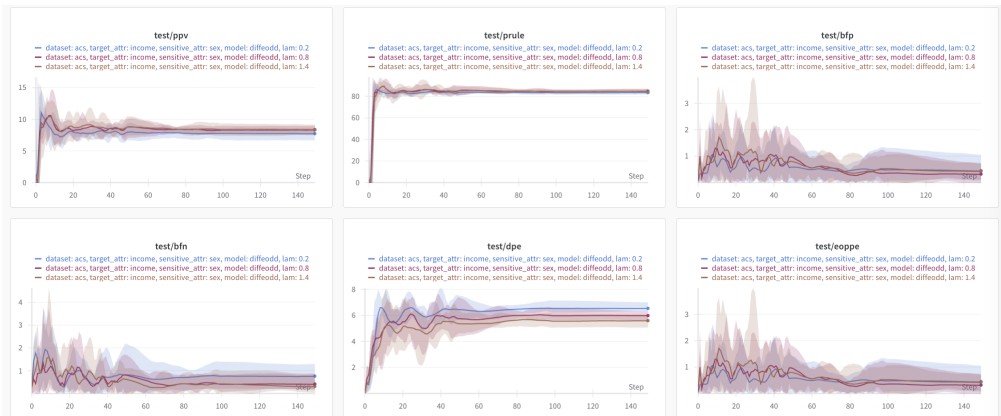

Figure 16: ACC, AUC and $\triangle_{EO}$ (Shown as "eoppe").

## K.2 EOPP ADDITIONAL RESULTS

**Adult (Gender)**

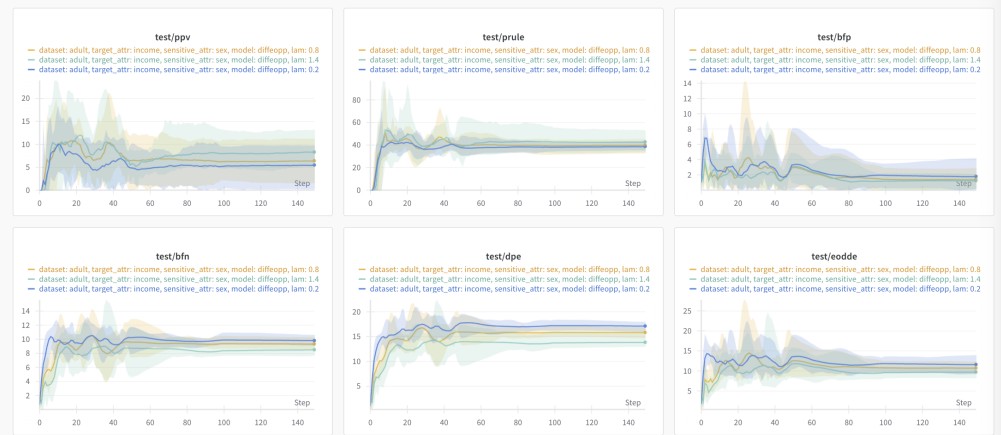

Figure 17: Other metrics ($\triangle_{PPV}$ ($\downarrow$), $PRULE$($\uparrow$), $\triangle_{BFP}$ ($\downarrow$), $\triangle_{BFN}$ ($\downarrow$) ), $\triangle_{DP}$ (Shown as 'dpe'), and eodde.

**Adult (Gender)**

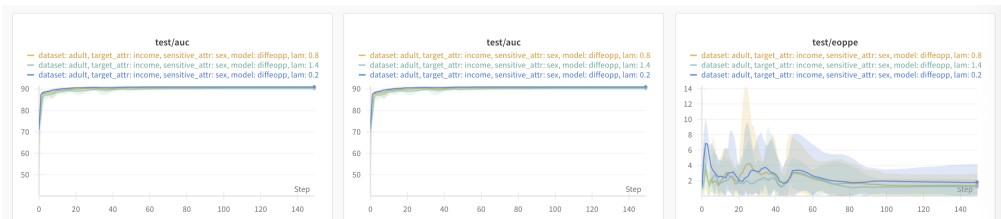

Figure 18: ACC, AUC and $\triangle_{EO}$ (Shown as "eoppe").

**ACS-I (Gender) ACS-I (Gender)**

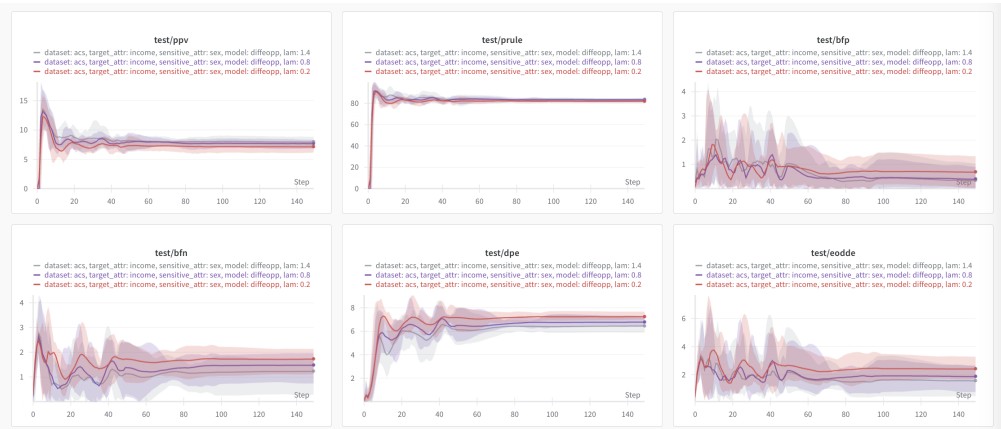

Figure 19: Other metrics ($\triangle_{PPV}$ ($\downarrow$), $PRULE$($\uparrow$), $\triangle_{BFP}$ ($\downarrow$), $\triangle_{BFN}$ ($\downarrow$) ), $\triangle_{DP}$ (Shown as "dpe"), and eodde.

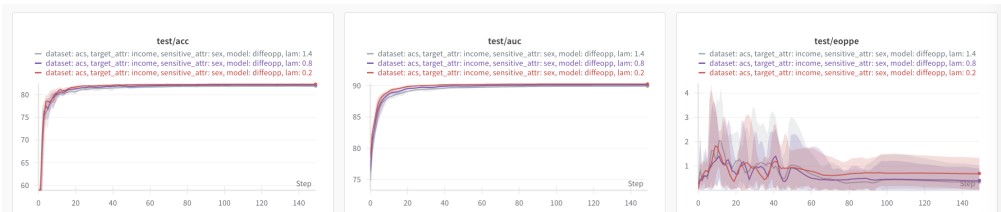

Figure 20: ACC, AUC and $\triangle_{EO}$ (Shown as "eoppe").

- **Dataset**: Adult
- **Sensitive Attribute**: Gender (represented as "Sex")
- **Hyperparameters**: Learning rate = $1 \times 10^{-2}$, $\alpha = 5 \times 10^{-2}$, batch size = 1024, and $\beta = 1.0$ (kept consistent across all regularizers).

**Training Fairness loss: HSIC, CS, and DP**

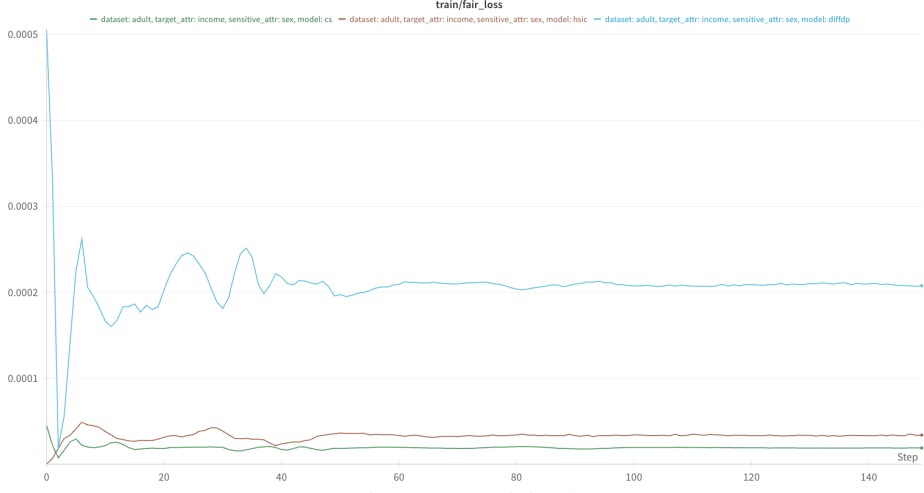

Figure 21: Training loss.

**Training Fairness Loss: HSIC and CS**

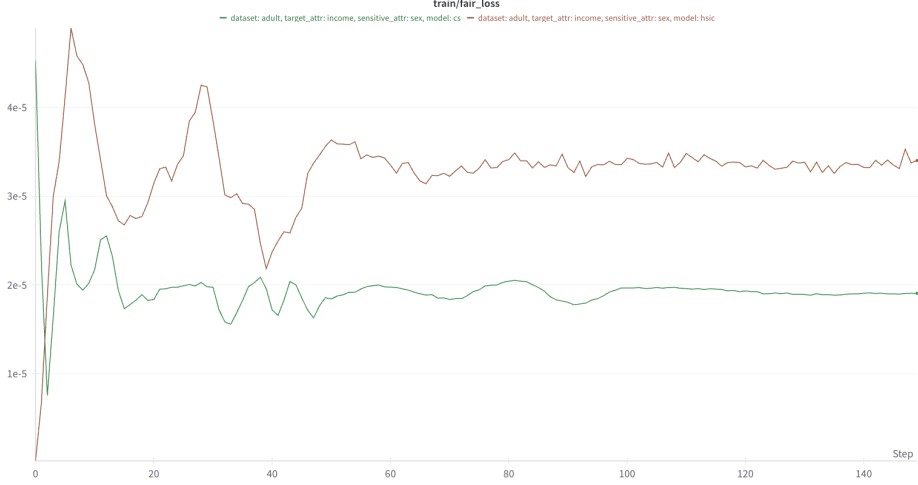

Figure 22: Training loss (excluding DP regularizer line) to more clearly observe the gap between HSIC and CS regularizers.

**Test $\Delta_{DP}$ and $\Delta_{EO}$:**

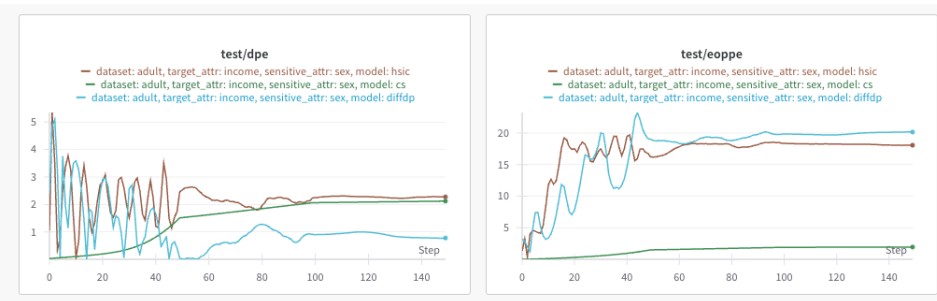

Figure 23: Test $\Delta_{DP}$ and $\Delta_{EO}$.

# L  ADDITIONAL ABLATIONS AND EXPERIMENTS WITH MULTIPLE SENSITIVE ATTRIBUTES

## L.1  ABLATION OVER KERNEL FAMILIES (GAUSSIAN/ LAPLACIAN/POLYNOMIAL)

Beyond the bandwidth study, we also compare different *kernel families* used inside the CS regularizer. We consider three standard choices:

- **Gaussian (RBF)**: $k_{\mathrm{rbf}}(\mathbf{u}, \mathbf{v}) = \exp\left(-\frac{\|\mathbf{u}-\mathbf{v}\|_2^2}{2\sigma^2}\right)$.

- **Laplacian**: $k_{\mathrm{lap}}(\mathbf{u}, \mathbf{v}) = \exp\left(-\frac{\|\mathbf{u}-\mathbf{v}\|_2}{\sigma}\right)$.

- **Polynomial (degree 2)**: $k_{\mathrm{poly}}(\mathbf{u}, \mathbf{v}) = \left(\gamma\,\mathbf{u}^\top\mathbf{v} + 1\right)^2, \quad \gamma = 1/\sigma$.

On the Adult–Income task (sensitive attribute Sex), we keep the model architecture, optimizer, training schedule, and regularization weight $\lambda$ identical to the main experiment, and only change the kernel family used in the CS loss. For a fair comparison, we use the same bandwidth $\sigma_x = \sigma_y = \sigma_{\mathrm{cross}} = 1$ for all three kernels. We summarize the final results from Figure 26 into Table 12.

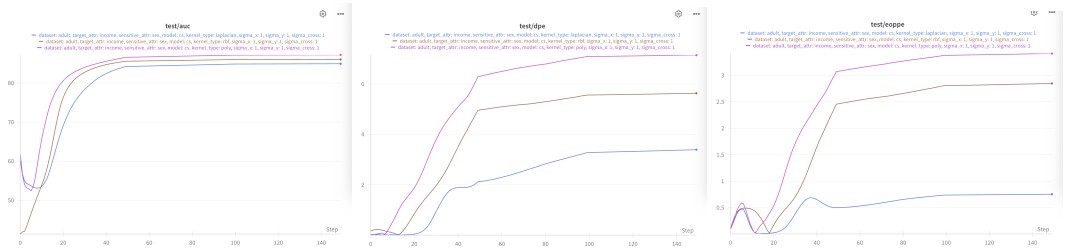

Figure 24: Ablation study: Accuracy and fairness of different kernel functions (Gaussian/Laplacian/polynomial) on `Adult` (sex).

| Kernel type | AUC (%) ↑ | Acc (%) ↑ | $\Delta_{\mathrm{EO}}$ ↓ | $\Delta_{\mathrm{DP}}$ ↓ |
|---|---|---|---|---|
| Laplacian | 84.2 | 83.5 | 0.75 | 3.3 |
| Gaussian RBF | 85.8 | 84.7 | 2.5 | 5.6 |
| Polynomial | 86.3 | 85.1 | 3.3 | 6.5 |

Table 12: Ablation study: `CS` performance on different kernel functions (Gaussian/ Laplacian/polynomial)

From Table 12, all three kernels achieve strong predictive performance (AUC $\approx$ 84–86%), but they trace out different points on the accuracy–fairness trade-off curve:

- The **Laplacian kernel** yields the **smallest EO and DP gaps**, i.e., the best group fairness, at the cost of a small drop in AUC/accuracy.

- The **polynomial kernel** attains slightly **higher AUC/accuracy**, but with noticeably larger EO/DP gaps.

- The **Gaussian (RBF) kernel** lies between the two, offering a **balanced trade-off**: it preserves most of the performance benefits of the polynomial kernel while significantly improving fairness compared to polynomial and remaining closer to Laplacian.

These results confirm that CS is compatible with multiple kernel families and that the choice of kernel can be used to tune the fairness–utility trade-off. In the main paper, we adopt the Gaussian kernel as a default because it provides a **stable, middle-ground trade-off** and is widely used in dependence measures (MMD/HSIC), making our comparison to existing divergences more direct.

## L.2 ABLATION OVER KERNEL BANDWIDTH $\sigma$

To study the effect of the kernel bandwidth in the `CS` regularizer, we fix the model and training setup used on Adult–Income (sensitive attribute Sex) and vary the kernel bandwidths while keeping all other hyperparameters fixed (same optimizer, learning rate, batch size, and $\alpha$ as in the main Adult experiments).

We use an RBF kernel $k_\sigma(\mathbf{u}, \mathbf{v}) = \exp\left(-\frac{\|\mathbf{u}-\mathbf{v}\|_2^2}{2\sigma^2}\right)$, and set $\sigma_x = 1$ for the prediction output, while varying $\sigma_x = \sigma_y = \sigma_{\text{cross}} \in \{0.5, 1, 2, 5, 10, 15, 20\}$ for the sensitive attribute and cross terms in the CS loss. For each configuration we record: (i) the $\Delta_{\text{EO}}$ ('test/eoppe'), (ii) the $\Delta_{\text{DP}}$ ('test/dpe'), (iii) test accuracy, and (iv) test AUC at the final epoch.

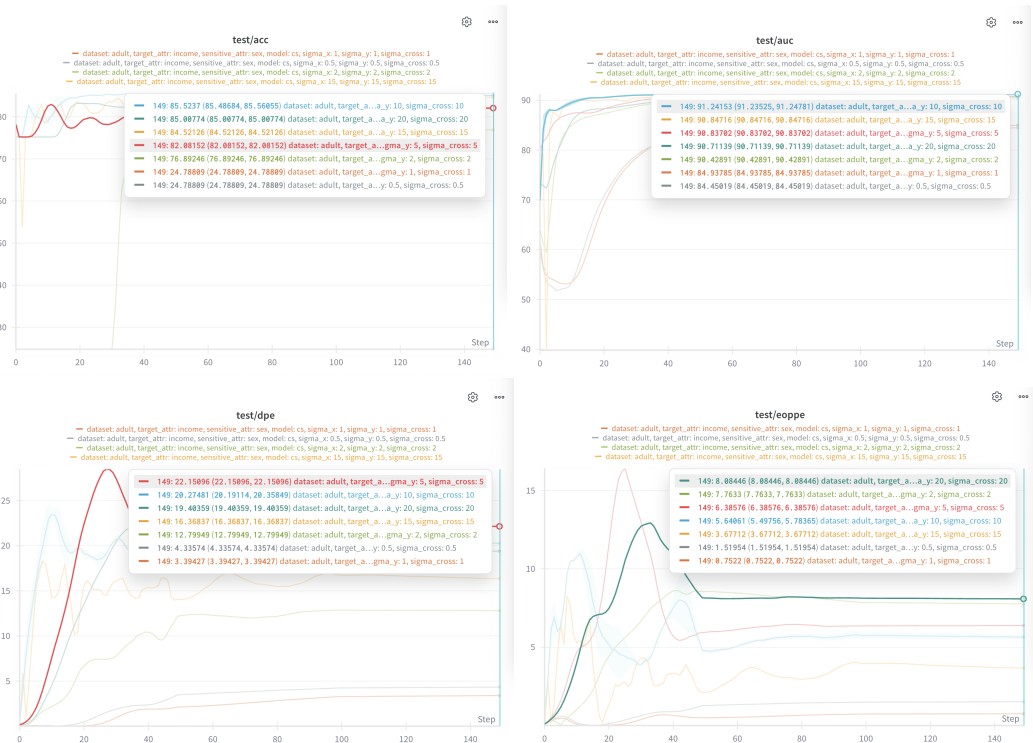

Figure 25: Ablation study: Ablation over kernel bandwidth of `CS` on `Adult` (sex). **How to read these figures: in each subfigure, the lines are ordered in the *legend* from top to bottom according to their values, from highest to lowest.**

From Table 13, we see two observations:

- **Extremely small bandwidths** ($\sigma \leq 1$) Here the RBF kernel becomes extremely peaked. The CS loss forces almost pointwise independence, which makes optimization unstable: accuracy collapses

| $\sigma_y = \sigma_x = \sigma_{\mathrm{cross}}$ | $\Delta_{\mathrm{EO}} \downarrow$ | $\Delta_{\mathrm{DP}} \downarrow$ | ACC (%) $\uparrow$ | AUC (%) $\uparrow$ |
|---|---|---|---|---|
| 0.5 | 1.52 | 4.34 | 24.8 | 84.5 |
| 1 | 0.75 | 3.39 | 24.8 | 84.9 |
| 2 | 7.76 | 12.80 | 76.9 | 90.4 |
| 5 | 6.39 | 22.15 | 82.1 | 90.8 |
| 10 | 5.64 | 20.27 | 85.5 | 91.2 |
| 15 | 3.68 | 16.37 | 84.5 | 90.8 |
| 20 | 8.08 | 19.40 | 85.0 | 90.7 |

Table 13: Ablation study: Kernel bandwidth on the proposed CS fairness regularizer.

to $\approx 25\%$ and AUC drops to $\approx 84\text{--}85\%$. The very small DP/EO gaps in this regime are therefore misleading—they correspond to a nearly random classifier.

- **Moderate bandwidths ($\sigma \in [2, 20]$)** In this regime, the classifier maintains **high utility** (AUC $\approx 90\text{--}91\%$, accuracy between $77\%$ and $85.5\%$). Fairness varies smoothly with $\sigma$:

- $\Delta_{\mathrm{EO}}$ generally improves when moving from too-local kernels ($\sigma = 2, 5$) to more moderate ones; $\sigma = 15$ achieves the smallest $\Delta_{\mathrm{EO}}$ among the stable runs.

- $\Delta_{\mathrm{DP}}$ is best at $\sigma = 2$, but this comes with noticeably lower accuracy. For $\sigma \in \{10, 15, 20\}$, both DP and EO are within a similar, reasonable range while utility is highest.

Overall, the results show that CS is **not hypersensitive** to the exact kernel bandwidth: once $\sigma$ is chosen in a reasonable range, the method achieves consistently high AUC with a stable fairness–utility trade-off. In our main experiments we therefore use a moderate bandwidth (e.g., $\sigma = 10$) that lies in this stable region, balancing strong accuracy (AUC $\approx 91\%$) with substantially reduced $\Delta_{\mathrm{DP}}/\Delta_{\mathrm{EO}}$.

### L.3 EXPERIMENTS WITH MULTIPLE SENSITIVE ATTRIBUTES

To evaluate CS on multi-attribute fairness, we extend the Adult setting from a single sensitive attribute to an **intersectional attribute** combining sex and race. We construct four groups $S \in \{0, 1, 2, 3\}$ as White-Male, White-Female, Non-White-Male, and Non-White-Female. For each group $g$ we define: $\mathrm{DP}_g = \mathbb{P}(\hat{Y} = 1 \mid S = g)$, $\mathrm{EO}_g = \mathbb{P}(\hat{Y} = 1 \mid Y = 1, S = g)$, $\mathrm{Acc}_g = \mathbb{P}(\hat{Y} = Y \mid S = g)$. We then report the **intersectional demographic-parity gap** $\Delta\mathrm{DP}^{\mathrm{inter}} = \max_g \mathrm{DP}_g - \min_g \mathrm{DP}_g$, the **intersectional equal-opportunity gap** $\Delta\mathrm{EO}^{\mathrm{inter}} = \max_g \mathrm{EO}_g - \min_g \mathrm{EO}_g$, and the **worst-group accuracy** $\mathrm{Acc}_{\min} = \min_g \mathrm{Acc}_g$. The code snippet of these metrics are in Appendix N Using the same MLP architecture and $\alpha$ ('lam' in the figure) $= 0.5$, we compare CS with three representative dependence-based regularizers (diffDP, HSIC, diffEOpp). The results are summarized below:

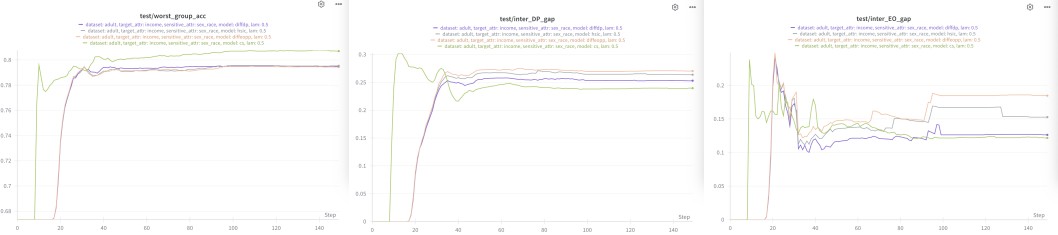

Figure 26: Evaluation of CS on the Adult (sex) as the sensitive attribute.

From Table 14, we observe that CS attains the **smallest** intersectional $\Delta\mathrm{DP}^{\mathrm{inter}}/\Delta\mathrm{EO}^{\mathrm{inter}}$ while also achieving the **highest** worst-group accuracy, indicating that the proposed Cauchy–Schwarz regularizer remains effective even when fairness is evaluated over four intersectional subgroups rather than a single sensitive attribute.

In particular, CS improves both $\Delta\mathrm{DP}^{\mathrm{inter}}$ and $\Delta\mathrm{EO}^{\mathrm{inter}}$ over HSIC and DP regularizer, and slightly improves $\mathrm{Acc}_{\min}$ compared to all baselines. These results suggest that the tighter dependence

| Method | $\Delta\mathrm{DP}^{\mathrm{inter}} \downarrow$ | $\Delta\mathrm{EO}^{\mathrm{inter}} \downarrow$ | $\mathrm{Acc}_{\min} \uparrow$ |
|---|---|---|---|
| diffDP | 0.255 | 0.112 | 0.792 |
| HSIC | 0.262 | 0.152 | 0.791 |
| diffEOpp | 0.268 | 0.183 | 0.790 |
| CS (ours) | **0.241** | **0.108** | **0.803** |

Table 14: Evaluation of `CS` on the `Adult` (sex) as the sensitive attribute.

control provided by `CS` translates into more balanced treatment across intersectional groups without sacrificing worst-case utility.

## M   DISCUSSION: WHEN AND WHY IS CS EXPECTED TO OUTPERFORM OTHER DIVERGENCES?

To address the question of when CS is practically preferable to other dependence measures, we summarize the regimes where CS is theoretically and empirically advantageous:

- **Heavy-tailed or skewed group distributions.** When one group exhibits heavier prediction tails (e.g., more extreme probabilities), KL and DP penalties can be dominated by these tails. In contrast, the CS divergence, through its $L_2$-normalization of group embeddings, limits the influence of such extremes and yields a more stable fairness penalty.

- **Scale-mismatched representations.** When latent embeddings for different groups differ markedly in variance or norm (a common scenario in deep models), Euclidean-based MMD can report large distances even when the group embeddings are well aligned in direction. CS compares *normalized* embeddings and therefore provides a tighter and more meaningful notion of "closeness" for fairness.

- **Imbalanced group sizes.** In highly imbalanced datasets, group-conditional densities are estimated with very different effective sample sizes. In such cases, KL and HSIC can fluctuate considerably with the minority group's empirical variance, whereas the cosine-style normalization implicit in CS makes the fairness loss less sensitive to this sampling noise.

These regimes are not hypothetical: the datasets in Section 5 (Adult, COMPAS, ACS, CelebA-A) all exhibit at least one of these characteristics. This helps explain why CS often achieves lower $\Delta_{\mathrm{DP}}$ and $\Delta_{\mathrm{EO}}$ at comparable or better utility in our experiments.

## N   CODE SNIPPET

```python
def calculate_intersectional_metrics(y_pred, y_target, sensitive):
    if isinstance(y_pred, torch.Tensor):
        y_pred = y_pred.detach().cpu().numpy()
    if isinstance(y_target, torch.Tensor):
        y_target = y_target.detach().cpu().numpy()
    if isinstance(sensitive, torch.Tensor):
        sensitive = sensitive.detach().cpu().numpy()

    y_pred = y_pred.flatten()
    y_target = y_target.flatten()
    sensitive = sensitive.flatten()

    y_pred_binary = (y_pred > 0.5).astype(int)

    groups = [0, 1, 2, 3]

    rates = {}
    for g in groups:
        mask = sensitive == g
        if mask.sum() > 0:
            rates[g] = y_pred_binary[mask].mean()
```

```
22        else:
23            rates[g] = 0.0
24
25    dp_gap = max(rates.values()) - min(rates.values())
26
27    tprs = {}
28    for g in groups:
29        mask = (sensitive == g) & (y_target == 1)
30        if mask.sum() > 0:
31            tprs[g] = y_pred_binary[mask].mean()
32        else:
33            tprs[g] = 0.0
34
35    eo_gap = max(tprs.values()) - min(tprs.values())
36
37    accs = {}
38    for g in groups:
39        mask = sensitive == g
40        if mask.sum() > 0:
41            accs[g] = (y_pred_binary[mask] == y_target[mask]).mean()
42        else:
43            accs[g] = 0.0
44    worst_group_acc = min(accs.values())
45
46    return {
47        "intersectional_DP_gap": dp_gap,
48        "intersectional_EO_gap": eo_gap,
49        "worst_group_acc": worst_group_acc,
50    }
```

Listing 1: Calculation of Intersectional Fairness Metrics

