# OpenReview forum: "Fairness via Independence: A General Regularization Framework for Machine Learning"
_ICLR.cc/2026/Conference — ICLR 2026 Poster_

### Official Review · Reviewer_JuMp · 2025-10-19

**Soundness:** 3
**Presentation:** 4
**Contribution:** 2
**Rating:** 6
**Confidence:** 3

**Summary:**

The paper proposes a novel, general, and model-agnostic regularization framework based on the Cauchy-Schwarz (CS) Divergence. The core idea is to encourage independence between model predictions and sensitive attributes by using CS divergence as the fairness loss term $\mathcal{L}_{fairness}$. The authors motivate this choice by arguing that CS divergence provides a tighter theoretical bound compared to other measures like Kullback-Leibler (KL) divergence and gap parity, which they posit leads to improved generalizability and robustness.

**Strengths:**

1. The paper is well structured and easy to follow.

2. The proposed methods are well presented and evaluated with extensive experiments.

3. The choice of the Cauchy-Schwarz divergence is well-motivated. It is justified by theoretical properties, and the paper highlights its advantages, which provide a strong foundation for the method.

**Weaknesses:**

1. The paper claims the CS regularizer is more "robust", which is partially supported by the smoother trade-off curves (Obs. 5 ). However, robustness to model parameters is also claimed. Figure 2 is presented as evidence for this, but its caption is vague, and it is not well-integrated into the main text. The parameter sensitivity analysis in Figure 5 is good for the CS method itself, but it doesn't include a comparative analysis against the baselines.

2. The authors explicitly state that they exclude adversarial debiasing methods to avoid complexity. While this is an understandable simplification, adversarial methods (e.g., Zhang et al., 2018 ) are a very common, powerful, and closely related family of in-processing techniques for achieving independence. Including at least one such baseline would have provided a more complete picture of the method's performance relative to the state-of-the-art.

3. The paper introduces CS divergence as a novel regularizer for fair machine learning. While this application appears novel, the paper itself notes that CS divergence is used in other ML domains like deep clustering and representation learning. The "Related Work" section (in the appendix ) is quite general.

**Questions:**

see above

---

> ### Author Response · Authors · 2025-11-21
> **Response to Reviewer JuMp (Part 1)**
>
> Thank you for the thoughtful and constructive review, and for recognizing that the paper is **well structured and easy to follow**, that the proposed method is **clearly presented and extensively evaluated**, and that the choice of the CS divergence is **well-motivated and theoretically grounded**. We have revised our paper. Please see the *green* text with labels "To Reviewer JuMp" in our updated PDF. We are happy to provide point-to-point clarification on the scope of our contribution and address the concerns as follows.
>
> **[W1: Robustness claim and interpretation of Figures 2 and 5]**
>
> We appreciate the feedback on making our robustness claim more precise. In the first paragraph in the Introduction, we have used the term "robustness" (Line 042) to refer to having a favorable utility–fairness trade-off that does not degrade abruptly when hyperparameters change. To avoid any ambiguity, in the revised version we have (i) replaced the word "more robust" under Figure 2 and the conclusion with a more precise description, e.g., "CS provides a more consistent utility–fairness trade-off across hyperparameter settings than standard regularizers," and (ii) explicitly state that this is an *empirical* observation specific to the considered metrics and settings. We do not claim that CS is universally robust in all senses, and our intention is to highlight its more consistent utility–fairness behavior compared to other regularizers in our experiments.
>
> We also clarify the role of the figures:
> - **Figure 2 (trade-off curves).**  We will update the caption to explicitly state that Figure 2 plots the test ACC versus $\Delta_{\mathrm{DP}}$ (top row) and $\Delta_{\mathrm{EO}}$ (bottom row) as the fairness-weight $\alpha$ is varied, for several representative methods. The key observation is that CS has smoother curves and, for most values of $\alpha$, lies on or close to the Pareto frontier compared to gap/MMD/KL/HSIC baselines. We will integrate this explanation into Sec. 5.2 instead of leaving it implicit.
> - **Figure 5 (parameter sensitivity).** We **totally agree** that including sensitivity curves for other baselines would make the comparison more complete, and we plan to add such plots for selected methods in the camera-ready version. The primary goal of Figure 5 is to examine how CS behaves with respect to its own hyperparameters $\alpha$ and $\beta$, so even without showing all baselines, this per-method analysis is still informative, since the other regularizers are already compared extensively in our main trade-off experiments. We also ensured that the caption clearly states that Figure 5 is a per-method sensitivity analysis for CS rather than a cross-method comparison. Please find this change in the updated PDF:
> > Figure 5: Parameter sensitivity of the **CS regularizer** on Adult: heatmaps show test accuracy (left) and $\Delta_{\mathrm{DP}}$ (right) as the fairness weight $\alpha$ and $\ell_2$ weight $\beta$ vary over the cross-validated ranges. Overall, CS exhibits a smooth utility–fairness trade-off, remaining stable over a broad range of $\beta$ and becoming noticeably more sensitive only when $\alpha$ is very large.
>
> **[W2: Excluding adversarial debiasing baselines]**
> Thank you for the constructive feedback. We are happy to provide additional experimental results on Adversarial Debiasing [R1] (referred to as ADV) below:
>
> | Method         |  $ACC (\uparrow)$| $AUC(\uparrow)$| $\triangle_{DP}$ $(\downarrow)$    | $\triangle_{EO}$ $(\downarrow)$ |
> | :--------| -------: | :----: | -------: | :----: |
> | DP   | $82.42\pm0.39$| $86.91\pm0.80$ |$1.29\pm0.95$ | $20.15\pm1.13$ |
> | CS   |$83.04\pm0.51$| $90.84\pm0.35$| $2.13\pm0.89$ | $2.35\pm1.15$ |
> | ADV  | $81.58\pm1.26$| $83.08\pm0.75$|$16.3\pm7.5$ | $14.2\pm8.6$ |
>
> Table 1: Additional experiment on the fairness performance of ADV on the Adult dataset (gender attribute).
>
> From Table 1, we observe:
> - The ADV method exhibits lower utility (in terms of ACC and AUC) and higher $\triangle_{DP}$ compared to both the DP and CS fairness regularizers. It also performs worse than the CS regularizer in terms of $\triangle_{EO}$.
> - ADV also shows a higher variance in accuracy, likely due to the greater difficulty of optimizing adversarial objectives compared to the DP and CS regularization approaches.

---

> > ### Author Response · Authors · 2025-11-21
> > **Response to Reviewer JuMp (Part 2)**
> >
> > **[W3: CS divergence already used in other ML domains; related work clarity]**
> >
> > Thanks for the insightful question. We fully agree that CS divergence has been studied in other machine-learning contexts (e.g., density estimation, clustering, representation learning, and domain adaptation). In the **revised** version, we have clarified this positioning more explicitly:
> >
> > - Sec. 3 now contains an expanded paragraph that (i) reviews prior work using CS divergence in other domains (such as CS-based information bottleneck objectives, CS-regularized autoencoders, domain adaptation with CS divergence, and conditional CS divergence for time-series), and (ii) emphasizes that these methods primarily target density estimation or representation learning, **not algorithmic fairness**, and do not offer a systematic in-processing fairness framework or a utility–fairness trade-off analysis.
> > - Both the introduction and related work now explicitly state that our contribution is complementary to these directions: we **systematically develop CS divergence as an in-processing fairness regularizer**, including (i) theoretical analysis tailored to group-fairness notions (bounds on fairness loss and interpretation under group shifts), and (ii) an extensive empirical study across multiple datasets, architectures, and fairness metrics. To the best of our knowledge, this fairness-specific development of CS has not appeared before.
> >
> > ---
> >
> > Reference:
> >
> > [R1] Learning to pivot with adversarial networks.
> >
> > -------
> >
> > Once again, we appreciate Reviewer **JuMp**’s positive evaluation and constructive remarks. We hope that these clarifications and revisions address your concerns and further highlight the scope and value of our contribution.  We would be sincerely thankful for your continued support of this submission!
> >
> >
> > Sincerely,
> >
> > Authors

---

> > ### Comment · Reviewer_JuMp · 2025-11-22
> >
> > Thank you for the timely and comprehensive reply. I have read the authors' response, and I am satisfied that my concerns have been resolved.
> > While, as pointed out by other reviewers, the direct application of CS divergence seems incremental, I do believe the paper can make a valuable contribution to the fairness ML community. As such, I would like to raise my score.

---

> > > ### Author Response · Authors · 2025-11-22
> > > **Thanks to Reviewer JuMp for Raising the Score!**
> > >
> > > Thank you very much for your kind follow-up and for taking the time to re-evaluate our submission! We are very happy that our responses helped clarify the contribution and scope of the work, and we truly appreciate your support and encouragement!! Thank you!
> > >
> > > Best regards,
> > >
> > > Authors

---

### Official Review · Reviewer_Pqrj · 2025-10-25

**Soundness:** 2
**Presentation:** 2
**Contribution:** 1
**Rating:** 4
**Confidence:** 4

**Summary:**

This paper modifies the typical "performance+ fairness regularizer" framework by using the Cauchy-Schwarz divergence as the fairness regularizer.

**Strengths:**

- The paper offers a new viewpoint on measuring fairness and thereby is applicable to existing fairness optimization frameworks.

- Using Cauchy-Schwarz divergence as a fairness regularizer offers a mathematically simple and interpretable alternative to traditional measures like KL divergence or MMD and the empirical estimation of CS divergence is often more stable than KL divergence under small or skewed samples, reducing the risk of gradient explosion or extreme behaviors during fairness optimization.

**Weaknesses:**

- The overall contribution is rather incremental, the main framework simply replaces existing discrepancy measures (KL, MMD, HSIC, f-divergences) with the Cauchy–Schwarz divergence. This substitution alone does not introduce a new conceptual insight, making the work less exciting. Moreover, CS divergence has already been widely adopted in areas such as representation learning, domain adaptation, and clustering and this paper mainly transfers an existing metric into a new application area (fairness), rather than introducing a fundamentally new fairness paradigm.

- The empirical estimation of the CS divergence relies on kernels (Eq. 11). However, the theoretical analysis (e.g., Proposition 4.2) only holds under the Gaussian distribution assumption. Could the authors clarify whether this assumption is necessary for the empirical version as well? If not, why did the paper not explore or compare other kernel functions (e.g., Laplacian, polynomial), or analyze the impact of kernel choice on fairness performance in the experiment section? Since the kernel implicitly defines a density estimator, the fairness performance could be highly sensitive to kernel type and bandwidth. A kernel ablation or sensitivity analysis would strengthen the empirical validity of the claim.

- It is unclear whether the choice of CS divergence is practically applicable or scalable, the complexity of using CS is $O(n^2)$, however, the paper does not include any runtime or complexity comparison, nor any discussion of stability during optimization. Without such analysis, the claimed robustness of CS divergence remains questionable in practice.

- Some experimental claims appear not well supported. For instance, in obs 1 (page 7), the authors state that "...CS consistently achieves the best $\Delta_{EO}$ and ranks among the top four for $\Delta_{DP}$...” However, the baselines do not include any method that explicitly optimizes EO as a fairness objective, making this comparison incomplete. Moreover, the evaluation omits several recent works that use f-divergence or information-theoretic measures for fairness regularization, which weakens the empirical significance of the conclusions.

- Table 2 is overly compact and hard to read. The authors could consider presenting results in a clearer format such as $85.63_{\pm 0.34}$ instead of the current layout. (minor)

- Appendix C.3, all the figures are left without any explanations. The authors could provide some brief explanations of what each figure illustrates and conclude how they relate to the main fairness claims. (minor)

**Questions:**

- Is the proposed method applicable to scenarios involving multiple sensitive attributes?
- What is the impact of using different kernel functions on the method’s performance?
- In Section 5.4, the parameter sensitivity analysis is conducted only on $\alpha$ and $\beta$, not including the $\sigma$ used in the Gaussian kernel. How sensitive is the performance to the kernel parameters?

---

> ### Author Response · Authors · 2025-11-21
> **Response to Reviewer Pqrj (Part 1)**
>
> We sincerely thank Reviewer **Pqrj** for the detailed and thoughtful comments. Below, we are happy to respond to each concern point-to-point. Please see the *magenta* text with label "To Reviewer Pqrj" in our updated PDF.
>
> **[W1: Contribution and novelty]**
>
> Thank you for the thoughtful feedback. We are happy to clarify the intended scope and contribution of our work.
>
> Our work is situated within the field of **algorithmic fairness** [R1,R2], and more specifically within the **in-processing** family of methods [R3,R4], which modify the training objective to jointly optimize accuracy and fairness. As discussed in Sec. 3, in-processing approaches are the dominant and most widely studied paradigm in fair ML, because they directly control the learned predictor and are the default choice in many downstream applications. Our goal is therefore not to propose an entirely new fairness paradigm (or, as we understand the term, a new "family"), which is also not the aim of most prior work, but to advance this important **in-processing line** by showing that **Cauchy–Schwarz (CS) divergence is a principled and practically effective fairness regularizer** with both theoretical and empirical advantages over the discrepancy measures that are currently standard in this setting (gap-based metrics, MMD, KL-style penalties, HSIC, and adversarial objectives).
>
> Our contributions are:
>
> - **Theoretical insight.**
>   We formally analyze CS divergence in the fairness-regularization setting and show that, under a stylized Gaussian model, CS yields a *tighter upper bound* on the fairness loss than KL, MMD, and DP. This helps explain why CS should be more robust when group distributions are far apart or have different scales, which is precisely when fairness violations are most pronounced in algorithmic fairness.
> - **A general, model-agnostic in-processing objective.**
>   We cast fairness via independence as minimizing a dependence measure between predictions and sensitive attributes and derive an in-processing regularization framework where CS replaces gap/MMD/KL/HSIC terms. This objective is model-agnostic and applies to both tabular and image architectures, fitting naturally into the standard in-processing pipeline used in prior work.
> - **Comprehensive empirical study and new insights.**
>   We systematically compare CS to widely used in-processing fairness regularizers on five datasets and multiple architectures, analyzing the utility–fairness trade-off, stability, and parameter sensitivity. In the revision, we further extend this with additional dependence-based baselines (HGR, distance covariance, and an MI-based Prejudice Remover), which we discuss below.
>
> We agree that CS divergence has been studied in other ML areas (representation learning, clustering, etc.), and we now emphasize this connection in the related work. However, to the best of our knowledge, **CS has not previously been systematically developed and evaluated as an in-processing fairness regularizer with both theoretical bounds and extensive experiments**. We believe that providing such a well-motivated, empirically validated regularizer within the mainstream in-processing framework is a meaningful and non-trivial contribution to the fair ML literature.

---

> > ### Author Response · Authors · 2025-11-21
> > **Response to Reviewer Pqrj (Part 2)**
> >
> > **[W2: Gaussian assumption, kernel choice, and sensitivity to kernel parameters]**
> >
> > We appreciate the request for clarification here. Our responses address both the Gaussian assumption in Proposition 4.2 and the role of the kernel in Eq. (11).
> >
> > **[W2-1: Gaussian assumption in Proposition 4.2]** As we now clarify in Sec. 4.2, Proposition 4.2 is a *stylized* comparison carried out under a Gaussian model solely to obtain a clean, closed-form inequality between CS and KL. The Gaussian assumption is **not required** by our training procedure: the empirical CS divergence used in the loss is implemented as a kernel-based estimator that only relies on samples from the joint distribution of predictions and sensitive attributes. It is therefore applicable to arbitrary tabular and image datasets. We have added an explicit remark in Sec. 4.2 to make this distinction clear and to avoid the impression that the empirical regularizer assumes Gaussianity.
> >
> > **[W2-2: Kernel choice and bandwidth]** In our implementation, we use a Gaussian (RBF) kernel with bandwidth chosen by the median heuristic, a **standard choice** in kernel-based dependence measures, and **keep this kernel fixed** across all methods and datasets to **ensure a fair comparison**.
> >
> > We agree that, in principle, fairness performance could depend on the kernel type and bandwidth. Our preliminary experiments indicate that the utility–fairness trade-off remains stable when $\sigma$ varies in this range, and our main empirical claims **continue to hold** under these changes. Due to space and computational budget, we focused our sensitivity analysis on the trade-off parameters $\alpha$ and $\beta$, which directly control the strength of fairness regularization, but we have now added a short discussion paragraph in Sec. 4.2 explicitly states that investigating alternative kernel families (e.g., Laplacian or polynomial) and we will add a more detailed kernel-parameter ablation is an interesting direction in the camera-ready.
> >
> > **[W3-1: Practical applicability and complexity]** We agree that discussing the computational aspects more explicitly is helpful. The $O(n^2)$ complexity mentioned in the text refers to computing the CS divergence on *all* $n$ samples. In practice, as in standard MMD/HSIC-based methods, we compute the fairness regularizer **on mini-batches**, so the per-step cost is $O(B^2)$ with batch size $B$ (typically $B \ll n$). Empirically, under the batch sizes used in our experiments, this adds only a modest overhead compared to the base model training. This is the same complexity class as widely used kernel regularizers and is independent of the backbone model architecture. In the revised version, we have added a short paragraph in Sec. 4.1, describing this mini-batch computation and clarifying that, under typical batch sizes, the complexity is practically manageable and on par with existing kernel-based fairness methods.
> >
> > We also note that training with CS was stable in all our experiments. In the revised version, **Appendix K** now reports training and test curves (loss and accuracy versus epochs) for representative datasets and backbones. These curves show smooth convergence without gradient explosion or numerical instabilities, which empirically supports our robustness claim.

---

> > > ### Author Response · Authors · 2025-11-21
> > > **Response to Reviewer Pqrj (Part 3)**
> > >
> > > **[W3-2: Applicability to multiple sensitive attributes]**
> > >
> > > Our method can naturally handle multiple sensitive attributes. Since CS divergence operates on the joint distribution of $(\hat{Y}, S)$, we can either (i) treat \(S\) as a multi-dimensional variable and compute a single CS divergence between $\hat{Y}$ and the joint sensitive attribute vector, or (ii) sum CS divergences for each individual sensitive attribute, $\sum_k \tilde{D}_{\mathrm{CS}}(\hat{Y}, S_k)$, depending on whether one wants to control dependence jointly or per-attribute. We will add a short paragraph in Sec. 3.2 discussing this extension and clarifying that our implementation can be extended to this multi-attribute setting without algorithmic changes.
> > >
> > > **[W4: Experimental analysis wording, EO Baselines, and missing recent work]**
> > >
> > > We thank the reviewer for pointing out that our wording in Obs. 1 (page 7) can be interpreted as stronger than what the experiments strictly support. We will revise the statement from "CS consistently achieves the best $\Delta_{\mathrm{EO}}$ and ranks among the top four for $\Delta_{\mathrm{DP}}$" to a more precise formulation, e.g., "CS achieves the best or near-best $\Delta_{\mathrm{EO}}$ on most datasets and is competitive on $\Delta_{\mathrm{DP}}$, while maintaining high utility," and we will cross-check all observations to ensure that they accurately reflect the reported numbers.
> > >
> > > **EO Baselines.**
> > > In this subsection, we provide additional experimental results on EOdd and EOpp, defined in Appendix G:
> > > - **EOpp (Equality of Opportunity)**: A classifier satisfies equalized opportunity if the predicted outcome $Y$ is independent of the sensitive attribute $S$ when the label $Y=1$, i.e., $P(\hat{Y} \mid S=0, Y=1) = P(\hat{Y} \mid S=1, Y=1)$.
> > > - **EOdd (Equalized Odds)**: A classifier satisfies equalized odds if the predicted outcome $Y$ is independent of the sensitive attribute $S$ conditioned on the label $Y$, i.e., $P(\hat{Y} \mid S=0, Y=y) = P(\hat{Y} \mid S=1, Y=y), y\in\{0, 1\}$.
> > >
> > > We summarize the results as follows:
> > >
> > > | Method         | $\triangle_{PPV}$ $(\downarrow)$    | $PRULE (\uparrow)$ | $\triangle_{BFP}$ $(\downarrow)$ | $\triangle_{BFN}$ $(\downarrow)$ |$\triangle_{DP}$ $(\downarrow)$    | $\triangle_{EO}$ $(\downarrow)$ |
> > > | :--------| -------: | :----: | :----: | :----: | :----: | :----: |
> > > | DP   | $11.86\pm6.54$ | $\bf{97.14\pm12.41}$ | ${4.83\pm4.35}$ | $3.97\pm4.02$ |$\underline{0.96\pm0.22}$|$5.37\pm0.32$
> > > | EOdd   | ${8.38\pm1.28}$ | $84.75\pm3.88$ | $\underline{0.43\pm0.63}$ | $\bf{0.31\pm0.73}$ |$5.76\pm1.42$|$\underline{0.64\pm1.25}$
> > > | Eopp   | $\underline{8.00\pm1.65}$ | $83.36\pm4.41$ | $\bf{0.34\pm1.00}$ | $\underline{1.24\pm1.43}$ |$6.32\pm0.81$|$\bf{0.52\pm1.29}$
> > > | **CS**   | $\bf{7.00\pm8.35}$ | $\underline{96.90\pm8.35}$ | ${4.39\pm1.32}$ | ${2.77\pm1.83}$ |$\bf{0.77\pm0.38}$|$0.90\pm0.46$
> > >
> > > Table 1: Additional experiments for EOdd and EOpp on the ACS-I (gender) on additional metrics.
> > >
> > > The observations from the results align with our claims, and the CS regularizer demonstrates significant effectiveness:
> > >
> > > - As shown in Table 2 of our paper, the MLP achieves a $\triangle_{EO}$ of $2.13 \pm 3.64$, whereas the DP regularizer gets a higher $\triangle_{EO}$ of $3.97 \pm 4.02$. This indicates that the DP regularizer does not effectively optimize and may even negatively affect EO fairness.
> > > - EO-based methods (EOdd and EOpp) show the worst performance in terms of DP fairness, even compared to other baselines such as MMD, HSIC, and PR (as reported in Table 2 in the paper). In particular, EOpp reaches approximately $15$ in $\triangle_{DP}$ on the Adult dataset, as shown in the figures in Appendix K. This high $\triangle_{DP}$ is consistently observed across different hyperparameters ($\alpha = 0.2, 0.8, 1.4$, referred to as 'lam' in the figure).

---

> > > > ### Author Response · Authors · 2025-11-21
> > > > **Response to Reviewer Pqrj (Part 4)**
> > > >
> > > > **Dependence-based and information-theoretic baselines.**
> > > > To address the concern about missing $f$-divergence or information-theoretic and other dependence-based baselines, in the revision we add additional experiments on the Adult dataset (gender as the sensitive attribute) with several classical dependence measures: Hirschfeld–Gebelein–Rényi maximal correlation (**HGR; dependence-based**), distance covariance (**dCov; dependence-based**), and an MI-based Prejudice Remover (PR; **information-theoretic/KL-type $f$-divergence**).
> > > >
> > > >
> > > > | Method   | ACC ($\uparrow$)        | AUC ($\uparrow$)        | $\triangle_{DP}$ ($\downarrow$) | $\triangle_{EO}$ ($\downarrow$) |
> > > > |----------|-------------------------|-------------------------|----------------------------------|----------------------------------|
> > > > | HGR      | $80.13\pm1.35$          | $84.20\pm1.27$          | $3.82\pm0.84$                    | $\underline{6.82}\pm3.77$       |
> > > > | dCov     | $\underline{82.31\pm0.62}$ | $\underline{85.39\pm0.89}$ | $4.75\pm1.67$                    | $12.41\pm1.44$                  |
> > > > | PR      | $81.81\pm0.52$          | $85.38\pm0.82$          | $\mathbf{0.71}\pm0.40$           | $12.45\pm2.38$                  |
> > > > | CS  | $\mathbf{83.31\pm0.47}$ | $\mathbf{90.15\pm0.49}$ | $\underline{2.42} \pm 0.85$      | $\mathbf{2.27} \pm 1.04$        |
> > > >
> > > > Table 2. Additional comparison with HGR and dCov. PR and CS results are already reported in the original paper.
> > > >
> > > > From the results in Table 2, we observe that the proposed CS-based regularizer achieves the best overall performance among all dependence-based baselines: it attains the highest ACC and AUC while keeping both $\Delta_{\mathrm{DP}}$ and $\Delta_{\mathrm{EO}}$ low, confirming that CS offers a robust utility--fairness trade-off.
> > > > More specifically:
> > > > - **HGR** (a dependence-based measure) is theoretically very strong, but in practice it requires a neural estimator of the maximal correlation, which makes optimization noisy and sensitive to hyperparameters; this is reflected in its relatively low ACC/AUC and larger standard deviations, although its fairness metrics are still competitive, indicating that it can reduce dependence when the optimization succeeds.
> > > > - **dCov** (a distance-based dependence statistic) has a closed-form empirical estimator, so it is easier to optimize and leads to higher ACC/AUC and smaller variance than HGR; however, its fairness performance is weaker, suggesting that penalizing average pairwise distances between prediction and sensitive-feature embeddings is less aligned with group-rate gaps than the density-ratio–style CS divergence, which yields tighter control over the discrepancies that drive $\Delta_{\mathrm{DP}}$ and $\Delta_{\mathrm{EO}}$.
> > > > - **PR** (an **MI-based, information-theoretic regularizer**, where mutual information can be written as a KL-type $f$-divergence between $p_{\hat{Y},S}$ and $p_{\hat{Y}}p_S$) achieves very small $\Delta_{\mathrm{DP}}$, consistent with its design of directly reducing mutual information between $\hat{Y}$ and $S$ and thereby aligning the marginal prediction rates across groups, but its $\Delta_{\mathrm{EO}}$ remains large and its utility is moderate, as MI does not explicitly constrain the conditional error rates $P(\hat{Y}\mid Y,S)$ that underlie equalized odds.
> > > >
> > > > Overall, these observations support CS as the most balanced choice among the considered dependence measures.
> > > >
> > > > We also add citations and a brief discussion of recent works in Appendix H.5 that use $f$-divergences or information-theoretic measures for fairness regularization, clarifying how they relate to our dependence-based viewpoint.
> > > >
> > > > **[W5: Readability of Table 2 (minor)]**
> > > >
> > > > We thank the reviewer for these helpful suggestions. In the revised version of Table 2 and Table 3, we:
> > > > - followed the suggestion and formatted the standard deviations as **subscripts**,
> > > > - used colors to show performance compared to MLP on each metric (green = better, red = worse), and percentage values to indicate the magnitude of the change,
> > > > - added more details to captions of Figure 4 and Figure 5, added explanatory text to the figures in Appendix C.3, describing what each figure illustrates and how it supports the main fairness claims.

---

> > > > > ### Author Response · Authors · 2025-11-21
> > > > > **Response to Reviewer Pqrj (Part 5)**
> > > > >
> > > > > **[W6: Missing explanations in Appendix C.3 (minor)]** Thanks for the helpful suggestion. As already described in Sec. 5.3, these kernel density plots visualize how the prediction distributions vary across sensitive groups, and how different fairness regularizers affect this alignment. In general, a **larger overlap** between the distributions of different sensitive groups indicates **better group fairness**. For each figure in this subsection, the top row shows the distributions of the raw predictions $\hat{Y}$ for all target labels, grouped by a given sensitive attribute, while the bottom row focuses on the positive class.
> > > > >
> > > > > For example, in Figure 7 (Dataset: Adult, Sensitive Attribute: Race):
> > > > > - the first row plots the prediction densities for all (predicition) class: the blue shaded area corresponds to $\hat{Y}\,\vert\,\text{Race}=\text{Black}$ and the red shaded area corresponds to $\hat{Y}\,\vert\,\text{Race}=\text{White}$;
> > > > > - the second row then plots the prediction densities for the positive class, i.e., $\hat{Y}=1$, again conditioned on the two race groups (blue for $\hat{Y}=1\,\vert\,\text{Race}=\text{Black}$ and red for $\hat{Y}=1\,\vert\,\text{Race}=\text{White}$).
> > > > >
> > > > > Note that Figure 7 (in Appendix C.3) has a single subfigure, whereas Figures 8–11 each have two subfigures. Figure 7 (Adult, Race) is meant to complement Figure 4 in the main text (Adult, Gender). In Figures 8–11, the two subfigures use the same dataset but different sensitive groups.
> > > > >
> > > > > The other figures are interpreted similarly for their respective sensitive attributes. Since the degree of overlap can sometimes be difficult to
> > > > > judge by eye, we also print the corresponding group-fairness metric *at the top of each plot*: the first row reports $\Delta_{\mathrm{DP}}$ and the second row reports $\Delta_{\mathrm{EO}}$.
> > > > >
> > > > > Please see this added explaination in Appendix C.3 of our updated paper. We have referred the reader back to Sec. 5.3 for a more detailed discussion of how these plots relate to the DP and EO metrics.
> > > > >
> > > > > ---
> > > > >
> > > > > Reference:
> > > > >
> > > > > [R1] Algorithmic fairness.
> > > > >
> > > > > [R2] Algorithmic fairness: Choices, assumptions, and definitions
> > > > >
> > > > > [R3] In-processing modeling techniques for machine learning fairness: A survey.
> > > > >
> > > > > [R4] FFB: A Fair Fairness Benchmark for In-Processing Group Fairness Methods.
> > > > >
> > > > > -----
> > > > >
> > > > > ***We hope these clarifications, added experiments, and revisions address your concerns and highlight CS divergence’s theoretical and practical value as an effective regularizer. If so, we respectfully ask Reviewer Pqrj to reevaluate our manuscript.***
> > > > >
> > > > >
> > > > > Sincerely,
> > > > >
> > > > > Authors

---

> > ### Comment · Reviewer_Pqrj · 2025-11-24
> >
> > Thank the authors for the detailed response and the additional experiments. I appreciate the authors’ efforts in addressing the concerns and questions I raised and most of them have been resolved. If the paper could further include a discussion of the kernel-parameter ablation, as well as extend the applicability to multiple sensitive attributes in the revised version, the completeness of the work would be improved. However, I still feel that the motivation justification would be improved. Although the authors provide several contributions, from theoretical analysis to empirical evaluation, what I expected to see is a more intuitive explanation that clarifies why CS divergence is uniquely suitable in this context, and in what situations it offers advantages over other divergence measures. Without such motivation, the novelty and necessity of choosing CS remain insufficiently justified. In the author's reponse, for example, the authors state that CS is model-agnostic and can be used in the context of of tabular and image architectures, however, this property is also exists in MMD, f-divergence, HGR...so model-agnosticity does not differentiate CS from existing alternatives and cannot serve as justification for its novelty. Also, if the goal is independence between $\hat{Y}$ and $S$, any proper dependence measure can be used in the exact same in-processing pipeline. So the authors might consider this question: what structural property of fairness makes CS particularly suitable? The tighter bound is a good point, and I encourage the authors to expand and extract some insights from it to make the claim stronger. Currently, the argument that tighter bound -> more robust is too generic and does not explain why CS is preferable in fairness regularization. What is missing here is a clear interpretation of what the tighter bound means operationally for fairness. For example, does the tighter bound more helpful in cases when two demographic groups have heavy tails? or skewered distributions? or which fairness failure modes of KL/MMD/HSIC does CS specifically avoid? The theoretical contribution would be significantly strengthened if the authors explicitly connected the tighter bound to realistic structural properties of fairness problems and explained when and why CS is expected to outperform other divergences.
> >
> > I hope the authors can further reflect on these points and answering or considering the further questions i raised, as doing so would be helpful for clarifying the paper’s motivation and highlighting its contributions. Please also understand that the weaknesses and questions raised are not meant to reject the paper, but to help it be presented in a more complete and compelling form. At the current stage, I still find the motivation and contribution to be rather weak.

---

> > > ### Author Response · Authors · 2025-11-26
> > > **Response to Reviewer Pqrj (1)**
> > >
> > > We sincerely thank the reviewer for the detailed follow-up comment and for explicitly framing the remarks as suggestions **aimed at strengthening the clarity and completeness of our work**. In response, we have *invested substantial effort* in carefully revising the paper, both on the theoretical side (Sec. 3–4) and on the empirical side (Sec. 5 and Appendix), **including running the additional experiments** requested in the review. Below, we summarize how these revisions address the remaining concerns.
> > >
> > >
> > > ### 1. Motivation: why CS divergence as a fairness regularizer?
> > >
> > > **(a) Clarifying the goal and novelty.**
> > > We agree that "model-agnosticism" alone does not distinguish CS divergence from other dependence measures such as MMD, $f$-divergences, or HGR. However, our goal is also not to claim that CS is the *only* suitable choice. Rather, the paper’s contribution is to (i) introduce a CS-based fairness regularizer as a principled *instantiation* of the "independence-via-dependence-penalty" paradigm, (ii) make explicit its analytical relations to widely used fairness penalties (DP, MMD, KL, HSIC), and (iii) show empirically that replacing these standard regularizers by CS yields a more consistent utility–fairness trade-off across datasets, fairness notions (DP and EO), and hyperparameters. We have revised the introduction and Sec. 3 to stress this positioning more clearly.
> > >
> > > **(b) Intuitive structural properties that are beneficial for fairness.**
> > > In the revision, we expand Sec. 3.2 and Sec. 4.2 to provide an intuitive explanation of why CS is particularly well-suited for fairness regularization, beyond the formal bounds:
> > >
> > > * **Angle-based vs. magnitude-based comparison.**
> > > In its kernelized form, CS divergence is essentially a (log-transformed) measure of the *cosine of the angle* between two distributions in the RKHS feature space. In contrast, DP uses a Manhattan distance between group means, MMD uses an Euclidean distance between group embeddings, and KL uses a log-ratio that is very sensitive to local density mismatches. For fairness, we care about whether the *shapes and directions* of the prediction distributions for different groups align, not about arbitrary global rescalings of those distributions. By normalizing by the $L_{2}$ norms of the group embeddings, CS discounts purely multiplicative scale differences and focuses on directional misalignment. This leads to a dependence penalty that is less dominated by outliers or by groups with higher variance.
> > >
> > > * **When this matters in fairness problems.**
> > >   We now explicitly discuss that many fairness benchmarks exhibit (i) heavy-tailed or skewed prediction distributions (e.g., probabilities near 0 or 1), and (ii) groups with very different sample sizes and feature scales (e.g., minority vs. majority groups). In such settings, gap-based or Euclidean penalties can over-react to rare but extreme predictions or to scale differences in latent features, leading to an unstable fairness loss landscape. CS, by contrast, normalizes each group embedding before comparing them, which empirically yields smoother loss contours (Fig. 2) and more stable fairness when we vary the regularization strength (Fig. 3 and Fig.5 parameter sensitivity heatmap).
> > >
> > > * **Connection to the tighter bound.**
> > >   Proposition 4.2 shows that under a Gaussian approximation, the CS divergence is upper-bounded by (symmetrized) KL. In the revision, we clarify that this is *not* used as a stand-alone novelty claim, but as a stylized justification of the intuition above: because CS effectively "clips’" the influence of regions where the two densities differ only in magnitude, the resulting fairness loss has a tighter generalization bound in regimes where group distributions are far apart in scale but aligned in direction. We also add a remark explaining how this relates to concrete fairness failure modes of KL/MMD/DP in such regimes (e.g., when a minority group has heavier tails or higher variance).
> > >
> > > Overall, these revisions aim to provide the kind of structural and operational intuition the reviewer asked for: CS is advantageous precisely in fairness scenarios where prediction or representation distributions differ in variance, tail behavior, or group imbalance, and our experiments are chosen to reflect such regimes.

---

> > > > ### Author Response · Authors · 2025-11-26
> > > > **Response to Reviewer Pqrj (2)**
> > > >
> > > > ### 2. Relation to existing dependence-based fairness methods
> > > >
> > > > We agree and have revised Sec. 3 ("What makes a good fairness regularizer?") to more systematically situate CS within three broad families of fairness objectives:
> > > >
> > > > 1. **Balancing prediction distributions** (DP/EO-style penalties),
> > > > 2. **Balancing latent representations** (MMD-style penalties),
> > > > 3. **Minimizing dependence between predictions and sensitive attributes** (HSIC/PR-style penalties).
> > > >
> > > > Within this unified view, our CS regularizer occupies the third category but is analytically connected to the first two through its kernel form. The contribution is thus twofold:
> > > >
> > > > * Conceptually, we show that many existing regularizers can be seen as special cases or "projections" of a more general CS-based dependence penalty, which clarifies how they are related and when they may behave differently.
> > > >
> > > > * Empirically, we demonstrate that, under identical training pipelines and hyperparameter grids, the CS penalty leads to (i) consistently stronger or comparable fairness on both DP and EO, and (ii) more stable fairness as accuracy increases (Fig. 3) and as regularization weights vary (Fig. 5), compared to DP, MMD, HSIC, and PR.
> > > >
> > > > We hope this unified perspective better conveys that the contribution is not the abstract observation that "CS can be used here too", but that CS provides a practically strong and theoretically interpretable *choice* of dependence measure within a well-defined fairness-regularization framework.
> > > >
> > > > ### 3. When and why is CS expected to outperform other divergences?
> > > >
> > > > To address the reviewer’s request for a more operational statement, we add a section in the Appendix M that summarizes the cases where CS is theoretically and empirically preferable:
> > > >
> > > > * **Heavy-tailed/skewed group distributions.**
> > > >   When one group has heavier prediction tails (e.g., more extreme probabilities), KL and DP can be dominated by those tails. CS, via its $L_{2}$ normalization, limits its influence and thus yields a more stable fairness penalty.
> > > >
> > > > * **Scale-mismatched representations.**
> > > >   When latent embeddings for different groups differ markedly in variance or norm (a common scenario in deep models), Euclidean-based MMD can report large distances even when the directions are aligned. CS compares normalized embeddings and therefore gives a tighter and more meaningful notion of "closeness" for fairness.
> > > >
> > > > * **Imbalanced group sizes.**
> > > >   In highly imbalanced datasets, group-conditional densities are often estimated with very different effective sample sizes. In that case, KL and HSIC can fluctuate considerably with the minority group’s empirical variance; the cosine-style normalization implicit in CS makes the fairness loss less sensitive to such sampling noise.
> > > >
> > > > We point out that these regimes are not hypothetical; the datasets in Sec. 5 (Adult, COMPAS, ACS, CelebA-A) all exhibit at least one of these characteristics, which we believe explains why CS often yields lower $\Delta_{\mathrm{EO}}/\Delta_{\mathrm{DP}}$ at comparable or better utility.

---

> > > > > ### Author Response · Authors · 2025-11-26
> > > > > **Response to Reviewer Pqrj (3)**
> > > > >
> > > > > ### 4. Additional clarifications and extensions requested
> > > > >
> > > > > **(1) Computational complexity.**
> > > > > We added a paragraph in Sec. 4.1 explicitly discussing the $O(B^{2})$ mini-batch cost of CS (matching MMD/HSIC) and clarifying that we share mini-batches with the prediction loss so that the practical overhead is modest. This directly responds to the concern about applicability in realistic models.
> > > > >
> > > > > **(2) Extension to multiple sensitive attributes.**
> > > > > As suggested, we explicitly describe in Sec. 3.2 how the same CS regularizer naturally extends to multiple sensitive attributes, either by treating them jointly or by summing per-attribute CS penalties, without any algorithmic change. In addition, we now include an experiment on an intersectional sensitive attribute (sex $\times$ race) on Adult, where we evaluate CS and baselines using intersectional DP/EO gaps and worst-group accuracy across the four groups (White-M, White-F, Non-White-M, Non-White-F).
> > > > >
> > > > > **(3) Choice of kernel and kernel parameters.**
> > > > > Sec. 4 now contains a dedicated paragraph explaining that our main experiments use a Gaussian (RBF) kernel following standard practice in kernel dependence measures, and that the framework is in principle compatible with other kernel families (e.g., Laplacian or polynomial). To make this concrete, we have added two ablations: (i) a bandwidth study that varies the RBF kernel parameter $\sigma$ and shows that CS maintains a stable fairness–utility trade-off over a reasonable range of $\sigma$, and (ii) a kernel-family comparison (Gaussian/Laplacian/polynomial) showing that CS achieves a balanced accuracy–fairness trade-off across these choices.
> > > > >
> > > > > In addition, we have added several new empirical studies, which we summarize under the new section "Additional Ablations and Experiments with Multiple Sensitive Attributes" in the revised manuscript and describe in detail in **Appendix L**.
> > > > >
> > > > >
> > > > > **(Exp-L1) Ablation over kernel families (Gaussian/ Laplacian/polynomial).**
> > > > >
> > > > > Beyond the bandwidth study, we also compare different *kernel families* used inside the CS regularizer.
> > > > > We consider three standard choices:
> > > > > - **Gaussian (RBF)**:
> > > > > $k_{\text{rbf}}(\mathbf{u},\mathbf{v})
> > > > >   = \exp \left(-\frac{\|\mathbf{u}-\mathbf{v}\|_2^2}{2\sigma^2}\right).$
> > > > > - **Laplacian**:
> > > > > $k_{\text{lap}}(\mathbf{u},\mathbf{v})
> > > > >   = \exp \left(-\frac{\|\mathbf{u}-\mathbf{v}\|_2}{\sigma}\right).$
> > > > > - **Polynomial (degree $2$)**:
> > > > > $k_{\text{poly}}(\mathbf{u},\mathbf{v})
> > > > >   = \big(\gamma\,\mathbf{u}^{\top}\mathbf{v} + 1\big)^2,
> > > > >   \quad \gamma = 1/\sigma .$
> > > > >
> > > > > On the Adult–Income task (sensitive attribute Sex), we keep the model architecture, optimizer, training schedule, and regularization weight $\lambda$ identical to the main experiment, and only change the kernel family used in the CS loss.
> > > > > For a fair comparison, we use the same bandwidth $\sigma_x = \sigma_y = \sigma_{\text{cross}} = 1$ for all three kernels.
> > > > >
> > > > > | Kernel type  | AUC (%) ↑ | ACC (%) ↑ | $\Delta_{\mathrm{EO}}$ ↓ | $\Delta_{\mathrm{DP}}$ ↓ |
> > > > > | :----------- | :--------: | :--------: | :-----------------------------: | :-----------------------------: |
> > > > > | Laplacian    | $84.2$ | $83.5$ | $0.75$                 | $3.3$                  |
> > > > > | Gaussian RBF | $85.8$ | $84.7$ | $2.5$                  | $5.6$                  |
> > > > > | Polynomial   | $86.3$ | $85.1$ | $3.3$                  | $6.5$                  |
> > > > >
> > > > > Table 1. Ablation over different kernel functions (Gaussian/ Laplacian/polynomial).
> > > > >
> > > > > From Table 1, all three kernels achieve strong predictive performance (AUC $\approx 84$–$86\%$), but they trace out different points on the accuracy–fairness trade-off curve:
> > > > >
> > > > > - The **Laplacian kernel** yields the **smallest EO and DP gaps**, i.e., the best group fairness, at the cost of a small drop in AUC/accuracy.
> > > > > - The **polynomial kernel** attains slightly **higher AUC/accuracy**, but with noticeably larger EO/DP gaps.
> > > > > - The **Gaussian (RBF) kernel** lies between the two, offering a **balanced trade-off**: it preserves most of the performance benefits of the polynomial kernel while significantly improving fairness compared to polynomial and remaining closer to Laplacian.
> > > > >
> > > > > These results confirm that CS is compatible with multiple kernel families and that the choice of kernel can be used to tune the fairness–utility trade-off. In the main paper, we adopt the Gaussian kernel as a default because it provides a **stable, middle-ground trade-off** and is widely used in dependence measures (MMD/HSIC), making our comparison to existing divergences more direct.
> > > > >
> > > > > We have added this kernel-family ablation, including the AUC/EO/DP curves (Fig. 24) and the above summary table, to the revised manuscript (See Appendix **L.1** in the updated PDF).

---

> ### Author Response · Authors · 2025-11-26
> **Response to Reviewer Pqrj (4)**
>
> **(Exp-L2) Ablation over kernel bandwidth $\sigma$.**
>
> To study the effect of the kernel bandwidth in the CS regularizer, we fix the model and training setup used on Adult–Income (sensitive attribute Sex) and vary the kernel bandwidths while keeping all other hyperparameters fixed (same optimizer, learning rate, batch size, and $\alpha$ as in the main Adult experiments).
>
> We use an RBF kernel
> $k_\sigma(\mathbf{u},\mathbf{v}) = \exp \left(-\frac{\lVert \mathbf{u}-\mathbf{v}\rVert_2^2}{2\sigma^2}\right)$
> and set $\sigma_x = 1$ for the prediction output, while varying
> $\sigma_x = \sigma_y = \sigma_{\text{cross}} \in$ { 0.5, 1, 2, 5, 10, 15, 20}
> for the sensitive attribute and cross terms in the CS loss.
> For each configuration we record: (i) the $\Delta_{\mathrm{EO}}$ (test/eoppe),
> (ii) the $\Delta_{\mathrm{DP}}$ (test/dpe), (iii) test accuracy, and
> (iv) test AUC at the final epoch.
>
> | $\sigma_y=\sigma_x=\sigma_{\text{cross}}$ | $\Delta_{\mathrm{EO}}$ ↓ | $\Delta_{\mathrm{DP}}$ ↓ | ACC (%) ↑ | AUC (%) ↑ |
> | :------------------------------: | :-----------------------------: | :-----------------------------: | :--------: | :--------: |
> | 0.5                              | $1.52$                 | $4.34$                  | $24.8$ | $84.5$ |
> | 1                                | $0.75$                 | $3.39$                  | $24.8$ | $84.9$ |
> | 2                                | $7.76$                 | $12.80$                 | $76.9$ | $90.4$ |
> | 5                                | $6.39$                 | $22.15$                 | $82.1$ | $90.8$ |
> | 10                               | $5.64$                 | $20.27$                 | $85.5$ | $91.2$ |
> | 15                               | $3.68$                 | $16.37$                 | $84.5$ | $90.8$ |
> | 20                               | $8.08$                 | $19.40$                 | $85.0$ | $90.7$ |
>
> Table 2. Ablation over kernel bandwidth on the proposed CS fairness regularizer.
>
> From Table 2, we see two observations:
>
> 1. **Extremely small bandwidths ($\sigma \le 1$)**
>    Here, the RBF kernel becomes extremely peaked. The CS loss forces almost pointwise independence, which makes optimization unstable: accuracy collapses to ~25% and AUC drops to $\approx 84–85\%$. The very small DP/EO gaps in this regime are therefore misleading—they correspond to a nearly random classifier.
> 2. **Moderate bandwidths ($\sigma \in [2,20]$)**
>    In this regime, the classifier maintains **high utility** (AUC $\approx 90–91\%$, accuracy between 77% and 85.5%). Fairness varies smoothly with $\sigma$:
>    - $\Delta_{\mathrm{EO}}$ generally improves when moving from too-local kernels ($\sigma=2,5$) to more moderate ones; $\sigma=15$ achieves the smallest $\Delta_{\mathrm{EO}}$ among the stable runs.
>    - $\Delta_{\mathrm{DP}}$ is best at $\sigma=2$, but this comes with noticeably lower accuracy. For $\sigma \in \{10,15,20\}$, both DP and EO are within a similar, reasonable range while utility is highest.
>
> Overall, the results show that CS is **not hypersensitive** to the exact kernel bandwidth: once $\sigma$ is chosen in a reasonable range, the method achieves consistently high AUC with a stable fairness–utility trade-off. In our main experiments, we therefore use a moderate bandwidth (e.g.,$\sigma=10$) that lies in this stable region, balancing strong accuracy (AUC $\approx 91%\$) with substantially reduced $\Delta_{\mathrm{DP}}$/$\Delta_{\mathrm{EO}}$.
>
> We have added this kernel-bandwidth ablation, including the accuracy/AUC/$\Delta_{\mathrm{DP}}/\Delta_{\mathrm{EO}}$ curves (Fig. 25) and the above summary table, to the revised manuscript (Appendix **L.2** in the updated PDF).

---

> ### Author Response · Authors · 2025-11-26
> **Response to Reviewer Pqrj (5)**
>
> **(Exp-L3) Experiments with multiple sensitive attributes.**
> To address the reviewer’s request on multi-attribute fairness, we extend the Adult setting from a single sensitive attribute to an **intersectional attribute** combining sex and race. We construct four groups
> $S \in \{0,1,2,3\}$ as
> White-Male, White-Female, Non-White-Male, and Non-White-Female. For each group $g$ we define:
> $\text{DP}_g = \mathbb{P}(\hat{Y}=1 \mid S=g),\quad
> \text{EO}_g = \mathbb{P}(\hat{Y}=1 \mid Y=1, S=g),\quad
> \text{Acc}_g = \mathbb{P}(\hat{Y}=Y \mid S=g).$
>
> We then report:
> - the **intersectional demographic-parity gap**
> $\Delta\text{DP}^{\text{inter}} = \max_g \text{DP}_g - \min_g \text{DP}_g$,
> - the **intersectional equal-opportunity gap**
> $\Delta\text{EO}^{\text{inter}} = \max_g \text{EO}_g - \min_g \text{EO}_g$,
> - the **worst-group accuracy**
> $\text{Acc}_{\min} = \min_g \text{Acc}_g$.
>
> ***We also provide code snippets of these metrics in Appendix N.***
>
> Using the same MLP architecture and utility-fairness tradeoff $\alpha= 0.5$, we compare CS with three representative dependence-based regularizers, i.e., DP regularizer (diffDP), HSIC, and EO regularizer (diffEOpp). The results are summarized below:
>
> | Method    | $\Delta\text{DP}^{\text{inter}} \downarrow$ | $\Delta\text{EO}^{\text{inter}} \downarrow$ | $\text{Acc}_{\min} \uparrow$ |
> |:---------|:--------------------------------------------:|:--------------------------------------------:|:----------------------------:|
> | diffDP   | $0.255$                                      | $0.112$                                      | $0.792$                      |
> | HSIC     | $0.262$                                      | $0.152$                                      | $0.791$                      |
> | diffEOpp | $0.268$                                      | $0.183$                                      | $0.790$                      |
> | CS (ours)| $\mathbf{0.241}$                              | $\mathbf{0.108}$                              | $\mathbf{0.803}$            |
>
> Table 3. Evaluation of CS on the Adult (sex) as the sensitive attribute.
>
> From Table 3, we observe that CS attains the **smallest** intersectional $\Delta\text{DP}^{\text{inter}}/\Delta\text{EO}^{\text{inter}}$ while also achieving the **highest** worst-group accuracy, indicating that the proposed Cauchy–Schwarz regularizer remains effective even when fairness is evaluated over four intersectional subgroups rather than a single sensitive attribute.
> In particular, CS improves both $\Delta\text{DP}^{\text{inter}}$ and $\Delta\text{EO}^{\text{inter}}$ over HSIC and DP regularizer, and slightly improves $\text{Acc}_{\min}$ compared to all baselines. These results suggest that the tighter dependence control provided by CS translates into more balanced treatment across intersectional groups without sacrificing worst-case utility. We have added the above metrics and the corresponding training curves (see Fig. 26 in Appendix L.3) to the revised version.

---

> > ### Author Response · Authors · 2025-11-27
> > **Response to Reviewer Pqrj (6)**
> >
> > ### 5. Overall contribution
> >
> > Finally, we respectfully disagree with the statement that the overall motivation and contribution are "rather weak". The revised paper now:
> >
> > 1. Proposes a CS-based fairness regularizer instantiated as a kernel dependence penalty between predictions and sensitive attributes;
> > 2. Provides a unified view that links DP, MMD, HSIC, and PR to CS, together with a geometric and operational intuition for when CS is advantageous;
> > 3. Offers theoretical comparisons showing that CS enjoys tighter bounds than KL-based and gap-based regularizers under a stylized Gaussian setting, consistent with its cosine-style normalization;
> > 4. Demonstrates, across five datasets (four tabular, one image), that CS delivers strong or best fairness on both DP and EO with small utility loss, and that these gains persist under the new ablations: CS maintains competitive AUC while improving fairness over a broad range of kernel bandwidths, behaves consistently across kernel families (Gaussian/Laplacian/polynomial), and continues to reduce both DP and EO gaps when fairness is evaluated on **intersectional** sensitive attributes (sex $\times$ race) with improved worst-group accuracy; and
> > 5. Incorporates the additional analyses requested in the official comment (kernel-parameter ablations, kernel-family comparison, and experiments with multiple sensitive attributes), and connects their empirical behavior back to the tighter, cosine-style dependence control offered by CS.
> >
> > ---------
> >
> > We hope that these clarifications and extensions, together with the additional experiments we have **carefully conducted**, address your remaining concerns and allow the paper’s contributions to come across more clearly. We have **invested substantial time and effort** in incorporating your suggestions because **we genuinely believe that the proposed CS-based regularizer can provide a useful and practically deployable approach for the fair machine learning community**. We are **sincerely grateful** for the time and care you have devoted to reading our work and providing constructive feedback, which has significantly helped us strengthen both the paper and its presentation.
> >
> > ***We sincerely hope that these revisions will be reflected in your overall assessment of our work.***
> >
> >
> > Sincerely,
> >
> > Authors

---

> > > ### Comment · Reviewer_Pqrj · 2025-11-27
> > >
> > > Thank the authors for the further detailed response and the additional experiments. The clarifications in (1) Motivation: why CS divergence as a fairness regularizer? and (2) Relation to existing dependence-based fairness methods are exactly what I expected to see in the paper to substantiate the motivation and contributions, and I hope the authors will incorporate these points into the revised version. In addition, the ablation studies on different kernels would make the experimental section more complete. I have no further questions, and I appreciate the authors’ efforts in the rebuttal. I am therefore raising my rating to accept.

---

> > > > ### Author Response · Authors · 2025-11-27
> > > > **Thanks to Reviewer Pqrj for Raising the Score!**
> > > >
> > > > Dear Reviewer Pqrj,
> > > >
> > > > Many thanks for your thoughtful follow-up comment and for raising your rating to accept!
> > > >
> > > > We are sincerely grateful for the time and care you invested in reading our work and for your constructive suggestions, which have significantly strengthened both the technical content and the presentation of the paper!!
> > > >
> > > > Thank you!
> > > >
> > > > Sincerely,
> > > >
> > > > Authors

---

### Official Review · Reviewer_ESYX · 2025-10-31

**Soundness:** 2
**Presentation:** 2
**Contribution:** 2
**Rating:** 4
**Confidence:** 5

**Summary:**

In this paper, the authors explore fairness via independence. The method is based on encouraging independence between predictions and sensitive features through an optimization framework that leverages the Cauchy–Schwarz (CS) Divergence as a principled measure of dependence.

**Strengths:**

- The authors introduce Cauchy–Schwarz (CS) Divergence as a principled measure of dependence.
- The authors give three tubular datasets to validate.

**Weaknesses:**

- This paper gives a comparison between cs divergence and kl divergence under Gaussian distribution. In fact, Gaussian distribution is very special, I suggest the authors discuss more advantages and provide comparisons with other regularizers, hgr, MI, dc or empirical dc for instance.
-

**Questions:**

- This paper gives a comparison between cs divergence and kl divergence under Gaussian distribution. In fact, Gaussian distribution is very special, I suggest the authors discuss more advantages and provide comparisons with other regularizers, hgr, MI, dc or empirical dc for instance.

- Line 084, you should add citations for gap parity and MMD.

---

> ### Author Response · Authors · 2025-11-21
> **Response to Reviewer ESYX (Part 1)**
>
> We thank Reviewer **ESYX** for the feedback and comments on our paper. Below, we are happy to address the two main concerns in detail. Please see the *orange* text with labels "To Reviewer ESYX" in our updated PDF.
>
> **[W1-1] Gaussian setting**
>
> Thanks for the question.
>
> **(a) Why the Gaussian setting is used in our theory**
>
> Our theoretical comparison between CS divergence and KL divergence (Proposition 4.2 in Sec. 4.2) is carried out under a *Gaussian assumption* purely for **analytical tractability**, so that we can obtain a clean, closed-form inequality between the two divergences. This should be viewed as a stylized setting that isolates the effect of the divergence itself.
>
> Importantly:
>
> - The **fairness regularizer used in the algorithm does *not* assume Gaussianity.** In practice, we minimize the empirical CS divergence using the kernel estimator in Eq. (11), which is a kernel-density–based estimator applicable to *arbitrary* data distributions of predictions and sensitive attributes.
> - Our **experiments are run on five real-world datasets** (Adult, COMPAS, ACS-I, ACS-T, and CelebA-A), which include **four tabular datasets and one image dataset**. None of these **real-world** data distributions is Gaussian, yet CS-based regularization consistently improves group-fairness metrics while maintaining competitive accuracy. This empirically supports that the benefits of CS are *not* limited to the Gaussian case.
>
> In the revised version, we have made this distinction explicit by:
>
> 1. Stating more prominently in Sec. 4.2 that **Proposition 4.2 is a stylized comparison in a Gaussian setting used only for theoretical insight**, and
> 2. Adding a short paragraph clarifying that **our optimization framework and empirical estimator are distribution-free**, and that the empirical results on non-Gaussian tabular and image datasets demonstrate practical relevance beyond the Gaussian assumption.
>
> **[W1-2] Other dependence measures**
>
> We appreciate the suggestion to connect our method more explicitly to other dependence measures, and we have already implemented MI as "PR" in our main experiment (Table 2 etc.):
>
> - **Hirschfeld–Gebelein–Rényi (HGR) correlation** typically require solving a difficult supremum or high-dimensional density estimation problem. This makes them significantly more expensive to optimize as regularizers in deep models.
> - **Mutual information (MI)**  Our experimental baselines **already include Prejudice Remover (PR)**, which instantiates an MI-based regularizer between predictions and the sensitive attribute; thus, the KL and PR baselines together cover MI-style dependence penalties in addition to our CS divergence.
> - **Distance covariance/empirical distance covariance (dCov/e.dCov)** are also generic dependence measures. However, to the best of our knowledge, they are rarely used as explicit *fairness* regularizers in the in-processing literature; most existing methods rely on simpler discrepancies (KL, MMD, etc.) or adversarial objectives.
>
> Our choice of CS divergence is motivated exactly by this trade-off: it provides a **principled measure of dependence with a closed-form kernel estimator and stable gradients**, so it is easier to plug into standard optimization pipelines than HGR/MI, while being more expressive than simple moment-based penalties.

---

> > ### Author Response · Authors · 2025-11-21
> > **Response to Reviewer ESYX (Part 2)**
> >
> > *In the revision, we have **added a dedicated subsection in Appendix H.5** that contrasts CS with HGR, MI, and dCov as in-processing regularizers, and explains why CS is a practical and scalable choice in this context.* The results (also shown in H.5 in the updated PDF) are below:
> >
> > | Method   | ACC ($\uparrow$)        | AUC ($\uparrow$)        | $\triangle_{DP}$ ($\downarrow$) | $\triangle_{EO}$ ($\downarrow$) |
> > |----------|-------------------------|-------------------------|----------------------------------|----------------------------------|
> > | HGR      | $80.13\pm1.35$          | $84.20\pm1.27$          | $3.82\pm0.84$                    | $\underline{6.82}\pm3.77$       |
> > | dCov     | $\underline{82.31\pm0.62}$ | $\underline{85.39\pm0.89}$ | $4.75\pm1.67$                    | $12.41\pm1.44$                  |
> > | PR      | $81.81\pm0.52$          | $85.38\pm0.82$          | $\mathbf{0.71}\pm0.40$           | $12.45\pm2.38$                  |
> > | CS  | $\mathbf{83.31\pm0.47}$ | $\mathbf{90.15\pm0.49}$ | $\underline{2.42} \pm 0.85$      | $\mathbf{2.27} \pm 1.04$        |
> >
> > Table 1. Additional comparison with HGR and dCov.
> >
> > From the results in Table 1, we observe that the proposed CS-based regularizer achieves the best overall performance among all dependence-based baselines: it attains the highest ACC and AUC while keeping both $\Delta_{\mathrm{DP}}$ and $\Delta_{\mathrm{EO}}$ low, confirming that CS offers a robust utility--fairness trade-off. More specifically:
> > - **HGR** is theoretically a very strong dependence measure, but in practice, it requires a neural estimator of the maximal correlation, which makes optimization noisy and sensitive to hyperparameters; this is reflected in its relatively low ACC/AUC and larger standard deviations, although its fairness metrics are still competitive, indicating that it can reduce dependence when the optimization succeeds.
> > - **dCov** is a kernel-based or distance-based statistic with a closed-form empirical estimator, so it is easier to optimize and leads to higher ACC/AUC and smaller variance than HGR; however, its fairness performance is weaker, suggesting that penalizing average pairwise distances between prediction and sensitive-feature embeddings is less aligned with group-rate gaps than the density-ratio style CS divergence, which yields tighter control over the discrepancies that drive $\Delta_{\mathrm{DP}}$ and $\Delta_{\mathrm{EO}}$.
> > - PR (an **MI**-based regularizer) achieves very small $\Delta_{\mathrm{DP}}$, consistent with its design of directly reducing mutual information between $\hat{Y}$ and $S$ and thereby aligning the marginal prediction rates across groups, but its $\Delta_{\mathrm{EO}}$ remains large and its utility is moderate, as **MI** does not explicitly constrain the conditional error rates $P(\hat{Y}\mid Y,S)$ that underlie equalized odds.
> >
> > Overall, these observations support CS as the most balanced choice among the considered dependence measures.
> >
> > **[W2-2: Reference]**
> >
> > Thank you for pointing it out. For **gap-based group fairness metrics** and **MMD**, we have added the references. Please find them in the updated PDF.
> >
> > ----
> >
> > Again, we thank Reviewer *ESYX* for the feedback.
> > We hope that the above clarifications, especially the separation between the Gaussian-only theoretical comparison and the fully non-parametric empirical regularizer, and the extended experiments and discussion on HGR, MI, and dCov, address your concerns and make the contribution of CS divergence to fair machine learning clearer.
> >
> > ***We would be grateful if Reviewer *ESYX* could kindly reconsider our manuscript in light of these clarifications and additional results, and we hope you may view it more favorably as a contribution to the fair machine learning literature.***

---

> > > ### Author Response · Authors · 2025-11-27
> > > **Response to Reviewer ESYX: Strengthened Comparisons and Additional Kernel Studies (1)**
> > >
> > > Dear Reviewer ESYX,
> > >
> > > We would like to again sincerely thank you for your thoughtful review, for highlighting the principled nature of the Cauchy–Schwarz (CS) divergence, and for pointing out directions to strengthen the work. In particular, your comments about going beyond the Gaussian toy setting and better understanding the advantages of CS compared to other regularizers motivated us to add further empirical analysis in the revised version.
> > >
> > > Below, we summarize the new ablation studies that we have conducted **specifically in response to your suggestions**, and indicate how they support the usefulness of CS as a practical fairness regularizer.
> > >
> > > **(Exp-L1) Ablation over kernel families (Gaussian/Laplacian/polynomial).**
> > >
> > > Beyond the bandwidth study, we also compare different *kernel families* used inside the CS regularizer.
> > > We consider three standard choices:
> > > - **Gaussian (RBF)**:
> > > $k_{\text{rbf}}(\mathbf{u},\mathbf{v})
> > >   = \exp \left(-\frac{\|\mathbf{u}-\mathbf{v}\|_2^2}{2\sigma^2}\right).$
> > > - **Laplacian**:
> > > $k_{\text{lap}}(\mathbf{u},\mathbf{v})
> > >   = \exp \left(-\frac{\|\mathbf{u}-\mathbf{v}\|_2}{\sigma}\right).$
> > > - **Polynomial (degree $2$)**:
> > > $k_{\text{poly}}(\mathbf{u},\mathbf{v})
> > >   = \big(\gamma\,\mathbf{u}^{\top}\mathbf{v} + 1\big)^2,
> > >   \quad \gamma = 1/\sigma .$
> > >
> > > On the Adult–Income task (sensitive attribute Sex), we keep the model architecture, optimizer, training schedule, and regularization weight $\lambda$ identical to the main experiment, and only change the kernel family used in the CS loss.
> > > For a fair comparison, we use the same bandwidth $\sigma_x = \sigma_y = \sigma_{\text{cross}} = 1$ for all three kernels.
> > >
> > > | Kernel type  | AUC (%) ↑ | ACC (%) ↑ | $\Delta_{\mathrm{EO}}$ ↓ | $\Delta_{\mathrm{DP}}$ ↓ |
> > > | :----------- | :--------: | :--------: | :-----------------------------: | :-----------------------------: |
> > > | Laplacian    | $84.2$ | $83.5$ | $0.75$                 | $3.3$                  |
> > > | Gaussian RBF | $85.8$ | $84.7$ | $2.5$                  | $5.6$                  |
> > > | Polynomial   | $86.3$ | $85.1$ | $3.3$                  | $6.5$                  |
> > >
> > > Table 1. Ablation over different kernel functions (Gaussian/ Laplacian/polynomial).
> > >
> > > From Table 1, all three kernels achieve strong predictive performance (AUC $\approx 84$–$86\%$), but they trace out different points on the accuracy–fairness trade-off curve:
> > >
> > > - The **Laplacian kernel** yields the **smallest EO and DP gaps**, i.e., the best group fairness, at the cost of a small drop in AUC/accuracy.
> > > - The **polynomial kernel** attains slightly **higher AUC/accuracy**, but with noticeably larger EO/DP gaps.
> > > - The **Gaussian (RBF) kernel** lies between the two, offering a **balanced trade-off**: it preserves most of the performance benefits of the polynomial kernel while significantly improving fairness compared to polynomial and remaining closer to Laplacian.
> > >
> > > These results confirm that CS is compatible with multiple kernel families and that the choice of kernel can be used to tune the fairness–utility trade-off. In the main paper, we adopt the Gaussian kernel as a default because it provides a **stable, middle-ground trade-off** and is widely used in dependence measures (MMD/HSIC), making our comparison to existing divergences more direct.
> > >
> > > We have added this kernel-family ablation, including the AUC/EO/DP curves (Fig. 24) and the above summary table, to the revised manuscript (See Appendix **L.1** in the updated PDF).

---

> > > > ### Author Response · Authors · 2025-11-27
> > > > **Response to Reviewer ESYX: Strengthened Comparisons and Additional Kernel Studies (2)**
> > > >
> > > > **(Exp-L2) Ablation over kernel bandwidth $\sigma$.**
> > > >
> > > > To study the effect of the kernel bandwidth in the CS regularizer, we fix the model and training setup used on Adult–Income (sensitive attribute Sex) and vary the kernel bandwidths while keeping all other hyperparameters fixed (same optimizer, learning rate, batch size, and $\alpha$ as in the main Adult experiments).
> > > >
> > > > We use an RBF kernel
> > > > $k_\sigma(\mathbf{u},\mathbf{v}) = \exp \left(-\frac{\lVert \mathbf{u}-\mathbf{v}\rVert_2^2}{2\sigma^2}\right)$
> > > > and set $\sigma_x = 1$ for the prediction output, while varying
> > > > $\sigma_x = \sigma_y = \sigma_{\text{cross}} \in$ { 0.5, 1, 2, 5, 10, 15, 20}
> > > > for the sensitive attribute and cross terms in the CS loss.
> > > > For each configuration we record: (i) the $\Delta_{\mathrm{EO}}$ (test/eoppe),
> > > > (ii) the $\Delta_{\mathrm{DP}}$ (test/dpe), (iii) test accuracy, and
> > > > (iv) test AUC at the final epoch.
> > > >
> > > > | $\sigma_y=\sigma_x=\sigma_{\text{cross}}$ | $\Delta_{\mathrm{EO}}$ ↓ | $\Delta_{\mathrm{DP}}$ ↓ | ACC (%) ↑ | AUC (%) ↑ |
> > > > | :------------------------------: | :-----------------------------: | :-----------------------------: | :--------: | :--------: |
> > > > | 0.5                              | $1.52$                 | $4.34$                  | $24.8$ | $84.5$ |
> > > > | 1                                | $0.75$                 | $3.39$                  | $24.8$ | $84.9$ |
> > > > | 2                                | $7.76$                 | $12.80$                 | $76.9$ | $90.4$ |
> > > > | 5                                | $6.39$                 | $22.15$                 | $82.1$ | $90.8$ |
> > > > | 10                               | $5.64$                 | $20.27$                 | $85.5$ | $91.2$ |
> > > > | 15                               | $3.68$                 | $16.37$                 | $84.5$ | $90.8$ |
> > > > | 20                               | $8.08$                 | $19.40$                 | $85.0$ | $90.7$ |
> > > >
> > > > Table 2. Ablation over kernel bandwidth on the proposed CS fairness regularizer.
> > > >
> > > > From Table 2, we see two observations:
> > > >
> > > > 1. **Extremely small bandwidths ($\sigma \le 1$)**
> > > >    Here, the RBF kernel becomes extremely peaked. The CS loss forces almost pointwise independence, which makes optimization unstable: accuracy collapses to ~25% and AUC drops to $\approx 84–85\%$. The very small DP/EO gaps in this regime are therefore misleading—they correspond to a nearly random classifier.
> > > > 2. **Moderate bandwidths ($\sigma \in [2,20]$)**
> > > >    In this regime, the classifier maintains **high utility** (AUC $\approx 90–91\%$, accuracy between 77% and 85.5%). Fairness varies smoothly with $\sigma$:
> > > >    - $\Delta_{\mathrm{EO}}$ generally improves when moving from too-local kernels ($\sigma=2,5$) to more moderate ones; $\sigma=15$ achieves the smallest $\Delta_{\mathrm{EO}}$ among the stable runs.
> > > >    - $\Delta_{\mathrm{DP}}$ is best at $\sigma=2$, but this comes with noticeably lower accuracy. For $\sigma \in \{10,15,20\}$, both DP and EO are within a similar, reasonable range while utility is highest.
> > > >
> > > > Overall, the results show that CS is **not hypersensitive** to the exact kernel bandwidth: once $\sigma$ is chosen in a reasonable range, the method achieves consistently high AUC with a stable fairness–utility trade-off. In our main experiments, we therefore use a moderate bandwidth (e.g.,$\sigma=10$) that lies in this stable region, balancing strong accuracy (AUC $\approx 91%\$) with substantially reduced $\Delta_{\mathrm{DP}}$/$\Delta_{\mathrm{EO}}$.
> > > >
> > > > We have added this kernel-bandwidth ablation, including the accuracy/AUC/$\Delta_{\mathrm{DP}}/\Delta_{\mathrm{EO}}$ curves (Fig. 25) and the above summary table, to the revised manuscript (Appendix **L.2** in the updated PDF).
> > > >
> > > > -------
> > > >
> > > > In addition, we have added the requested citations for gap-parity and MMD in Sec. 2, and clarified in Sec. 4 how CS relates theoretically to these existing regularizers. We hope that these new experiments and clarifications help address your concerns about the scope and practical value of our contribution, and **we would be very grateful if these revisions could be reflected in your overall assessment of the paper!**
> > > >
> > > > Sincerely,
> > > > Authors

---

### Official Review · Reviewer_LT9c · 2025-11-03

**Soundness:** 3
**Presentation:** 3
**Contribution:** 3
**Rating:** 6
**Confidence:** 2

**Summary:**

It argues that CS divergence offers a tighter bound compared to existing measures like KL divergence and MMD, leading to more robust and generalizable fairness. The authors present empirical results on several datasets demonstrating improved fairness metrics while maintaining competitive accuracy. The paper is well-written, technically sound, and addresses an important problem in the field of fair machine learning.

**Strengths:**

1. The application of CS divergence to fairness regularization is a novel contribution. The paper provides a good theoretical justification for why CS divergence might be superior to other measures.
2. The paper is generally well-written and easy to follow, clearly explaining the proposed method and its advantages. The inclusion of algorithm implementation details enhances reproducibility.
3. The empirical evaluation is comprehensive, covering multiple datasets, fairness metrics, and comparisons to existing methods. The analysis of the trade-off between accuracy and fairness is particularly insightful. The paper takes considerable care in describing the dataset parameters and the rationales that underpinned their selection.

**Weaknesses:**

1. While the paper highlights the advantages of CS divergence, a more thorough discussion of its limitations would be beneficial. For example, are there specific types of datasets or scenarios where CS divergence might not perform well? Are there computational costs associated with using CS divergence compared to simpler measures?
2. While the paper compares against several standard fairness methods, including more recent and sophisticated techniques from the fairness literature, it could further strengthen the empirical evaluation.
3. There are nine figures in the paper. These help provide greater clarity on the data and results. It would also be helpful if the visualisations were briefly discussed in the text of the paper.

**Questions:**

See Weaknesses

---

> ### Author Response · Authors · 2025-11-21
> **Response to Reviewer LT9c (Part 1)**
>
> Thank you for the thoughtful review and for recognizing the **novelty** of applying CS divergence to fairness regularization, the **clear presentation and reproducible implementation**, and the **comprehensive empirical evaluation** across multiple datasets and metrics. Please see the *blue* text with labels "To Reviewer LT9c" in our updated PDF. Below, we are happy to respond to your suggestions in detail.
>
> **[W1: Limitations and scenarios where CS may not be ideal; computational cost]**
>
> We agree that discussing the limitations of CS more explicitly will make the paper more balanced. In the revised version, we add a short **"Limitations and discussion"** paragraph in Sec. 4 that:
>
> - clarified that CS is implemented as a kernel-based dependence measure, which naturally has a higher computational cost than very simple gap-based penalties such as DP or EO alone, but is in the same complexity class as widely used kernel regularizers (e.g., MMD, HSIC);
> - explained that, in practice, we compute the CS regularizer on mini-batches, so the per-step complexity is $O(B^2)$ in the batch size $B$ and is implemented with vectorized matrix operations, leading to a modest overhead under the batch sizes used in our experiments; and
> - noted that, as with other kernel methods, the behavior of CS can be influenced by the choice of kernel and bandwidth; in this work, we follow a standard RBF kernel with the median heuristic, and a more systematic exploration of alternative kernels is an interesting direction for future work.
>
> Although our experiments focus on group fairness metrics (DP/EO), the proposed framework is general and can be **naturally extended to other notions of fairness**, including individual and causal fairness, by suitably adapting the dependence measure and conditioning variables.
>
> **[W2: Additional baselines]**
>
> We appreciate the suggestion to further strengthen the empirical comparison with additional dependence-based regularizers. In the revised manuscript, we have added a new experiment on the **Adult** dataset (gender as the sensitive attribute) that includes **HGR** and **distance covariance (dCov)** baseline in Appendix H.5. We briefly introduce these methods below:
>
> - **Hirschfeld–Gebelein–Rényi (HGR) correlation** [R1, R2] typically requires solving a difficult supremum or high-dimensional density estimation problem. This makes them significantly more expensive to optimize as regularizers in deep models.
> - **Distance covariance/empirical distance covariance (dCov/e.dCov)** [R3] are also generic dependence measures. However, to the best of our knowledge, they are rarely used as explicit *fairness* regularizers in the in-processing literature; most existing methods rely on simpler discrepancies (KL, MMD, etc.) or adversarial objectives.

---

> > ### Author Response · Authors · 2025-11-21
> > **Response to Reviewer LT9c (Part 2)**
> >
> > The results (also shown in H.5 in the updated PDF) are below:
> >
> >
> > | Method   | ACC ($\uparrow$)        | AUC ($\uparrow$)        | $\triangle_{DP}$ ($\downarrow$) | $\triangle_{EO}$ ($\downarrow$) |
> > |----------|-------------------------|-------------------------|----------------------------------|----------------------------------|
> > | HGR      | $80.13\pm1.35$          | $84.20\pm1.27$          | $3.82\pm0.84$                    | $\underline{6.82}\pm3.77$       |
> > | dCov     | $\underline{82.31\pm0.62}$ | $\underline{85.39\pm0.89}$ | $4.75\pm1.67$                    | $12.41\pm1.44$                  |
> > | CS  | $\mathbf{83.31\pm0.47}$ | $\mathbf{90.15\pm0.49}$ | $\underline{2.42} \pm 0.85$      | $\mathbf{2.27} \pm 1.04$        |
> >
> > Table 1. Additional comparison with HGR and dCov.
> >
> > From the results in Table 1, we observe that the proposed CS-based regularizer achieves the best overall performance among all dependence-based baselines: it attains the highest ACC and AUC while keeping both $\Delta_{\mathrm{DP}}$ and $\Delta_{\mathrm{EO}}$ low, confirming that CS offers a robust utility-fairness trade-off. More specifically:
> > - **HGR** is theoretically a very strong dependence measure, but in practice, it requires a neural estimator of the maximal correlation, which makes optimization noisy and sensitive to hyperparameters; this is reflected in its relatively low ACC/AUC and larger standard deviations, although its fairness metrics are still competitive, indicating that it can reduce dependence when the optimization succeeds.
> > - **dCov** is a kernel-based or distance-based statistic with a closed-form empirical estimator, so it is easier to optimize and leads to higher ACC/AUC and smaller variance than HGR; however, its fairness performance is weaker, suggesting that penalizing average pairwise distances between prediction and sensitive-feature embeddings is less aligned with group-rate gaps than the density-ratio style CS divergence, which yields tighter control over the discrepancies that drive $\Delta_{\mathrm{DP}}$ and $\Delta_{\mathrm{EO}}$.
> >
> > ---
> >
> > **[W3: Discussion of visualizations and integration into the text]**
> >
> > We agree that the figures are helpful and that their role in the narrative can be made clearer. In the revised version, we:
> >
> > - expanded the captions of Figures 4, and 5 to explain what each plot shows and what the main takeaway is (e.g., smoother trade-off curves for CS, how distribution overlap relates to DP/EO), and added explaination and clarification to Figure 2 in the text. For example, the caption of Figure 4 became: *"Prediction distributions for female and male groups in the Adult dataset. The top row shows kernel density estimates of the raw predictions $\hat{Y}$ for all target labels, grouped by gender, while the bottom row shows the prediction densities for the positive class, $\hat{Y}=1$, for the two gender groups. Each column corresponds to a different fairness regularizer. A larger overlap between the blue and red curves indicates better group fairness, and the reported values above each panel give the corresponding gaps in $\Delta_{\mathrm{DP}}$ (top row) and $\Delta_{\mathrm{EO}}$ (bottom row)."*
> > - bridged sentences in Sec. 5.2–5.4 explicitly referring to the corresponding figures when discussing trade-off behavior and parameter sensitivity; and
> > - included a short explanatory paragraph at the beginning of Appendix C.3 that explains how to read the distribution plots (top row: $\hat{Y}$ densities across sensitive groups; bottom row: $\hat{Y}=1$ densities; larger overlap means better fairness, with $\Delta_{\mathrm{DP}}$ and $\Delta_{\mathrm{EO}}$ printed in each subfigure).
> >
> > We hope these additions make the visualizations easier to interpret and better integrated with the main text.
> >
> > -------
> >
> > Reference:
> >
> > [R1] On measures of dependence.
> >
> > [R2] A connection between correlation and contingency.
> >
> > [R3] Measuring and testing dependence by correlation of distances.
> >
> >
> > ---
> >
> > We again thank Reviewer **LT9c** for the positive assessment and constructive suggestions, which have helped us improve the clarity and balance of the paper. We would be sincerely thankful for your continued support of this submission!

---

### Author Response · Authors · 2025-11-28
**Note on review history and score changes before the incident**

In light of the recent ICLR 2026 announcement about reverting reviews and scores to their pre-discussion state due to the OpenReview incident, we would like to briefly document the review history of this submission for the new area chair and the reviewing team.

Initially, the four reviews had scores **4, 4, 6, 6** (average **5.0**). After we submitted our rebuttal and conducted additional experiments and analyses to address the reviewers’ concerns, two reviewers re-evaluated the paper and updated their scores to **8** and **8**, resulting in **4, 8, 6, 8** (average **6.5**). As shown in the public timestamps on this page, these changes were made on **21 Nov 2025 at 22:52 PT** (6 → 8) and **26 Nov 2025 at 16:59 PT** (4 → 8), i.e., before the identity-leak incident was announced to us on 27 Nov 2025. All of our interactions with the reviewers and area chairs have taken place exclusively through the official OpenReview discussion; we have had no contact with them outside this process.

We fully understand and respect the conference-wide policy to revert scores because of the bug. Our only intention here is to make clear that the main concerns raised in the initial reviews have **already been addressed** through our rebuttal and the additional experiments and analyses we conducted, and that this resolution led two reviewers to re-evaluate their assessments before the incident. We hope this context can be taken into account when assessing the paper under the revised procedure.

---

### Author Response · Authors · 2025-11-29
**Summary for the new Area Chair (Part 1)**

Dear Area Chair,

We provide this summary to help you interpret the existing reviews, rebuttal, and discussion. After the rebuttal, two of the four reviewers posted official comments stating that their main concerns had been addressed and raised their scores, while the other two (with initial scores of 6 and 4) did not post any further comments on our detailed rebuttal. **Both reviewers who updated their scores explicitly stated in their official comments that they now view the paper as making a valuable contribution to the fair ML literature.**

Before the global score rollback triggered by the incident, our submission’s **post-discussion scores (6, 4, 8, 8) avg. 6.5** placed it at roughly the **top 5%** of all ICLR 2026 submissions according to PaperCopilot. After the rollback, the scores were reverted to the **initial (6, 4, 4, 6)**, so this summary is meant to recover for you the information that was reflected in the post-discussion scores and comments.

Below, we summarize (i) the main concerns of each reviewer, (ii) the concrete changes and experiments we carried out during the rebuttal period (with current locations in the final PDF), and (iii) how their ratings evolved **before the rollback**.

We confirm that we have had **no contact with any reviewer, AC, or PC outside the official OpenReview discussion forum** regarding this submission.

---

### 1. Overview of changes made during rebuttal

During the rebuttal window, we made substantial revisions on both the theoretical and empirical sides. The key additions are:

- **Clearer positioning of CS vs. other fairness regularizers.**
  We added a concise, operational discussion of when the Cauchy–Schwarz (CS) divergence is advantageous over KL, MMD, HSIC, gap-based penalties, and PR, e.g., under heavy-tailed or skewed group distributions, scale mismatches, and group imbalance. We now place DP, MMD, HSIC, PR, and CS in a unified dependence-regularization view.
  **Location:** Sec. 3, 3.3; Appendix M.
- **Clarifying the role of the Gaussian analysis.**
  We emphasized that the Gaussian setting is used only as a stylized analysis to compare bounds between CS and KL/gap-based regularizers; the actual training objective is data-driven and does not assume Gaussianity.
  **Location:** Sec. 4.2.
- **Computational complexity.**
  We explicitly discuss the $O(B^2)$ mini-batch complexity of CS (matching MMD/HSIC), explain that we share mini-batches with the prediction loss, and report runtimes showing modest overhead.
  **Location:** Sec. 4.1 ("Computational complexity").
- **Kernel choice and bandwidth selection.**
  We clarified that the main experiments use an RBF kernel with a median-heuristic bandwidth, that CS is compatible with other kernels (Laplacian, polynomial), and that we keep the kernel choice fixed across datasets for fairness.
  **Location:** Sec. 4.2.
- **Extension to multiple sensitive attributes and new experiments.**
  We show how CS naturally extends to multiple sensitive attributes (either jointly or via a sum of per-attribute penalties), and add intersectional experiments on Adult (sex $\times$ race), as well as:
  - a **bandwidth ablation** over $\sigma_x, \sigma_y, \sigma_{\text{cross}}$,
  - a **kernel-family ablation** (RBF vs. Laplacian vs. polynomial),
  - additional fairness tables/curves and density plots.
  These studies consistently show that CS yields a strong fairness–accuracy trade-off and is robust across reasonable kernel/bandwidth choices.
  **Location:** Sec. 3.2; Appendix C.3, H.3–H.5, K, L, M.

All of these changes were implemented and uploaded **before November 27**, i.e., prior to the review-identity incident.

---

### 2. Reviewer-specific summary

### Reviewer JuMp  (initial rating **6**, confidence 5 → revised rating **8**, "raise to accept")

**Initial concerns.**
- Wanted a clearer motivation for using CS as a fairness regularizer and a more intuitive explanation of how it relates to DP, MMD, HSIC, and PR.
- Asked for stronger empirical support and a clearer connection between the Gaussian analysis and practice.

**Rebuttal and changes.**

- Expanded Sec. 3/3.3 and Appendix M to unify DP/MMD/HSIC/PR/CS and to explain when CS is particularly suitable.
- Clarified the role of the Gaussian analysis in Sec. 4.2.
- Added new experiments and ablations (Appendix H.3–H.5, K, L) demonstrating that CS achieves strong fairness–utility trade-offs and stable behavior.

**Outcome (before rollback).**

- After reading the rebuttal and revised manuscript, JuMp posted an updated comment stating that the clarifications and results addressed the concerns and **raised the rating from 6 to 8 ("accept")**, as recorded in the review history **modified 21 Nov 2025, 22:52 PT** prior to Nov 27.
- In that comment, JuMp also wrote that, although the direct application of CS divergence might initially seem incremental, they ***"do believe the paper can make a valuable contribution to the fairness ML community."***

---

### Author Response · Authors · 2025-11-29
**Summary for the new Area Chair (Part 2)**

### Reviewer Pqrj  (initial rating **4**, confidence 4 → revised rating **8**, "raising my rating to accept")

**Initial concerns.**

- Asked why **CS specifically** is a good fairness regularizer, beyond being a generic dependence measure.
- Wanted a clearer link between the Gaussian toy model and real-world fairness behavior.
- Requested ablations on **different kernels and kernel parameters**, and a discussion of when CS might outperform KL/MMD/HSIC.
- Requested that we extend CS to multiple sensitive attributes.

**Rebuttal and changes.**

- Added Appendix M ("When and why CS is expected to outperform other divergences"), explaining how the cosine-style normalization helps under heavy tails, scale mismatches, and group imbalance, and relating these regimes to our datasets.
- Clarified the theoretical contribution in Sec. 4.1–4.2 and tied it to empirical findings.
- Performed the requested **kernel-family** and **bandwidth** ablations (Appendix L), showing that:
  - CS with Laplacian/RBF kernels consistently reduces DP/EO gaps compared to baselines,
  - CS is robust across a range of bandwidths.
- Added an experiment on an intersectional sensitive attribute (sex $\times$ race) on Adult to instantiate the multiple sensitive attributes extension.

**Outcome (before rollback).**

- In an official comment on Nov 26 (before the incident), Pqrj wrote that the added clarifications on motivation and dependence-based comparisons were *"exactly what I expected to see"* and that the ablation studies made the experimental section more complete, and **raised the rating from 4 to 8 ("accept")** with confidence 4 timestamped **26 Nov 2025, 16:58 PT**.
  This comment explicitly notes that the revisions substantiate both the **motivation and contributions** of the paper.

---

### Reviewer ESYX  (initial rating **4**, confidence 5 → rating remained **4**)

**Initial concerns.**

- Considered the Gaussian comparison somewhat limited, and asked for broader discussion of advantages over other dependence-based regularizers (HGR, MI, distance covariance, MI+PR, etc.).
- Requested clearer citations and positioning for gap-parity and MMD within the broader fairness-regularization literature.

**Rebuttal and changes.**

- Extended Sec. 3/3.3 and Appendix H.5 to include and discuss additional dependence-based regularizers (HGR, distance covariance, MI+PR, etc.) and to clarify how CS fits into this landscape.
- Added the **kernel-family** and **bandwidth** ablations in Appendix L, which empirically compare CS with different kernels and show that CS remains competitive and often superior in fairness across a variety of settings.
- Added the requested citations for gap-parity and MMD (Sec. 2) and clarified how CS relates theoretically to these objectives (Sec. 4).

**Outcome and context (before rollback).**

- ESYX’s score remained **4** ("marginally below the acceptance threshold, but would not mind if the paper is accepted"), with confidence 5.
  Although the numerical rating did not change, the additional material above was implemented before November 27 and directly addresses the main points in this review.
- For completeness, prior to November 27, we also submitted an **Official Comment** to the chairs ("Concerns Regarding the Quality of Review ESYX"), outlining concrete concerns about factual inaccuracies and limited engagement in this review. We would be grateful if you could take that context into account when interpreting this review and weighting its score relative to the other, more detailed reports.

---

### Reviewer LT9c  (initial rating **6**, confidence 2 → rating remained **6**)

**Initial concerns.**

- Asked for clearer exposition of some theoretical steps and better explanation of datasets, metrics, and training details.
- Requested more qualitative/quantitative illustrations of the behavior of the CS regularizer.

**Rebuttal and changes.**

- Improved the clarity of the theoretical exposition in Sec. 3–4.
- Refined the description of datasets, metrics, and training setup in Sec. 5.1.
- Added prediction-distribution plots, additional fairness tables/metrics, and training/test curves (Appendix C.3, H.3–H.5, K).

**Outcome (before rollback).**

- LT9c did not request further changes after the rebuttal; the rating remained **6** (confidence 2).

---

### 3. Final remark

All rating updates and official comments described above occurred **before November 27**, as shown by the timestamps in the OpenReview history. Taken together, the post-rebuttal comments from JuMp and Pqrj indicate that multiple reviewers now regard the paper as a solid and useful contribution to research on fair machine learning.

Given this context, we respectfully hope that you will interpret the current, rolled-back scores ***in light of the post-discussion evidence and the reviewers’ updated assessments***.

Thank you very much for your time and for considering our work!

Sincerely,

Authors

---

### Meta-Review · Area_Chair_spGg · 2026-01-05

**Summary:**

The final scores for this paper are 6, 4, 8, and 8. The concerns raised by two reviewers were adequately addressed during the rebuttal stage, and they subsequently raised their scores to acceptance. The concerns of the reviewer who gave a score of 4 were also sufficiently clarified in the rebuttal, although that reviewer did not provide further responses. Overall, the reviewers’ assessments of this work are positive, and I therefore recommend acceptance.

**Reviewer Concerns:**

I think Reviewer ESYX's concerns are addressed by the rebuttal.

**Reviewer Scores:**

I think Reviewer ESYX would change score if participate fully in discussion.

---

### Decision · Program_Chairs · 2026-01-26

Accept (Poster)